# Electrostatics of salt-dependent reentrant phase behaviors highlights diverse roles of ATP in biomolecular condensates

Yi-Hsuan Lin[1,2†‡], Tae Hun Kim[1,2,3,4†§], Suman Das[1,5†], Tanmoy Pal[1], Jonas Wessén[1], Atul Kaushik Rangadurai[1,2,3,4], Lewis E Kay[1,2,3,4], Julie D Forman-Kay[1,2], Hue Sun Chan[1]*

[1]Department of Biochemistry, University of Toronto, Toronto, Canada; [2]Molecular Medicine, Hospital for Sick Children, Toronto, Canada; [3]Department of Molecular Genetics, University of Toronto, Toronto, Canada; [4]Department of Chemistry, University of Toronto, Toronto, Canada; [5]Department of Chemistry, Gandhi Institute of Technology and Management, Visakhapatnam, India

*For correspondence: huesun.chan@utoronto.ca

[†]These authors contributed equally to this work

Present address: [‡]HTuO Biosciences, Vancouver, Canada; [§]Department of Biochemistry, School of Medicine, Case Western Reserve University, Cleveland, United States

Competing interest: The authors declare that no competing interests exist.

## eLife Assessment

In this **important** study, the authors employed three types of theoretical/computational models (coarse-grained molecular dynamics, analytical theory and field-theoretical simulations) to analyze the impact of salt on protein liquid-liquid phase separation. These different models reinforce each other and together provide **convincing** evidence to explain distinct salt effects on ATP mediated phase separation of different variants of caprin1. The insights and general approach are broadly applicable to the analysis of protein phase separation. Still, modeling at the coarse-grained level misses key effects that have been revealed by all-atom simulations, including salt-backbone coordination and strengthening of pi-type interactions by salt.

**Abstract** Liquid-liquid phase separation (LLPS) involving intrinsically disordered protein regions (IDRs) is a major physical mechanism for biological membraneless compartmentalization. The multifaceted electrostatic effects in these biomolecular condensates are exemplified here by experimental and theoretical investigations of the different salt- and ATP-dependent LLPSs of an IDR of messenger RNA-regulating protein Caprin1 and its phosphorylated variant pY-Caprin1, exhibiting, for example, reentrant behaviors in some instances but not others. Experimental data are rationalized by physical modeling using analytical theory, molecular dynamics, and polymer field-theoretic simulations, indicating that interchain ion bridges enhance LLPS of polyelectrolytes such as Caprin1 and the high valency of ATP-magnesium is a significant factor for its colocalization with the condensed phases, as similar trends are observed for other IDRs. The electrostatic nature of these features complements ATP's involvement in π-related interactions and as an amphiphilic hydrotrope, underscoring a general role of biomolecular condensates in modulating ion concentrations and its functional ramifications.

## Introduction

Broad-based recent efforts have uncovered many intriguing features of biomolecular condensates, revealing and suggesting myriad known and potential biological functions (*Shin and Brangwynne, 2017*; *Tsang et al., 2020*; *Lyon et al., 2021*). These assemblies are underpinned substantially, though not exclusively, by liquid-liquid phase separation (LLPS) of intrinsically disordered regions (IDRs) as

well as folded domains of proteins and nucleic acids (*Cinar et al., 2019*; *Bremer et al., 2022*), while more complex equilibrium and non-equilibrium mechanisms also contribute (*Harmon et al., 2017*; *Lin et al., 2018*; *Kato and McKnight, 2018*; *Wurtz and Lee, 2018*; *McSwiggen et al., 2019*; *Lin et al., 2022*; *Mittag and Pappu, 2022*; *Shen et al., 2023*; *Pappu et al., 2023*).

Electrostatics plays major roles in biophysical and biochemical processes (*Honig and Nicholls, 1995*; *Zhou and Pang, 2018*). Because of the relatively high compositions of charge residues in IDRs, electrostatics is particularly important for IDR LLPS (*Nott et al., 2015*; *Lin et al., 2016*), which is often also facilitated by π-related interactions (*Wang et al., 2018*; *Vernon et al., 2018*), hydrophobicity, and hydrogen bonding (*Murthy et al., 2019*; *Cai et al., 2022*), and is modulated by temperature (*Lin et al., 2018*; *Dignon et al., 2019a*), hydrostatic pressure (*Cinar et al., 2019*; *Cinar et al., 2020*), osmolytes (*Cinar et al., 2019*), RNA (*Maharana et al., 2018*; *Tsang et al., 2019*; *Dutagaci et al., 2021*; *Laghmach et al., 2022*), salt, pH (*Wang et al., 2017*), molecular crowding (*André and Spruijt, 2020*; *Patel et al., 2022*; *Kumari et al., 2024*), and post-translational modifications (PTMs; *Shin and Brangwynne, 2017*; *Kim et al., 2019*; *Owen and Shewmaker, 2019*; *Snead and Gladfelter, 2019*). Multivalency underlies many aspects of IDR properties (*Borg et al., 2007*; *Marsh et al., 2012*; *Song et al., 2013*; *Brangwynne et al., 2015*; *Chen et al., 2015*). Here, we focus primarily on how PTM- and salt-modulated multivalent charge-charge interactions might alter IDR condensate behaviors and their possible functional ramifications. In general, electrostatic effects on IDR LLPS (*Nott et al., 2015*; *Dutagaci et al., 2021*; *Wang et al., 2017*; *Wang et al., 2021*) are dependent upon their sequence charge patterns (*Das and Pappu, 2013*; *Sawle and Ghosh, 2015*; *Lin et al., 2017b*; *Lin and Chan, 2017*; *Chang et al., 2017*; *Lin et al., 2019a*; *Amin et al., 2020*; *Lyons et al., 2023*; *Pal et al., 2024a*; *Pal et al., 2024b*). Intriguingly, some IDRs undergo reentrant phase separation (*Cinar et al., 2019*) or dissolution (*Banerjee et al., 2017*) when temperature, pressure (*Cinar et al., 2019*), salt (*Krainer et al., 2021*; *Hong et al., 2022*), RNA (*Banerjee et al., 2017*; *Alshareedah et al., 2019*), or concentrations of small molecules such as heparin (*Babinchak et al., 2020*) is varied. Reentrance, especially when induced by salt and RNA, suggest a subtle interplay between multivalent sequence-specific charge-charge interactions and hydrophobic, non-ionic (*Krainer et al., 2021*; *Hong et al., 2022*), cation-π (*Banerjee et al., 2017*; *Alshareedah et al., 2019*), or π-π interactions.

An important modulator of biomolecular LLPS is adenosine triphosphate (ATP). As energy currency, ATP hydrolysis is utilized to synthesize or break chemical bonds and drive transport to regulate 'active liquid' properties such as concentration gradients and droplet sizes (*Wurtz and Lee, 2018*; *Bertrand and Lee, 2022*). Examples include ATP-driven assembly of stress granules (*Jain et al., 2016*), splitting of bacterial biomolecular condensates (*Guilhas et al., 2020*), and destabilization of nucleolar aggregates (*Hayes et al., 2018*). ATP can also influence biomolecular LLPS without hydrolysis, akin to other LLPS promoters or suppressors (*Nguemaha and Zhou, 2018*; *Ghosh et al., 2019*) that are effectively ligands of the condensate scaffold (*Ruff et al., 2021*), or through ATP's effect on lowering free [Mg$^{2+}$] (*Wright et al., 2019*). Notably, as an amphiphilic hydrotrope (*Martins et al., 2022*) with intracellular concentrations much higher than that required for an energy source, ATP is also seen to afford an important function independent of hydrolysis by solubilizing proteins, preventing LLPS and destabilizing aggregates, as exemplified by measurements on several proteins including fused in sarcoma (FUS) (*Patel et al., 2017*).

Subsequent investigations indicate, however, that hydrolysis-independent [ATP] effects on biomolecular LLPS are neither invariably monotonic for a given system nor universal across different systems. For instance, ATP promotes, not suppresses, LLPS of an IgG1 antibody (*Tian and Qian, 2021*), basic IDPs (*Kota et al., 2024*), and enhances LLPS of full-length and the C-terminal domain (CTD) of FUS at low [ATP] but prevents LLPS at high [ATP] (*Kang et al., 2018*). The latter reentrant behavior has been surmised to arise from ATP binding bivalently (*Kang et al., 2018*; *Kang et al., 2019*) or trivalently (*Ren et al., 2022*) to charged residues arginine (R) or lysine (K) by a combination of cation-π and electrostatic interactions, an effect also seen in the ATP-mediated LLPS of basic IDPs (*Kota et al., 2024*). A similar scenario was invoked for the reentrant phase behavior of transactive response DNA-binding protein of 43 kDa (TDP-43; *Dang et al., 2021*). Most recently, ATP-mediated assembly-disassembly reentrant behavior similar to that of FUS CTD was also observed for the RG/RRG-rich IDR motif with a positive net charge from the heterogeneous nuclear ribonucleoprotein G (*Zhu et al., 2024*).

While π-related interactions are important for biomolecular LLPS in general (*Wang et al., 2018*; *Vernon et al., 2018*) and their interplay with electrostatics likely underlies reentrant biomolecular

phase behaviors modulated by RNA (*Banerjee et al., 2017*; *Alshareedah et al., 2019*) or simple salts (*Krainer et al., 2021*), the degree to which electrostatics alone can, in large measure, rationalize hydrolysis-independent ATP-modulated biomolecular phase reentrance has not been sufficiently appreciated. This question deserves attention. For instance, the suppression of cold-inducible RNA-binding protein condensation by ATP has been suggested to be electrostatically driven (*Zhou et al., 2021*). The aforementioned ATP-modulated reentrant phase behavior of FUS (*Kang et al., 2018*; *Kang et al., 2019*) is reminiscent of the 236-residue N-terminal IDR of DEAD-box RNA helicase Ddx4's lack of LLPS at low [NaCl] (<15–20 mM), LLPS at higher [NaCl] (*Lin et al., 2020*) and decreasing LLPS propensity when [NaCl] is further increased (*Nott et al., 2015*; *Lin et al., 2016*). Indeed, the finding that FUS CTD (net charge per residue (NCPR) = 15/156 = 0.096) exhibits ATP-dependent reentrant phase behaviors while the N-terminal domain (NCPR = 3/267 = 0.011) does not (*Kang et al., 2019*) is consistent with electrostatics-based theory for the difference in salt-dependent LLPS of polyelectrolytes and polyampholytes (*Lin et al., 2020*) and a recent atomic simulation study of direct and indirect salt effects on LLPS (*MacAinsh et al., 2024*).

With this in mind, we seek to delineate the degree to which theories focusing primarily on electrostatics can rationalize experimental ATP-related LLPS data on the 103-residue C-terminal IDR of human cytoplasmic activation/proliferation-associated protein-1 (Caprin1). Full-length Caprin1 (709 amino acid residues) is a ubiquitously expressed phosphoprotein that regulates stress (*Solomon et al., 2007*; *Towers et al., 2011*; *Vu et al., 2021*; *Song et al., 2022*) and neuronal (*El Fatimy et al., 2012*) granules, is necessary for normal cellular proliferation (*Ellis and Luzio, 1995*; *Wang et al., 2005*), and may be essential for long-term memory (*Nakayama et al., 2017*). Caprin1 dysfunction leads to multiple diseases such as nasopharyngeal carcinoma (*Yang et al., 2022*) as well as language impairment and autism spectrum disorder (*Pavinato et al., 2023*), via, for example, Caprin1's modulation of the function of the fragile X mental retardation protein (FMRP; *Tsang et al., 2019*; *Kim et al., 2019*; *El Fatimy et al., 2012*). The C-terminal 607–709 Caprin1 IDR, referred to simply as Caprin1 below, is biophysically and functionally significant: It is sufficient for LLPS in vitro (*Kim et al., 2019*), important for assembling stress granules in the cell (*Solomon et al., 2007*; *Towers et al., 2011*), and has a substantial body of experiments (*Kim et al., 2019*; *Wong et al., 2020*; *Kim et al., 2021*; *Toyama et al., 2022*) for comparison with theory. Since tyrosine phosphorylations of Caprin1 in vivo (*Hornbeck et al., 2015*) may regulate translation in neurons (*Kim et al., 2019*), the Caprin1 system is also useful for gaining insights into phosphoregulation of biomolecular condensates (*Monahan et al., 2017*; *Dignon et al., 2018*; *Carlson et al., 2020*).

Recent advances in theory and computation enable modeling of sequence-specific IDR LLPS (*Lin et al., 2016*; *MacAinsh et al., 2024*; *Dignon et al., 2018*; *Rauscher and Pomès, 2017*; *Das et al., 2018b*; *Das et al., 2018c*; *McCarty et al., 2019*; *Danielsen et al., 2019*; *Choi et al., 2019*; *Das et al., 2020*; *Hazra and Levy, 2020*; *Joseph et al., 2021*). Among the approaches, polymer chain models of IDRs are inherently more realistic in capturing sequence properties than models without a chain description such as patchy particle theory (*Nguemaha and Zhou, 2018*). For chain models, all-atom simulation offers a high degree of geometric and energetic realism (*Rauscher and Pomès, 2017*) but its high computational cost often makes it difficult to achieve sufficient sampling and equilibration for the large system sizes that are needed for modeling biomolecular LLPS processes (*MacAinsh et al., 2024*). However, even coarse-grained explicit-chain simulation affords more realistic geometric and energetic representations than analytical theory, but analytical theory offers significant advantages in numerical tractability (*Lin et al., 2023*). For our present purposes, the analytical rG-RPA formulation (*Lin et al., 2020*), which synthesizes Kuhn-length renormalization (renormalized Gaussian, rG) and random phase approximation (RPA; *Lin et al., 2016*) to treat both high-net-charge polyelectrolytes and essentially net-neutral polyampholytes (*Lin et al., 2020*), is particularly well suited for Caprin1 and its phosphorylated variant pY-Caprin1. To gain deeper insights into the pertinent physical principles and to assess possible limitations of this analytical approximation, we further leverage a methodological combination of rG-RPA (*Lin et al., 2020*), field-theoretic simulation (FTS) (*McCarty et al., 2019*; *Pal et al., 2021*), and coarse-grained explicit-chain molecular dynamics (MD) (*Dignon et al., 2018*; *Das et al., 2020*) to better elucidate the effects of salt, phosphorylation, and ATP on LLPS of Caprin1 and pY-Caprin1.

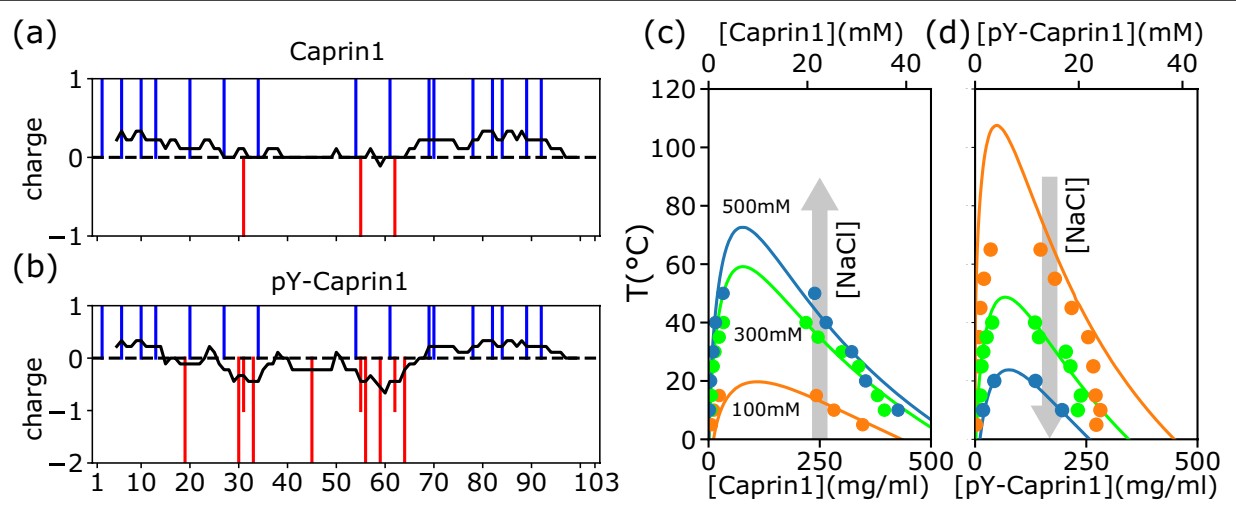

**Figure 1.** rG-RPA+FH theory predictions rationalize different salt dependence of Caprin1 and pY-Caprin1 LLPS. (**a, b**) Vertical lines indicate the sequence positions (horizontal variable) of positively charged residues (blue) and negatively charged residues or phosphorylated tyrosines (red) for (**a**) Caprin1 and (**b**) pY-Caprin1. (**c, d**) rG-RPA+FH coexistence curves (phase diagrams, continuous curves color-coded for the NaCl concentrations indicated) agree reasonably well with experiment (dots, same color code). The grey arrows in (**c, d**) highlight that when [NaCl] increases, LLPS propensity increases for (**c**) Caprin1 but decreases for (**d**) pY-Caprin1. As described in our prior RPA+FH and rG-RPA+FH formulations (*Lin et al., 2016*; *Lin et al., 2020*), the theoretical coexistence curves shown in (**c, d**) are determined by fitting an effective relative permittivity $\epsilon_r$ as well as the enthalpic and entropic parts of a FH parameter $\chi(T) = \epsilon_h/T^* + \epsilon_s$ to experimental data. For the present Caprin1 and pY-Caprin1 systems, the fitted $\epsilon_r = 80.5$, which is remarkably close to that of bulk water ($\epsilon_r \approx 78.5$). The fitted ($\epsilon_h, \epsilon_s$) is (1.0, 0.0) for Caprin1 and (1.0, −1.5) for pY-Caprin1. These fitted energetic parameters are equivalent (*Lin et al., 2016*) to $\Delta H \approx -1.1\,\mathrm{kcal\,mol^{-1}}$ and $\Delta S = 0.0$ for forming a residue-residue contact in the Caprin1 system (**c**) (i.e., it is enthalpically favorable), and $\Delta H \approx -1.1\,\mathrm{kcal\,mol^{-1}}$ and $\Delta S \approx -3.0\,\mathrm{cal\,mol^{-1}K^{-1}}$ for forming a residue-residue contact in the pY-Caprin1 system (**d**) (i.e., it is enthalpically favorable and entropically unfavorable).

The online version of this article includes the following source data for figure 1:

**Source data 1.** Experimental data points and numerical data for the theoretical curves in *Figure 1c and d*.

## Results

### Overview of key observations from complementary approaches

The complementary nature of our multiple methodologies allows us to focus sharply on the electrostatic aspects of the hydrolysis-independent role of ATP in biomolecular condensation by comparing ATP's effects with those of simple salt. Here, Caprin1 and pY-Caprin1 are modeled minimally as heteropolymers of charged and neutral beads in rG-RPA and FTS. ATP and ATP-Mg are modeled as simple salts (single-bead ions) in rG-RPA, whereas they are modeled with more structural complexity as short charged polymers (multiple-bead chains) in FTS, although the latter models are still highly coarse-grained. Despite this modeling difference, rG-RPA and FTS both rationalize experimentally observed ATP- and NaCl-modulated reentrant LLPS of Caprin1 and a lack of a similar reentrance for pY-Caprin1 as well as a prominent colocalization of ATP with the Caprin1 condensate. Consistently, the same contrasting trends in the effect of NaCl on Caprin1 and pY-Caprin1 are also seen in our coarse-grained MD simulations, although polymer field theories tend to overestimate LLPS propensity (*Shen and Wang, 2017*). The robustness of the theoretical trends across different modeling platforms underscores electrostatics as a significant component in the diverse roles of ATP in the context of its well-documented ability to modulate biomolecular LLPS via hydrophobic and π-related effects (*Patel et al., 2017*; *Kota et al., 2024*; *Kang et al., 2019*). Analyses of these other nonelectrostatic effects are mostly beyond the scope of the present work but their impact is nevertheless illustrated by the Flory-Huggins interactions augmented to rG-RPA to quantitatively account for experimental data and our MD simulation of the arginine-to-lysine Caprin1 mutants. These findings are detailed below.

## Physical theories of Caprin1 and phosphorylated Caprin1 LLPSs as those of polyelectrolytes and polyampholytes

The 103-residue Caprin1 is a highly charged IDR with 19 charged residues [*Figure 1a* and Appendix 1, *Appendix 1—figure 1*]: 15 R, 1 K, and 3 aspartic acids (D); fraction of charged residues = 19/103 = 0.184 and NCPR = 13/103 = 0.126. With a substantial positive net charge, Caprin1's phase behaviors are markedly different from those of polyampholytic IDRs with nearly zero net charge such as Ddx4 to which early sequence-specific LLPS theories were targeted (*Nott et al., 2015*; *Lin et al., 2016*). Instead, Caprin1 behaves like chemically synthesized polyelectrolytes (*Dobrynin and Rubinstein, 2005*). In contrast, when most or all of the 7 tyrosines (Y) in the Caprin1 IDR are phosphorylated (pY), negative charges are added to produce a near-net-neutral polyampholyte. Mass spectrometry indicates that the experimental sample of highly phosphorylated Caprin1 consists mainly of a mixture of IDRs with 6 or 7 phosphorylations (*Appendix 1—figure 2*). We refer to this experimental sample as pY-Caprin1 below. For simplicity, we use only the Caprin1 IDR with 7 pYs to model the behavior of this experimental sample in our theoretical/computational formulations, partly to avoid the combinatoric complexity of sequences with 5 or 6 pYs. Accordingly, since the charge of a pY is $\approx$ –2 at the experimental pH = 7.4, –14 charges are added to Caprin1 for our model pY-Caprin1, resulting in a polyampholyte with a very small NCPR = $-1/103 = -0.00971$ (*Figure 1b*). Both the experimental pY-Caprin1 (NCPR $\approx \pm 1/103 = \pm 0.00971$) and model pY-Caprin1 are expected to exhibit phase properties similar to other polyampholytic IDRs.

While sequence-specific RPA has been applied successfully to model electrostatic effects on the LLPSs of various polyampholytic IDRs (*Lin et al., 2018*; *Lin et al., 2016*; *Lin and Chan, 2017*; *Das et al., 2020*; *Wessén et al., 2021*), RPA is less appropriate for polyelectrolytes with large NCPR (*Mahdi and Olvera de la Cruz, 2000*; *Ermoshkin and Olvera de la Cruz, 2003*; *Orkoulas et al., 2003*) because of its treatment of polymers as ideal Gaussian chains (*Muthukumar, 2017*). Traditionally, theories for polyelectrolytes tackle their peculiar conformations by various renormalized blob constructs (*Dobrynin and Rubinstein, 2005*; *Mahdi and Olvera de la Cruz, 2000*), two-loop polymer field theory (*Muthukumar, 1996*), modified thermodynamic perturbation theory (*Budkov et al., 2015*), and renormalized Gaussian fluctuation (RGF) theory (*Shen and Wang, 2017*; *Shen and Wang, 2018*), among others. As such, these formulations are mostly designed for homopolymers, making them difficult to apply directly to heteropolymeric biopolymers. In order to analyze Caprin1 and pY-Caprin1 LLPSs, we utilize rG-RPA (*Lin et al., 2020*), which combines Gaussian chains of effective (renormalized) Kuhn length with the key idea of RGF (*Sawle and Ghosh, 2015*).

## Phase properties predicted by rG-RPA theory for Caprin1 and pY-Caprin1 with monovalent counterions and salt are in agreement with experiment

*Figure 1c and d* show that the salt- and temperature ($T$)-dependent phase diagrams predicted by rG-RPA with an augmented Flory-Huggins (FH) mean-field $\chi(T) = \epsilon_h/T^* + \epsilon_s$ parameter for nonelectrostatic interactions, where $\epsilon_h$ and $\epsilon_s$ are the enthalpic and entropic contributions, respectively, and $T^*$ is reduced temperature (*Lin et al., 2016*; *Lin et al., 2020*; Eq. 10 of *Lin et al., 2016* and 'rG-RPA+FH' theory in *Appendix 1*), are in reasonable agreement with experiment using bulk [Caprin1] (initial overall concentration) $\approx$ 200µM. (Concentrations are provided in molarity and also as mass density in *Figure 1* and subsequent figures.) The rG-RPA+FH results in *Figure 1c* indicate that (i) Caprin1 undergoes LLPS below 20 °C with 100 mM NaCl, and that (ii) LLPS propensity, quantified by the upper critical solution temperature (UCST), increases with [NaCl]. These predictions are consistent with experimental data, including the observation that Caprin1 does not phase separate at room temperature without salt, ATP, RNA, or other proteins, though Caprin1 LLPS can be triggered by adding wildtype (WT) and phosphorylated FMRP and/or RNA (overall [Caprin1] $\gtrsim$ 10 µM) (*Kim et al., 2019*), NaCl (*Wong et al., 2020*), or ATP (overall [Caprin1] = 400 µM) (*Kim et al., 2021*). The trend here is also in line with other theories of polyelectrolytes (*Shen and Wang, 2018*). In contrast, rG-RPA+FH results in *Figure 1d* for pY-Caprin1 shows decreasing LLPS propensity with increasing [NaCl], consistent with experimental data and the expected salt dependence of LLPS of nearly net-neutral polyampholytic IDRs such as Ddx4 (*Lin et al., 2016*).

Interestingly, the decrease in some of the condensed-phase [pY-Caprin1]s with decreasing $T$ (orange and green symbols for $\lesssim$ 20 °C in *Figure 1d* trending toward slightly lower [pY-Caprin1])

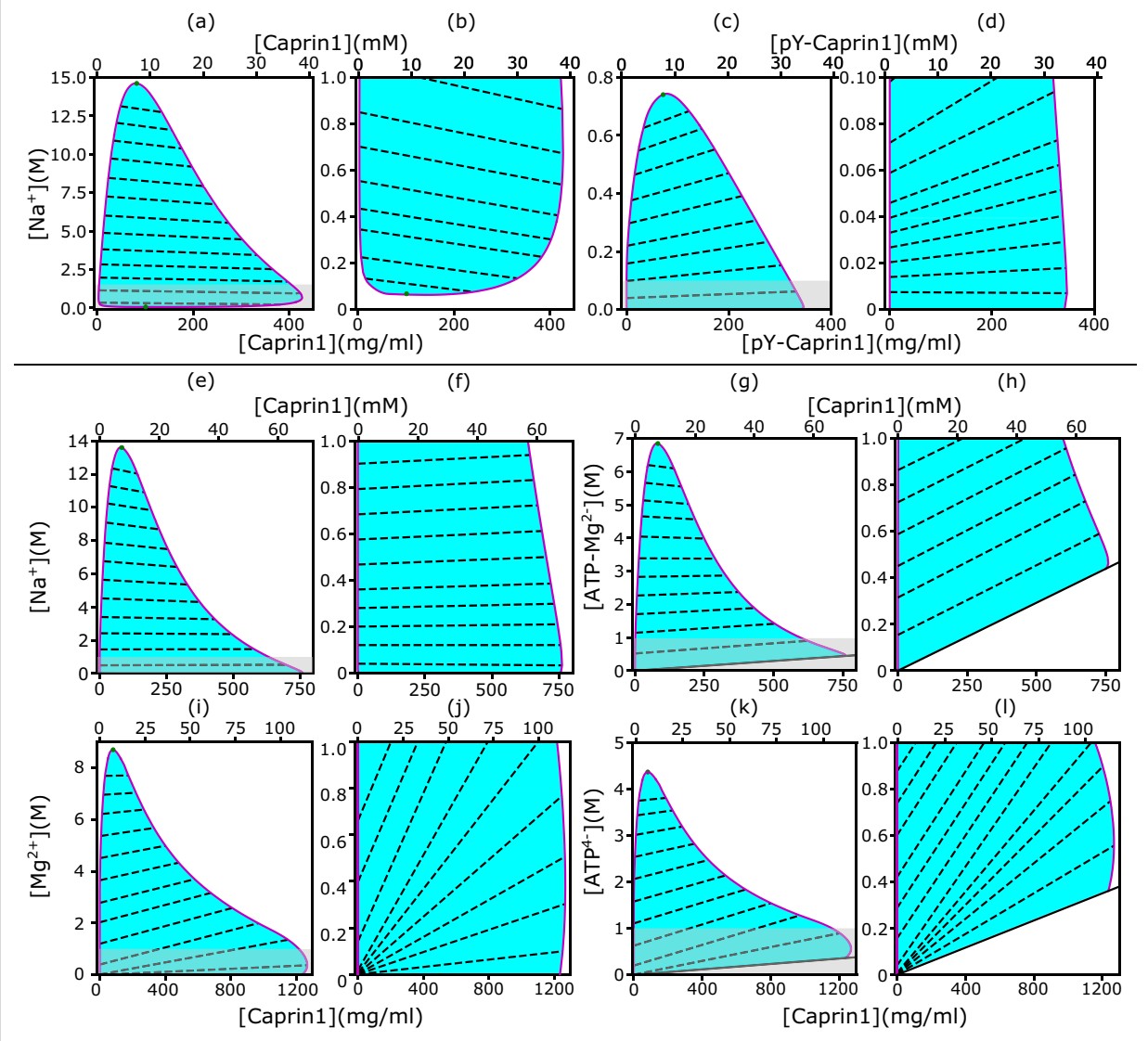

**Figure 2.** rG-RPA+FH theory rationalizes [NaCl]-modulated reentrant phase behavior of Caprin1. In each salt-protein phase diagram ($T$ = 300 K), tielines (dashed) connect coexisting phases on the boundary (magenta curve) of the cyan-shaded coexistence region. For clarity, zoomed-in views of the grey-shaded part in (**a, c, e, g, i, k**) are provided by the plots to the right, i.e., (**b, d, f, h, j, l**), respectively. The solid inclined lines in (**g, h, k, l**) mark the minimum counterion concentrations required for overall electric neutrality. Results are shown for monovalent cation and anion with Caprin1 (**a, b**) or pY-Caprin1 (**c, d**); or monovalent cation and divalent anion with Caprin1 (**e–h**); or divalent cation and tetravalent anion with Caprin1 (**i–l**). Cation-modulated reentrant phase behaviors is seen for a wide concentration range for Caprin1 in (**a, b**) but only a very narrow range of high Caprin1 concentrations in (**e, f, i, j**). The ($\epsilon_h, \epsilon_s$) values for computing the phase diagrams here for Caprin1 and pY-Caprin1, respectively, are the same as those used for **Figure 1c and d**.

The online version of this article includes the following source data for figure 2:

**Source data 1.** Numerical plotting data for the theory-predicted phase diagrams in **Figure 2**.

may suggest a hydrophobicity-driven lower critical solution temperature (LCST)-like reduction of LLPS propensity as temperature approaches ~ 0 °C as in cold denaturation of globular proteins (**Lin et al., 2018**; **Dignon et al., 2019a**), though the hypothetical LCST is below 0 °C and therefore not experimentally accessible. If that is the case, the LLPS region would resemble those with both an UCST and an LCST (**Cinar et al., 2019**). As far as simple modeling is concerned, such a feature may be captured by a FH model wherein interchain contacts are favored by entropy at intermediate to low temperatures and by enthalpy at high temperatures, thus entailing a heat capacity contribution in $\chi(T)$, with $\epsilon_h \rightarrow \epsilon_h(T)$, $\epsilon_s \rightarrow \epsilon_s(T)$ (**Lin et al., 2018**; **Dill et al., 1989**; **Kaya and Chan, 2003**), beyond the

temperature-independent $\epsilon_h$ and $\epsilon_s$ used in *Figures 1c, d , and 2*. Alternatively, a reduction in overall condensed-phase concentration can also be caused by formation of heterogeneous locally organized structures with large voids at low temperatures even when interchain interactions are purely enthalpic (Figure 4 of *Statt et al., 2020*).

## Salt-IDR two-dimensional phase diagrams are instrumental for exploring broader phase properties

*Figure 1c and d*, though informative, are computed by a restricted rG-RPA+FH that assumes a spatially uniform $[Na^+]$. For a more comprehensive physical picture, we now examine possible differences in salt concentration between the IDR-dilute and condensed phases by applying unrestricted rG-RPA+FH to compute two-dimensional salt-Caprin1/pY-Caprin1 phase diagrams (*Figure 2*).

As stated in *Materials and methods* and *Appendix 1*, here we define 'counterions' and 'salt ions', respectively, as the small ions with charges opposite and identical in sign to that of the net charge, $Q$, of a given polymer. For the Caprin1/NaCl system, since Caprin1's net charge is positive, $Na^+$ is salt ion and $Cl^-$ is counterion. Overall electric neutrality of the system implies that the concentrations ($\rho$'s) of polymer ($\rho_p$), counterions ($\rho_c$), and salt ions ($\rho_s$) are related by

$$|Q|\rho_p + z_s\rho_s = z_c\rho_c, \tag{1}$$

where $z_s$ and $z_c$ are, respectively, the valencies of salt ions and counterions. For Caprin1 and pY-Caprin1, $Q = +13$ and $-1$, respectively, and $(z_s, z_c) = (1, 1)$, $(1, 2)$, and $(2, 4)$ are models for different small-ion species in the system. Specifically, in *Figure 2*, we identify the $z_s = 1$ salt ion as $Na^+$ (*Figure 2a–f*) and the $z_c = 1$ counterion as $Cl^-$ (*Figure 2a–d*), the $z_c = 2$ counterion as $(ATP\text{-}Mg)^{2-}$ (*Figure 2g and h*), the $z_s = 2$ salt ion as $Mg^{2+}$ and the $z_c = 4$ counterion as $ATP^{4-}$ (*Figure 2i–l*). As mentioned above, in the present rG-RPA formulation, $(ATP\text{-}Mg)^{2-}$ and $ATP^{4-}$ are modeled minimally as a single-bead ion. They are represented by charged polymer models with more structural complexity in the FTS models below.

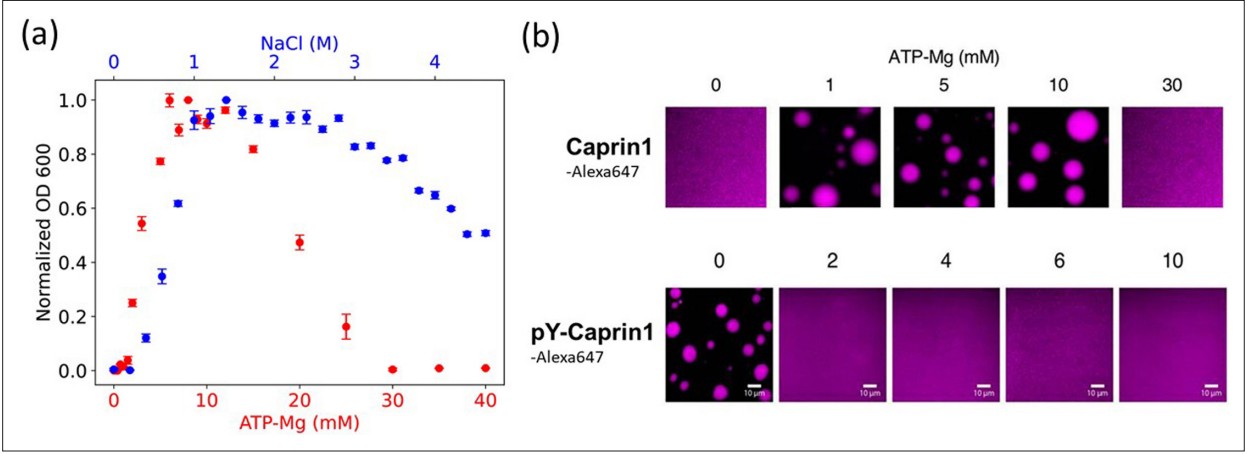

**Figure 3.** Experimental demonstration of [ATP-Mg]- and [NaCl]-modulated reentrant phase behavior for Caprin1. (**a**) Turbidity quantified by optical density at 600 nm (OD600, normalized by peak value) to assess Caprin1 LLPS propensity at [Caprin1]=200 µM [for ATP-Mg dependence (red), bottom scale] or [Caprin1]=300 µM [for NaCl dependence (blue), top scale], measured at room temperature (~23 °C). Error bars are one standard deviations of triplicate measurements, which in most cases was smaller than the plotting symbols. The ATP-Mg dependence seen here for 200 µM Caprin1 is similar to the results for 400 µM Caprin1 (Figure 6C of *Kim et al., 2021*). (**b**) Microscopic images of Caprin1 and pY-Caprin1 at varying [ATP-Mg] at room temperature, showing reentrant behavior for Caprin1 but not for pY-Caprin1. Each sample contains 200 µM of either Caprin1 or pY-Caprin1, with 1% of either Caprin1-Cy5 or pY-Caprin1-Cy5 (labeled with Cyanine 5 fluorescent dye) added for visualization, in a 25 mM HEPES buffer at pH 7.4. Scale bars represent 10 µm.

The online version of this article includes the following source data for figure 3:

**Source data 1.** Numerical values of the experimental data plotted in *Figure 3a*.

**Table 1.** Sodium ions are depleted in the Caprin1-condensed phase relative to the Caprin1-dilute phase.
Consistent with theory, [Na$^+$] is consistently lower in the Caprin1-condensed phase for two temperatures at which the measurements were performed.

| Bulk [Na$^+$] (mM) | $T$ (°C) | Caprin1-Dilute [Na$^+$] (mM) | Caprin1-Condensed [Na$^+$] (mM) |
|---|---|---|---|
| 300 | 25 | 341.3 ± 45.5 | 140.7 ± 6.0 |
| 300 | 35 | 289.5 ± 21.9 | 149.0 ± 2.5 |

uncertainty (±) is standard deviation of triplicate measurements.

## Behavioral trends of rG-RPA-predicted Na$^+$-Caprin1 two-dimensional phase diagrams are consistent with experiment

Notably, *Figure 2a and b* ($z_s = z_c = 1$) predicts that Caprin1 does not phase separate without Na$^+$, consistent with experiment, indicating that monovalent counterions alone (Cl$^-$ in this case) are insufficient for Caprin1 LLPS. When [Na$^+$] is increased, the system starts to phase separate at a small [Na$^+$] ≲ 0.1 M, with LLPS propensity increasing to a maximum at [Na$^+$] ~ 1 M before decreasing at higher [Na$^+$], in agreement with experiment (*Figure 3a*, blue data points) and consistent with Caprin1 LLPS propensity increasing with [NaCl] from 0.1 to 0.5 M (*Figure 1c*). The predicted reentrant dissolution of Caprin1 condensate at high [Na$^+$] in *Figure 2a* is consistent with measurement up to [Na$^+$] ≈ 4.6 M indicating a significant decrease in LLPS propensity when [Na$^+$] ≳ 2.5 M (*Figure 3a*), though the gradual decreasing trend suggests that complete dissolution of condensed droplets is not likely even when NaCl reaches its saturation concentration of ~6 M.

The negative tieline slopes in *Figure 2a and b* predict that Na$^+$ is partially excluded from the Caprin1 condensate. This 'salt partitioning' is most likely caused by Caprin1's net positive charge and is consistent with published research on polyelectrolytes with monovalent salt (*Shen and Wang, 2018*; *Eisenberg and Mohan, 1959*; *Zhang et al., 2016*). Here, the rG-RPA predicted trend is consistent with our experiment showing significantly reduced [Na$^+$] in the Caprin1-condensed phase compared to the Caprin1-dilute phase (*Table 1*), although the larger experimental reduction of [Na$^+$] in the Caprin1 condensed droplet relative to our theoretical prediction remains to be elucidated. In this regard, a similar experimental trend of Na$^+$ tielines was observed recently for the IDP A1-LCD (WT) with a positive (+8) net charge (*Posey et al., 2024*). In contrast, for the near-neutral, very slightly negative model pY-Caprin1 (*Figure 2c and d*), rG-RPA predicts LLPS at [Na$^+$] ≈ 0, and the positive tieline slopes indicate that [Na$^+$] is higher in the condensed than in the dilute phase. Consistent with *Figure 1d*, *Figure 2c* shows that pY-Caprin1 LLPS propensity always decreases with increasing [Na$^+$].

## rG-RPA-predicted salt-IDR two-dimensional phase diagrams underscore effects of counterion valency on LLPS

Interestingly, a different salt dependence of Caprin1 LLPS is predicted when the salt ion remains monovalent but the monovalent counterion Cl$^-$ is replaced by a divalent $z_c = 2$ anion modeling (ATP-Mg)$^{2-}$ (as a single-bead ion) under the simplifying assumption that ATP$^{4-}$ and Mg$^{2+}$ do not dissociate in solution. The corresponding rG-RPA results (*Figure 2e–h*) indicate that, in the presence of divalent counterions (needed for overall electric neutrality of the Caprin1 solution), Caprin1 can undergo LLPS without the monovalent salt (Na$^+$) ions (LLPS regions extend to [Na$^+$] = 0 in *Figure 2e and f*; that is, $\rho_s = 0$, $\rho_c > 0$ in *Equation 1*), because the configurational entropic cost of concentrating counterions in the Caprin1 condensed phase is lesser for divalent ($z_c = 2$) than for monovalent ($z_c = 1$) counterions as only half of the former are needed for approximate electric neutrality in the condensed phase.

Other predicted differences between monovalent (*Figure 2a and b*) and divalent (*Figure 2e and f*) counterions' impact on Caprin1 LLPS include: (i) The maximum condensed-phase [Caprin1] at low [Na$^+$] is lower with monovalent than with divalent counterions ([Caprin1] ~ 40 mM vs. ~ 70 mM). (ii) The [Na$^+$] at the commencement of reentrance (i.e., at the maximum condensed-phase [Caprin1]) is much higher with monovalent than with divalent counterions ([Na$^+$] ~ 1 M vs. ~ 0.1 M). (iii) [Na$^+$] is depleted in the Caprin1 condensate with both monovalent and divalent counterions when overall [Na$^+$] is high (negative tieline slopes for [Na$^+$] ≳ 2 M in *Figure 2a and e*). However, for lower overall

[Na$^+$], [Na$^+$] is slightly higher in the Caprin1 condensate with divalent but not with monovalent counterions (slightly positive tieline slopes for [Na$^+$] ≲ 2 M in *Figure 2e and f*). This prediction suggests that under physiological [Na$^+$] = 150~170 mM, monovalent positive salt ions such as Na$^+$ can be attracted, somewhat counterintuitively, into biomolecular condensates scaffolded by positively charged polyelectrolytic IDRs in the presence of divalent counterions. This phenomenon most likely arises from the attraction of the positively charge monovalent salt ions to the negatively charged divalent counterions in the protein-condensed phase because although the three negatively charged D residues in Caprin1 can attract Na$^+$, it is notable that Na$^+$ is depleted in condensed Caprin1 when the counterion is monovalent (*Figure 2a*).

## rG-RPA is consistent with experimental [ATP-Mg]-dependent Caprin1 reentrant phase behaviors

For the $z_s = 2$, $z_c = 4$ case in *Figure 2i–l* modeling (ATP-Mg)$^{2-}$ complex dissociating completely in solution into Mg$^{2+}$ salt ions and ATP$^{4-}$ counterions (modeled as single-bead ions), rG-RPA predicts Caprin1 LLPS with ATP$^{4-}$ (*Figure 2k and l*) in the absence of Mg$^{2+}$ (the LLPS region includes the horizontal axes in *Figure 2i and j*), likely because the configurational entropy loss of tetravalent counterions in the Caprin1 condensate is less than that of divalent and monovalent counterions. Tetravalent counterions also increase the theoretical maximum condensed-phase [Caprin1] to ≳ 120 mM. At the commencement of reentrance (maximum condensed-phase [Caprin1] in *Figure 2i and j*), [Mg$^{2+}$] ~ 0.4 M, which is intermediate between the corresponding [Na$^+$] ~ 1.0 and 0.1 M, respectively, for monovalent and divalent counterions with $(z_s, z_c) = (1, 2)$ and $(1, 1)$. All tieline slopes for Mg$^{2+}$ and ATP$^{4-}$ in *Figure 2i–l* are significantly positive, except in an extremely high-salt region with [Mg$^{2+}$] > 8 M, indicating that [(ATP-Mg)$^{2-}$] is almost always substantially enhanced in the Caprin1 condensate. These observations from analytical theory will be corroborated by FTS below with the introduction of structurally more realistic models of (ATP-Mg)$^{2-}$, ATP$^{4-}$ together with the possibility of simultaneous inclusion of Na$^+$, Cl$^-$, and Mg$^{2+}$ in the FTS models of Caprin1/pY-Caprin1 LLPS systems. Despite the tendency for polymer field theories to overestimate LLPS propensity and condensed-phase concentrations quantitatively because they do not account for ion condensation (*Shen and Wang, 2017*)—which can be severe for small ions with more than ±1 charge valencies as in the case of condensed [Caprin1] ≳ 120 mM in *Figure 2i–l*, our present rG-RPA-predicted semi-quantitative trends are consistent with experiments indicating [ATP-Mg]-dependent reentrant phase behavior of Caprin1 (*Figure 3a*, red data points, and *Figure 3b*) and that [Mg$^{2+}$] as well as [ATP$^{4-}$] are significantly enhanced in the Caprin1 condensate by a factor of ~5–60 for overall [ATP-Mg] = 3–30 mM (*Table 2*).

## Coarse-grained MD with explicit small ions is useful for investigating subtle salt dependence in biomolecular LLPS

To gain deeper insights, we extend the widely-utilized coarse-grained explicit-chain MD model for biomolecular condensates (*Dignon et al., 2018*; *Das et al., 2020*; *Silmore et al., 2017*) to include explicit small cations and anions (*Materials and methods*). ATP-mediated LLPS of short basic peptides was studied recently using all-atom simulations indicating ATP engaging in electrostatic and cation-π bridging interactions (*Kota et al., 2024*). Here, we limit the small ions in our coarse-grained MD simulations of Caprin1 and pY-Caprin1 LLPS to Na$^+$ and Cl$^-$, focusing on the physical origins of reentrance or lack thereof as well as the effects of ariginine-to-lysine (RtoK) mutations on Caprin1. Coarse-grained

**Table 2.** Colocalization of ATP-Mg in the Caprin1-condensed phase.
For three overall ATP-Mg concentrations at room temperature, the concentrations of ATP$^{4-}$ and Mg$^{2+}$ are all significantly higher in the Caprin1-condensed than in the Caprin1-dilute phase.

| | Caprin1-Dilute | | | Caprin1-Condensed | | |
|---|---|---|---|---|---|---|
| [ATP-Mg] (mM) | [Caprin1] (μM) | [Mg$^{2+}$] (mM) | [ATP$^{4-}$] (mM) | [Caprin1] (mM) | [Mg$^{2+}$] (mM) | [ATP$^{4-}$] (mM) |
| 3 | 67.7±5.0 | 2.85±0.05 | 2.76±0.07 | 29.9±3.8 | 70.7±6.0 | 143±30 |
| 10 | 26.4±1.2 | 8.57±0.14 | 8.53±0.97 | 35.3±3.5 | 137±12 | 197±11 |
| 30 | 117±3 | 28.2±0.3 | 27.6±0.8 | 28.0±2.0 | 134±7 | 174±22 |

models allow for the study of larger systems (IDPs of longer chain lengths and more IDPs in the system), though they cannot provide insights into more subtle structural and energetic effects as in all-atom simulations (*Kota et al., 2024*; *Krainer et al., 2021*; *MacAinsh et al., 2024*). For computational efficiency, here we neglect solvation effects that can arise from the directional hydrogen bonds among water molecules (see, e.g., *Shimizu and Chan, 2001*) by treating other aspects of the aqueous solvent implicitly as in most, although not all (*Lin et al., 2023*; *Wessén et al., 2021*) applications of the methodology (*Dignon et al., 2018*). Several coarse-grained interaction schemes were used in recent MD simulations of biomolecular LLPS (*Dignon et al., 2018*; *Das et al., 2020*; *Joseph et al., 2021*; *Regy et al., 2021*; *Dannenhoffer-Lafage and Best, 2021*; *Tesei et al., 2021*; *Wessén et al., 2022a*; *Wessén et al., 2023*; *Dignon et al., 2019b*). Since we are primarily interested in general principles rather than quantitative details of the phase behaviors of Caprin1 and its RtoK mutants, here we adopt the Kim-Hummer (KH) energies for pairwise amino acid interactions derived from contact statistics of folded protein structures (*Dignon et al., 2018*), which can largely capture the experimental effects of R vs K on LLPS (*Das et al., 2020*).

## Explicit-ion MD rationalizes experimentally observed [NaCl]-dependent Caprin1 reentrant phase behaviors and depletion of Na⁺ in Caprin1 condensate

Consistent with experiment (*Figure 3*) and rG-RPA (*Figure 2a–d*), explicit-ion coarse-grained MD results in *Figure 4* show [NaCl]-dependent reentrant phase behavior for Caprin1 but not for pY-Caprin1 (non-monotonic and monotonic trends indicated, respectively, by the grey arrows in *Figure 4a and b*). In other words, the critical temperature $T_{cr}$, which is defined as the maximum temperature (UCST) of a given phase diagram (binodal, or coexistence curve), increases then decreases with addition of NaCl for Caprin1 but $T_{cr}$ always decreases with increasing [NaCl] for pY-Caprin1. Moreover, consistent with the rG-RPA-predicted tielines in *Figure 2a–d* (negative slopes for Caprin1 and positive slopes for pY-Caprin1), *Figure 4e and g* show that Na⁺ is slightly depleted in the Caprin1 condensed droplet, exhibiting the same trend as that in experiment (*Figure 3a*, blue data points; and *Table 1*) but is enhanced in the pY-Caprin1 droplet (*Figure 4f and h*). Because model temperatures in *Figure 4a and b* and subsequent MD results are given in units of the MD-simulated $T_{cr}$ of WT Caprin1 at [NaCl] = 0 (denoted as $T_{cr}^0$ here), the $T_{cr}$'s of systems with higher or lower LLPS propensities than WT Caprin1 at zero [NaCl] is characterized, respectively, by $T_{cr}/T_{cr}^0 > 1$ or < 1.

*Figure 4e and g* show that [Cl⁻] is enhanced while [Na⁺] is depleted in the Caprin1 droplet. By comparison, *Figure 4f and h* show that both [Cl⁻] and [Na⁺] are enhanced in the pY-Caprin1 droplet with an excess of [Na⁺] to balance the negatively charged pY-Caprin1 (*Figure 4h*). The enhancement of [Cl⁻] in the Caprin1 condensed phase depicted in *Figure 4f and h* is further illustrated in *Figure 5a–d* by comparing the entire simulation box with a condensed droplet in the middle (*Figure 5a*) with individual distributions of the Caprin1 IDR (*Figure 5b*), Na⁺ (*Figure 5c*), and Cl⁻ (*Figure 5d*). A similar trend, also attributed to charge effects, was observed in explicit-water, explicit-ion MD simulations in the presence of a preformed condensate of the N-terminal RGG domain of LAF-1 with a positive net charge (*Zheng et al., 2020*). For Caprin1, *Figure 5e and f* suggests that, as counterion, Cl⁻ can coordinate two positively charged R residues and thereby stabilize indirect counterion-bridged interchain contacts among polycationic Caprin1 molecules to promote LLPS, consistent with an early lattice-model analysis of generic polyelectrolytes (*Orkoulas et al., 2003*) and a recent atomic simulation study of A1-LCD (*MacAinsh et al., 2024*).

## Explicit-ion MD offers insights into counterion-mediated interchain bridging interactions among condensed Caprin1 molecules

To assess the extent to which Cl⁻-mediated bridging interactions (as illustrated in *Figure 5f*) contribute to condensation of polyelectrolytic IDRs, we examine the relative positions of positively charged arginine residues (Arg⁺) and negatively charged counterions (Cl⁻) of a Caprin1 solution under phase-separation conditions in which essentially all Caprin1 molecules are in the condensed phase, using 4000 frames (MD snapshots) of an equilibrated salt-free ([NaCl] = 0) ensemble of 100 WT Caprin1 chains (net charge per chain = +13) with 1300 Cl⁻ counterions at $T < T_{cr}^0$ as an example (*Figure 6*). For simplicity, we focus on Arg⁺–Cl⁻ interactions because the overwhelming majority (15/16) of the positively charged residues in Caprin1 are arginines. The computed radial distribution function, $\rho(r)$, of

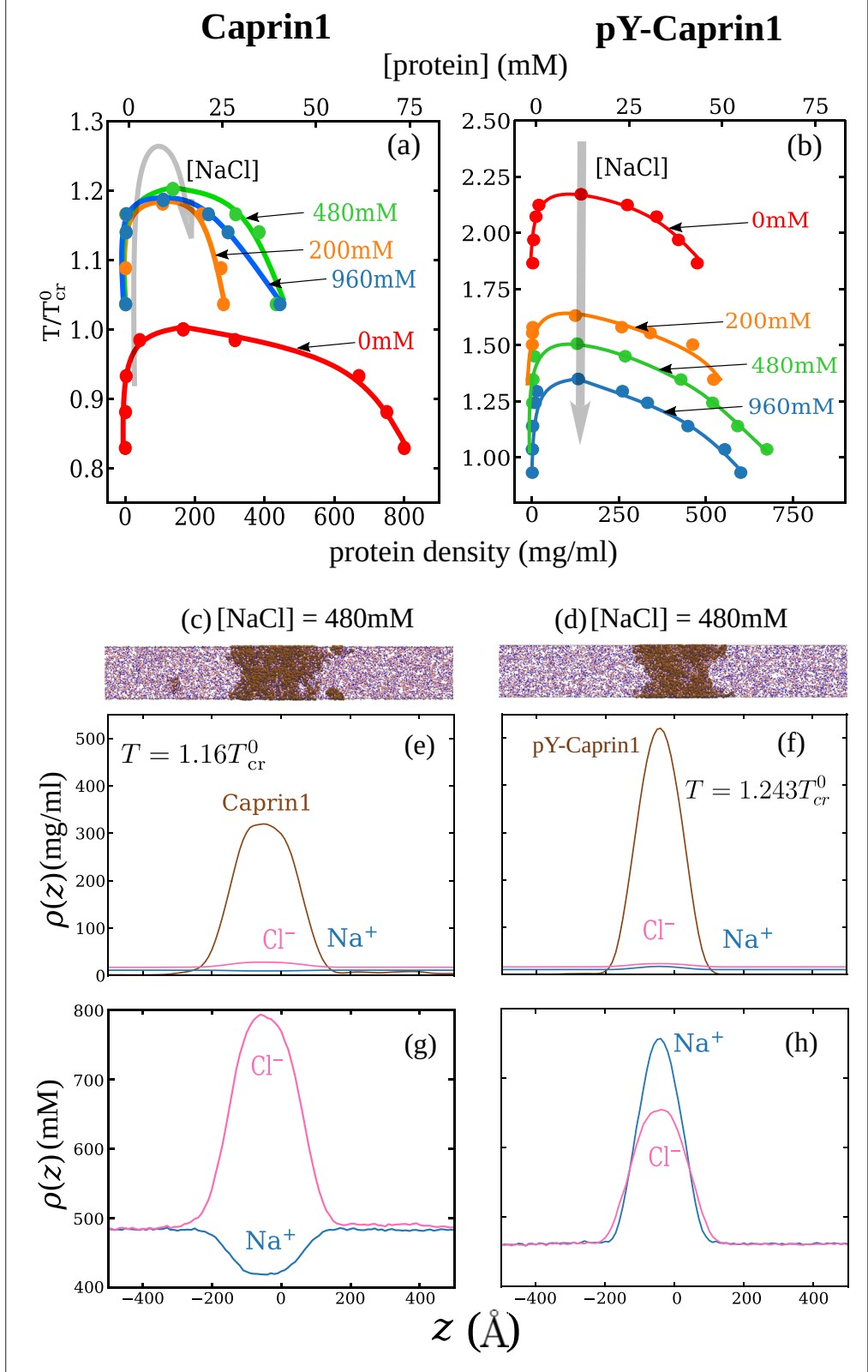

**Figure 4.** Explicit-ion coarse-grained MD rationalizes [NaCl]-modulated reentrant behavior for Caprin1 and lack thereof for pY-Caprin1. (**a**) Simulated phase diagrams (binodal curves) of Caprin1 at different temperatures plotted in units of $T_{cr}^0$ (see text). Symbols are simulated data points. Continuous curves are guides for the eye. Grey arrow indicates variation in [NaCl]. (**b**) Same as (**a**) but for pY-Caprin1. (**c**) A snapshot showing phase equilibrium between

*Figure 4 continued on next page*

*Figure 4 continued*

dilute and condensed phases of Caprin1 (brown chains) immersed in $Na^+$ (blue) and $Cl^-$ (red) ions simulated at [NaCl]=480 mM. (**d**) A similar snapshot for pY-Caprin1. (**e, f**) Mass density profiles, $\rho(z)$ (in units of mg/ml), of $Na^+$, $Cl^-$, and (**e**) Caprin1 or (**f**) pY-Caprin1 along the elongated dimension $z$ of the simulation box showing variations of $Na^+$ and $Cl^-$ concentrations between the protein-dilute phase (low $\rho$ for protein) and protein-condensed phase (high $\rho$ for protein) at the simulation temperatures indicated. (**g, h**) Corresponding zoomed-in concentration profiles $\rho(z)$ in units of mM for $Na^+$ and $Cl^-$. Additional mass density profiles for [NaCl]=200 mM and 400 mM are provided in *Appendix 1—figure 3*.

The online version of this article includes the following source data for figure 4:

**Source data 1.** Numerical plotting data for the coarse-grained molecular dynamics-simulated curves in *Figure 4*.

$Cl^-$ around a given $Arg^+$ exhibits a sharp peak at small $r$ that drops to a minium at $r \approx 11$ Å (*Figure 6a*), indicating a strong spatial association between the oppositely charged $Arg^+$ and $Cl^-$ as expected. Indeed, within the ensemble we analyze, 5,121,148/(4000×1300) = 98.5% of the $Cl^-$ ions are within 11 Å of an $Arg^+$. We next enumerate putative bridging interactions involving two $Arg^+$s on different Caprin1 chains and one $Cl^-$ (*Figure 6b*) by identifying three-bead configurations in which the distance of $Cl^-$ to each of the two $Arg^+$ is $\leq 11$ Å (within the dominant small-$r$ peak of $\rho(r)$ in *Figure 6a*), which implies that the distance between the two $Arg^+$s is $\leq 22$ Å. In our ensemble, 4,519,387/(4000×1300) = 86.9% of the $Cl^-$ counterions are identified to be in one or more of a total of 25,112,331 such putative bridging interaction configurations. This means that, on average, each $Cl^-$ is involved in 25,112,331/4,519,387 = 5.56 configurations, and thus are coordinating $\approx 4$ $Arg^+$s because there are 6 ($\approx 5.56$) ways of pairing 4 $Arg^+$s. *Figure 6c* shows the distribution of putative bridging configurations with respect to $Arg^+$–$Arg^+$ distance $R$. Spatial distributions of $Cl^-$ in these configurations are provided in *Figure 6d and e*, which are quite similiar to those of isolated $Arg^+$–$Cl^-$–$Arg^+$ systems for $R \lesssim 14$ Å (*Figure 6f and g*). Among the putative bridging configurations, we make an energetic distinction between true bridging and neutralizing (screening) configurations. Physically, a true bridging

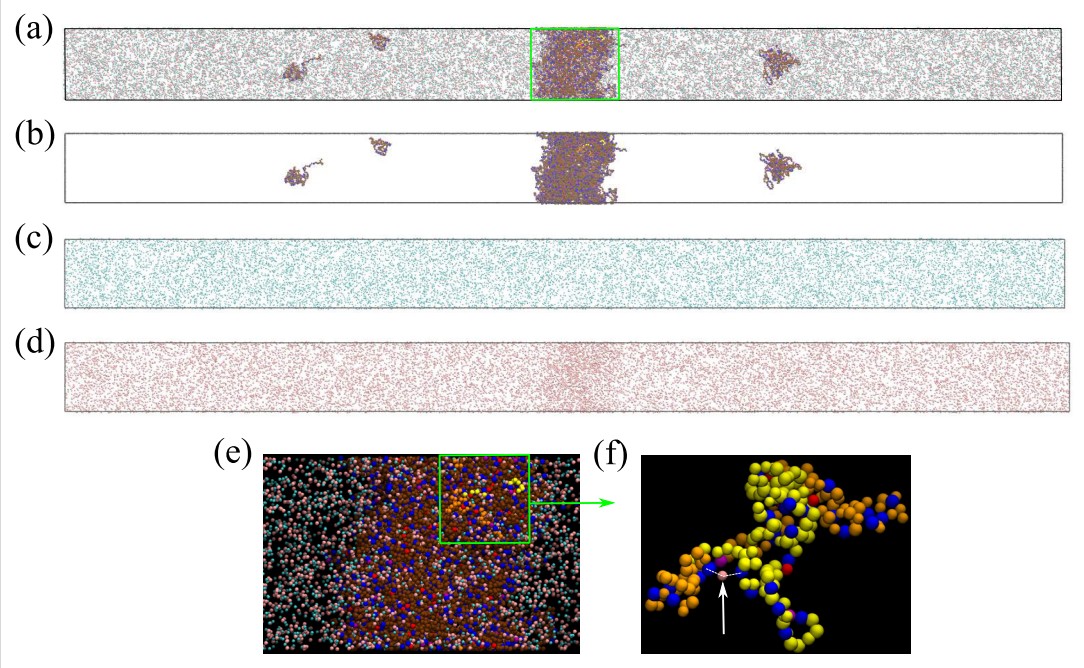

**Figure 5.** Counterions can stabilize Caprin1 condensed phase by favorable bridging interactions. (**a**) Snapshot from explicit-ion coarse-grained MD under LLPS conditions for Caprin1, showing the spatial distributions of Caprin1, $Na^+$, and $Cl^-$ (as in *Figure 4c*). The three components of the same snapshot are also shown separately in (**b**) Caprin1, (**c**) $Na^+$, and (**d**) $Cl^-$. (**e**) A zoomed-in view of the condensed droplet corresponding to the green box in (**a**), now with a black background and a different color scheme. (**f**) A further zoomed-in view of the part enclosed by the green box in (**e**) focusing on two interacting Caprin1 chains. A $Cl^-$ ion (pink bead indicated by the arrow) is seen interacting favorably with two arginine residues (blue beads) on the two Caprin1 chains (whose uncharged residues are colored differently by yellow or orange, lysine and aspartic acids in both chains are depicted, respectively, in magenta and red).

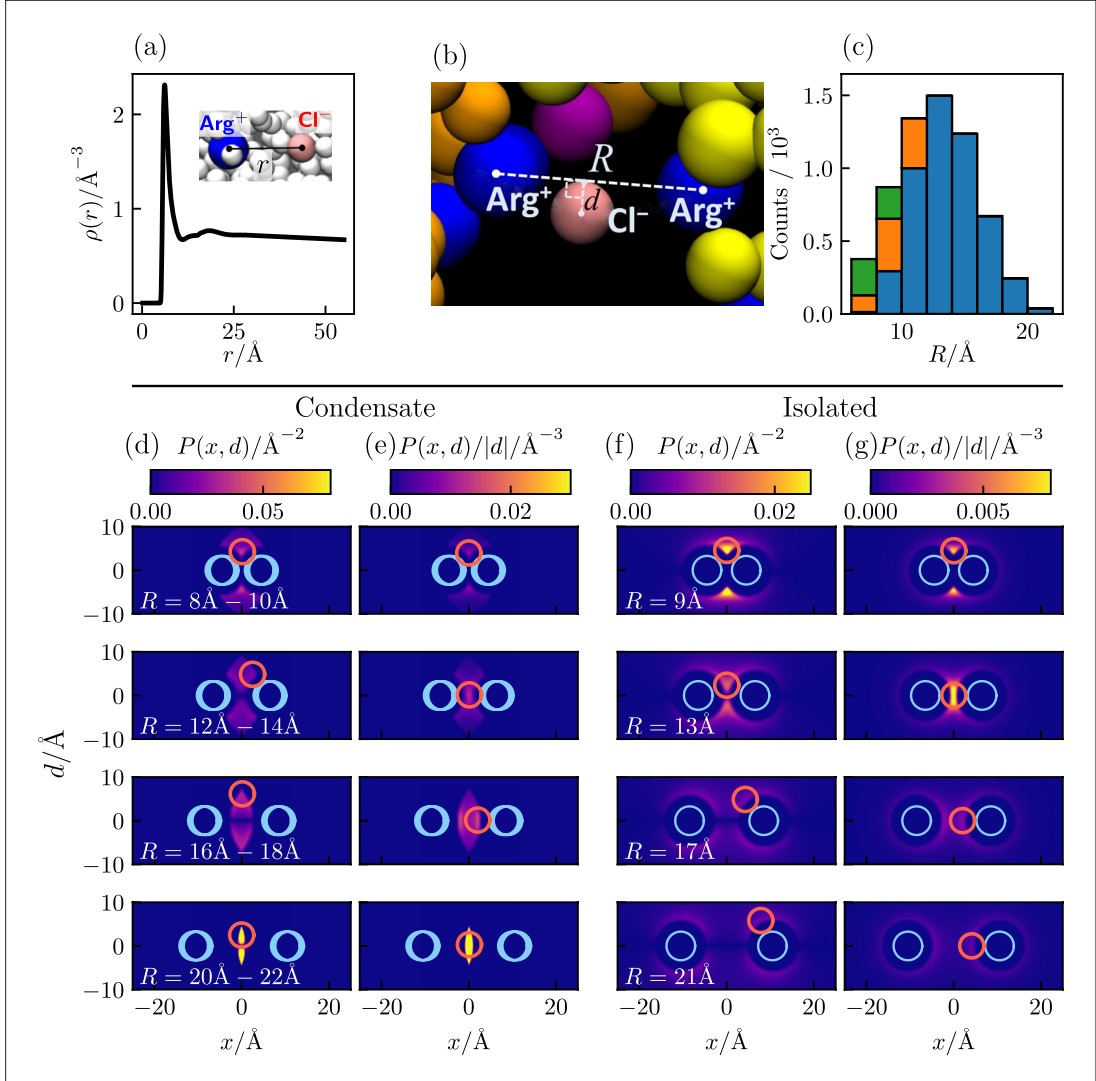

**Figure 6.** Counterion interactions in polyelectrolytic Caprin1. Shown distributions are averaged from 4000 equilibrated coarse-grained MD snapshots of 100 Caprin1 chains and 1300 $Cl^-$ counterions under phase-separation conditions ($T/T_{cr}^0 = 160/193 = 0.829$) in a 115×115×1610 Å³ simulation box in which essentially all Caprin1 chains are in a condensed droplet. (**a**) Radial distribution function of $Cl^-$ around a positively charged arginine residue ($Arg^+$). (**b**) A zoomed-in view of **Figure 5f** showcasing a putative bridging configuration with a $Cl^-$ interacting favorably with a pair of $Arg^+$s on two different Caprin1 chains. Configurational geometry is characterized by $Arg^+$--$Arg^+$ distance $R$ and the distance $d$ of the $Cl^-$ from the line connecting the two $Arg^+$s. (**c**) Distribution of putative bridging interaction configurations with respect to $R$. Numbers of true bridging, neutralizing, and intermediate configurations are, respectively, in blue, green and orange. (**d, e**) Heat maps of two-dimensional projections of spatial distributions of $Cl^-$ around two $Arg^+$s satisfying the putative bridging interaction conditions among the MD snapshots. (**f, g**) Corresponding projected distributions of isolated $Arg^+$--$Cl^-$--$Arg^+$ Boltzmann-averaged systems at model temperature $T$. Here, $P(x, d)$ is the total density of $Cl^-$ on a circle of radius $|d|$ perpendicular to the heat map at horizontal position $x$ (**d, f**); thus the average $Cl^-$ density at a given point $(x, d)$ is $P(x, d)/2\pi|d|$, the patterns of which are exhibited by $P(x, d)/|d|$ heat maps in (**e, g**). $P(x, d)$ is symmetric with respect to $d \leftrightarrow -d$ by construction, i.e., $P(x, d) = P(x, -d)$. In each heat map, the size and (ranges of) positions of model $Arg^+$s are indicated by blue circles; the size and the position or one of two positions (at $\pm d$) of maximum $Cl^-$ density is indicated by a magenta circle. The MD-simulated distributions of the condensed system (**d, e**) are quite similar to the theory-computed isolated system (**f, g**) for $R \lesssim 14$ Å, indicating that individual bridging interactions in the crowded Caprin1 condensates may be understood approximately by the electrostatics of an isolated, three-bead $Arg^+$--$Cl^-$--$Arg^+$ system. For larger $R$, the heat maps in (**f, g**) and (**d, e**) are not as similar because some of the configurations in the isolated system (**f, g**) are precluded by the requirement that $Arg^+$--$Cl^-$ distance < 11 Å for putative bridging interactions in (**d, e**).

The online version of this article includes the following source data for figure 6:

**Source data 1.** Numerical values of the coarse-grained molecular dynamics-simulated data plotted in **Figure 6a and c**.

configuration may be defined by an overall favorable (< 0) sum of (i) unfavorable Coulomb potential between two $Arg^+$ and (ii) the favorable Coulomb potential between the $Cl^-$ and one of the $Arg^+$s that is farther away from the $Cl^-$. By the same token, a neutralizing (screening) configuration may be defined by a corresponding overall unfavorable or neutral (≥ 0) sum of these two Coulomb potentials (i.e., the farther $Arg^+$–$Cl^-$ distance is larger than the $Arg^+$–$Arg^+$ distance). In this regard, and in more general terms, $Cl^-$ ions in bridging and neutralizing interactions may be considered, respectively, as a 'strong-attraction promoter' and a 'weak-attraction suppressor' of LLPS (*Nguemaha and Zhou, 2018*; *Ghosh et al., 2019*).

In the present analysis, we group putative bridging configurations by $R$ in bins of 2 Å (*Figure 6c*). Accordingly, we may classify $Cl^-$ positions satisfying the above condition of favorable (< 0) sum of Coulomb potentials for all $R$ values within the 2 Å range of the bin as in true bridging configurations (79.6%), those $Cl^-$ positions satisfying the above condition of unfavorable (≥ 0) sum of Coulomb potentials for all $R$ values in the 2 Å range as in neutralizing configurations (7.4%), and those that satisfy neither as 'intermediate' configurations (13.0%). Even with this more stringent criterion, ≈80% of putative bridging configurations are true bridging configurations. Because on average a $Cl^-$ counterion known to be involved in at least one putative bridging configuration is on average participating in ~ 5–6 such configurations, the probability that it is involved in at least one true bridging configuration is very high, at ≈ 1.0 − $(0.2)^5$ = 99.97%. Thus, even without taking into consideration bridging interactions involving lysines, we may reasonably conclude that an overwhelming majority (≈ 87%) of $Cl^-$ counterions in the coarse-grained MD system considered are engaged in condensation-driving true bridging interactions coordinating pairs of $Arg^+$ on different Caprin1 chains. Similar extensive $Cl^-$ and $Na^+$ bridging interactions are observed in a recent all-atom molecular dynamics study of LLPS of short peptides under a variety of overall salt concentrations (*MacAinsh et al., 2024*).

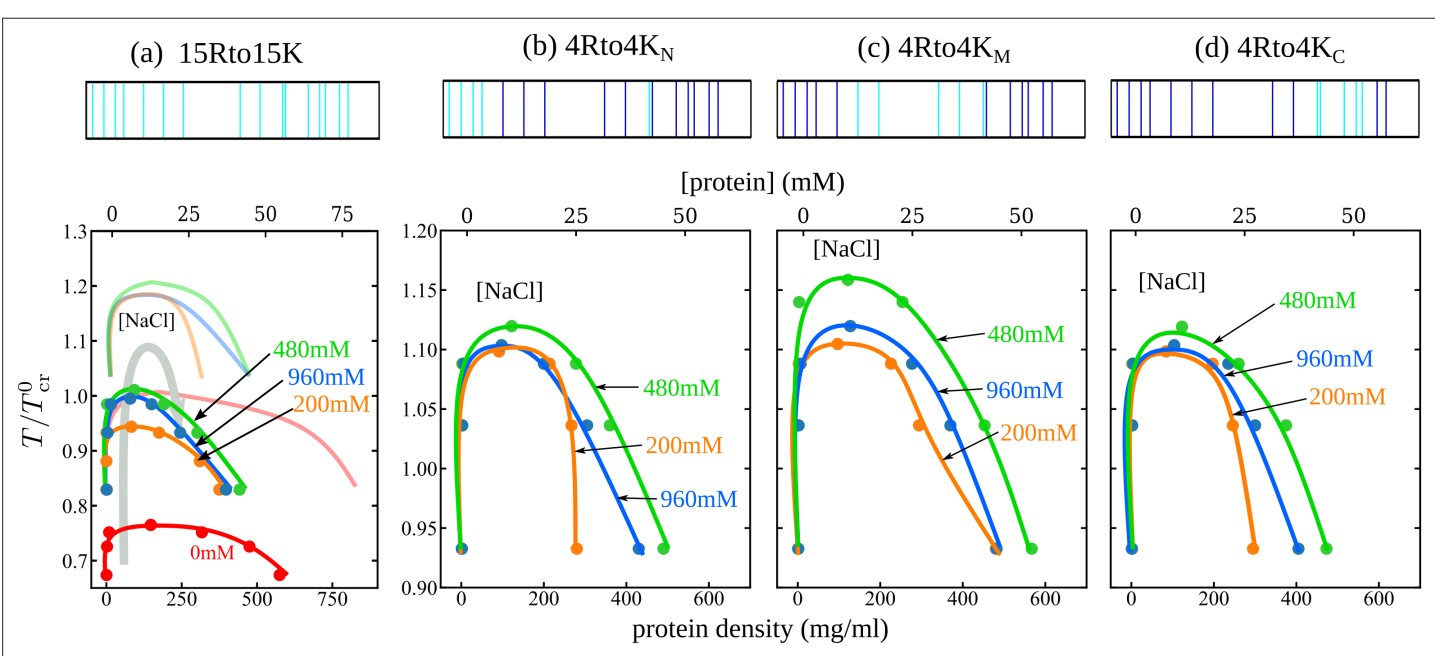

**Figure 7.** Explicit-ion coarse-grained MD rationalizes [NaCl]-modulated phase behavior for RtoK variants of Caprin1. Four variants studied experimentally (*Wong et al., 2020*) are simulated: (**a**) 15Rto15K, in which 15 R's in the WT Caprin1 IDR are substituted by K, (**b**) 4Rto4K_N, (**c**) 4Rto4K_M, and (**d**) 4Rto4K_C, in which 4 R's are substituted by K in the (**b**) N-terminal, (**c**) middle, and (**d**) C-terminal regions, respectively. Top panels show positions of the R (dark blue) and K (cyan) along the Caprin1 IDR sequence. Lower panels are phase diagrams in the same style as *Figure 4*. The phase diagrams for WT Caprin1 from *Figure 4a* are included as continuous curves with no data points in (**a**) for comparison.

The online version of this article includes the following source data for figure 7:

**Source data 1.** Numerical values of the coarse-grained molecular dynamics-simulated data points plotted as filled circles in the phase diagrams in *Figure 7*.

## Explicit-ion MD rationalizes [NaCl]-dependent phase properties of arginine-lysine mutants of Caprin1

We apply our MD methodology also to four RtoK Caprin1 variants, termed 15Rto15K, 4Rto4K_N, 4Rto4K_M, and 4Rto4K_C (*Appendix 1—figure 1*), which involve 15 or 4 RtoK substitutions (*Wong et al., 2020*). The simulated phase diagrams in *Figure 7* exhibit reentrant phase behaviors for all three 4Rto4K variants. While these results are consistent with experiments showing LLPS of these 4Rto4K variants commencing at different nonzero [NaCl]s (*Wong et al., 2020*), the simulated reentrant dissolution is not observed experimentally, probably because the actual [NaCl] needed is beyond the experimentally investigated or physically possible range of salt concentration. Simulated reentrant phase behaviors are also seen for 15Rto15K; but as will be explained below, its much lower simulated UCST is consistent with no experimental LLPS for this variant (*Wong et al., 2020*). Since our main focus here is on general physical principles, we do not attempt to fine-tune the MD parameters for a quantitative match between simulation and experiment. Experimentally, only WT exhibits a clear trend toward reentrant dissolution of condensed droplets (with a LLPS propensity plateau at [NaCl] ≈ 1.55−2.5 M, *Figure 3a*, blue data points), whereas the LLPS of 4Rto4K_M and 4Rto4K_C commences at [NaCl] ≈ 1.3 M, LLPS propensity then increases with [NaCl] (a trend consistent with the MD-predicted increasing LLPS propensity at low [NaCl]s in *Figure 7b and c*), but no sign of reentrant dissolution is seen up to the maximum [NaCl] = 2 M investigated experimentally for the RtoK variants (Figure 9B of *Wong et al., 2020*). In contrast, the MD phase diagrams in *Figure 7* show a maximum LLPS propensity (highest $T_{cr}$) at [NaCl] ≈ 0.5 M. This qualitative agreement with quantitative mismatch suggests that real Caprin1 LLPS is somewhat less sensitive to small monovalent ions than that stipulated by the present MD model. This question should be tackled in future studies by considering, for example, alternate pairwise amino acid interaction energies (*Das et al., 2020*; *Dignon et al., 2018*; *Joseph et al., 2021*; *Regy et al., 2021*; *Dannenhoffer-Lafage and Best, 2021*; *Tesei et al., 2021*; *Wessén et al., 2022a*; *Wessén et al., 2023*) and their temperature dependence (*Cinar et al., 2019*; *Dignon et al., 2019a*).

Limitations notwithstanding, the MD-simulated trend agrees largely with experiment. Predicted LLPS propensities quantified by the $T_{cr}$s in *Figure 7* follow the rank order of WT > 4Rto4K_M > 4Rto4K_N ≈ 4Rto4K_C > 15Rto15K, which is essentially identical to that measured experimentally, viz., WT > 4Rto4K_M > 4Rto4K_C > 4Rto4K_N > 15Rto15K (Figure 9B of *Wong et al., 2020*). In comparing theoretical and experimental LLPS, a low theoretical $T_{cr}$ can practically mean no experimental LLPS when the theoretical $T_{cr}$ is below the freezing temperature of the real system (*Lin et al., 2016*; *Brady et al., 2017*). *Figure 7a* shows that even the highest $T_{cr}$ for 15Rto15K (at model [NaCl] = 480 mM) is essentially at the same level as $T_{cr}^0$ for WT at [NaCl] = 0 ($T_{cr}/T_{cr}^0 \approx 1$). This MD prediction is consistent with the combined experimental observations of no LLPS for 15Rto15K up to at least [NaCl] = 2 M and no LLPS for WT Caprin1 at [NaCl] = 0 (Figure 9B and C of *Wong et al., 2020*).

## Field-theoretic simulation (FTS) is an efficient tool for studying multiple-component phase properties

We next turn to modeling of Caprin1 or pY-Caprin1 LLPS modulated by both ATP-Mg and NaCl. Because tackling such many-component LLPS systems using rG-RPA or explicit-ion MD is numerically challenging, here we adopt the complementary FTS approach (*Fredrickson, 2006*) outlined in *Materials and methods* for this aspect of our investigation. FTS is based on complex Langevin dynamics (*Parisi, 1983*; *Klauder, 1983*), which is related to an earlier formulation for stochastic quantization (*Parisi and Wu, 1981*; *Chan and Halpern, 1986*) and has been applied extensively to polymer solutions (*Fredrickson et al., 2002*; *Fredrickson, 2006*). Recently, FTS has provided insights into charge-sequence-dependent LLPS of IDRs (*McCarty et al., 2019*; *Pal et al., 2021*; *Wessén et al., 2022a*; *Lin et al., 2019b*; *Lin et al., 2023*). The starting point of FTS is identical to that of rG-RPA. FTS invokes no RPA and is thus advantageous over rG-RPA in this regard, although it is still limited by the lattice size used for simulation and its restricted treatment of excluded volume (*Pal et al., 2021*). Here we apply the protocol detailed in *Pal et al., 2021*; *Lin et al., 2023*.

## A simple model of ATP-Mg for FTS

Going beyond the single-bead model for (ATP-Mg)$^{2-}$ in our analytical rG-RPA theory (*Figure 2*), we now adopt a 6-bead polymeric representation of (ATP-Mg)$^{2-}$ (*Figure 8a*) in which four negative and

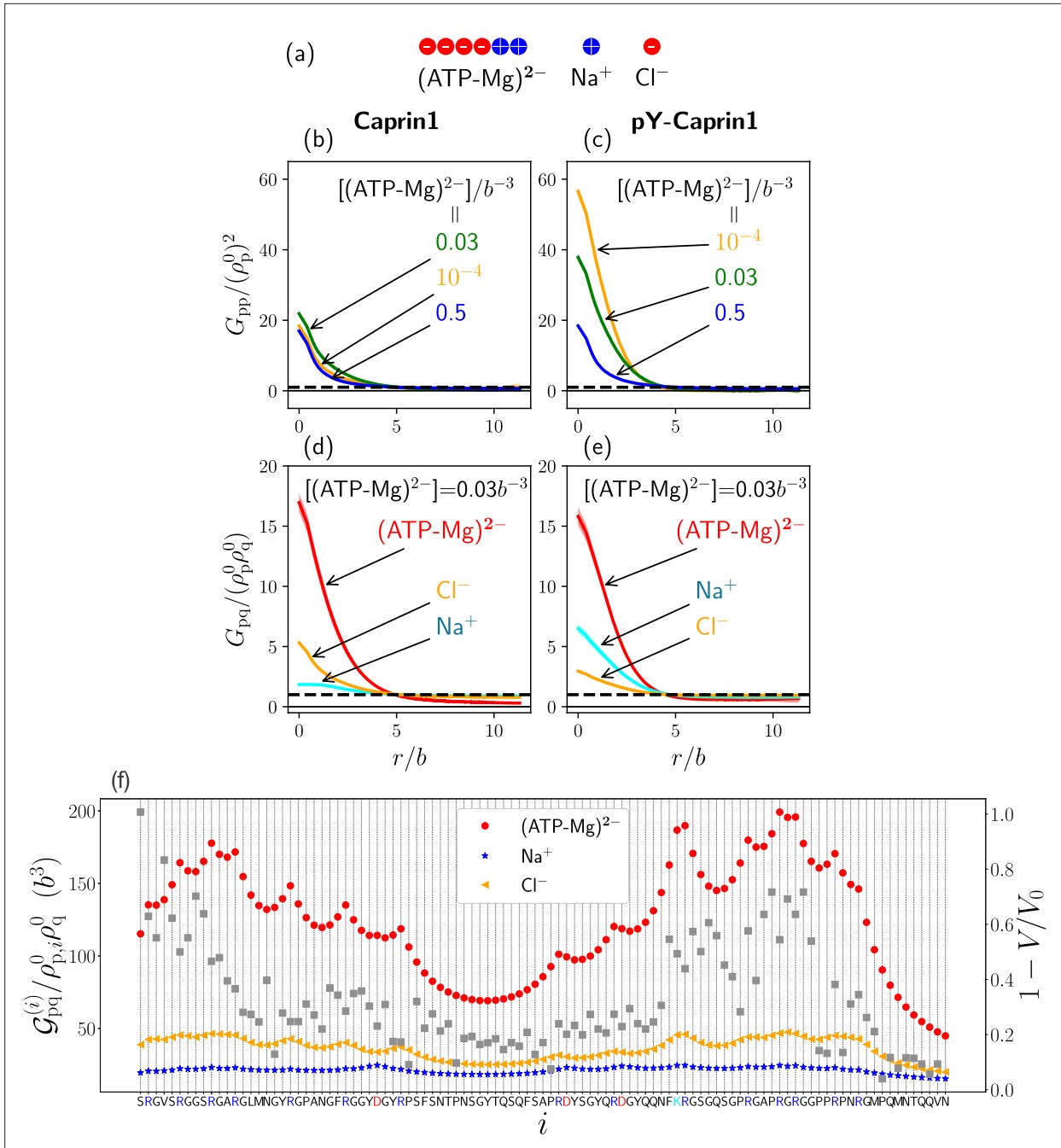

**Figure 8.** FTS rationalizes experimental observation of Caprin1-ATP interactions. (**a**) The 6-bead model for $(ATP-Mg)^{2-}$ and the single-bead models for monovalent salt ions used in the present FTS. (**b–e**) Normalized protein-protein correlation functions at three $[(ATP-Mg)^{2-}]$ values (**b, c**) and protein-ion correlation functions (**Equation 7**) at $[(ATP-Mg)^{2-}]/b^{-3} = 0.03$ (**d, e**) for Caprin1 (**b, d**) and pY-Caprin1 (**c, e**), computed for Bjerrum length $l_B = 7b$. Horizontal dashed lines are unity baselines (see text). (**f**) Values of position-specific integrated correlation $\mathcal{G}_{pq}^{(i)}/\rho_{p,i}^0\rho_q^0$ (left vertical axis) correspond to the relative contact frequencies between individual residues labeled by $i$ along the Caprin1 IDR sequence with q = $(ATP-Mg)^{2-}$, $Na^+$, or $Cl^-$ under the same conditions as (**d**) (**Equation 9**) (color symbols). Included for comparison are experimental NMR volume ratios $V/V_0$ data on site-specific Caprin1-ATP association (**Kim et al., 2021**). $V/V_0$ decreases with increased contact probability, although a precise relationship is yet to be determined. Thus, the plotted $1 - V/V_0$ (grey data points, right vertical scale) is expected to correlate with contact frequency.

The online version of this article includes the following source data for figure 8:

**Source data 1.** Numerical values of all theoretical and experimental data points plotted in **Figure 8**.

two positive charges serve to model ATP$^{4-}$ and Mg$^{2+}$ respectively. Modeling (ATP-Mg)$^{2-}$ as a short charged polymer enables application of existing FTS formulations for multiple charge sequences to systems with IDRs and (ATP-Mg)$^{2-}$. While the model in **Figure 8a** does not capture structural details, its charge distribution does correspond roughly to that of the chemical structure of (ATP-Mg)$^{2-}$. In developing FTS models involving IDR, (ATP-Mg)$^{2-}$, and NaCl, we first assume for simplicity that (ATP-Mg)$^{2-}$ does not dissociate and consider systems consisting of any given overall concentrations of IDR and (ATP-Mg)$^{2-}$ wherein all positive and negative charges on the IDR and (ATP-Mg)$^{2-}$ are balanced, respectively, by Cl$^-$ and Na$^+$ to maintain overall electric neutrality (**Figure 8a**).

## Phase behaviors can be probed by FTS density correlation functions

LLPS of FTS systems can be monitored by correlation functions (**Pal et al., 2021**). Here, we compute intra-species IDR self-correlation functions $G_{pp}(r)$ (**Figure 8b and c**) and inter-species cross-correlation functions $G_{pq}(r)$ between the IDR and (ATP-Mg)$^{2-}$ or NaCl (**Figure 8d and e**) at three different overall [(ATP-Mg)$^{2-}$] $= 10^{-4}b^{-3}$, $0.03b^{-3}$, and $0.5b^{-3}$, where $b$ may be taken as the peptide virtual bond length $\approx 3.8$ Å (**Materials and methods**). The correlation functions in **Figure 8b–e** are normalized by overall densities $\rho_p^0$ of the IDR and $\rho_q^0$ for (ATP-Mg)$^{2-}$, Na$^+$ or Cl$^-$, wherein density is the bead density for the given molecular species in units of $b^{-3}$. LLPS of the IDR is signaled by $G_{pp}(r)/(\rho_p^0)^2$ in **Figure 8b and c** dropping below the unity baseline (dashed) at large distance $r$ because it implies a spatial region with depleted IDR below the overall concentration, which is possible only if the IDR is above the overall concentration in at least another spatial region. In other words, $G_{pp}(r)/(\rho_p^0)^2 < 1$ for large $r$ indicates that IDR concentration is heterogeneous and thus the system is phase separated. For small $r$, $G_{pp}(r)/(\rho_p^0)^2$ is generally expected to increase because IDR chain connectivity facilitates correlation among residues local along the chain. On top of this, LLPS propensity may be quantified by $G_{pp}(r)/(\rho_p^0)^2$ for small $r$ because a higher value indicates a higher tendency for different chains to associate and thus a higher LLPS propensity (**Pal et al., 2021**).

## FTS rationalizes [ATP-Mg]-modulated Caprin1 reentrant phase behaviors and their colocalization in the condensed phase

[(ATP-Mg)$^{2-}$]-modulated reentrance is predicted by FTS for Caprin1 but not for pY-Caprin1: When [(ATP-Mg)$^{2-}$]$/b^{-3}$ varies from $10^{-4}$ to 0.03 to 0.5, small-$r$ values of the Caprin1 $G_{pp}(r)$ in **Figure 8b** initially increase then decrease, whereas the corresponding small-$r$ values of the pY-Caprin1 $G_{pp}(r)$ in **Figure 8c** decrease monotonically, consistent with rG-RPA (**Figure 2g, h, k, l**) and experiment (**Figure 3**). The inter-species cross-correlations in **Figure 8d, e** show further that when an IDR condensed phase is present at [(ATP-Mg)$^{2-}$] $= 0.03b^{-3}$ (as indicated by large-$r$ behaviors of $G_{pp}(r)/(\rho_p^0)^2$ in **Figure 8b, c**), (ATP-Mg)$^{2-}$ is colocalized with Caprin1 or pY-Caprin1 (high value of $G_{pq}/\rho_p^0\rho_q^0$ for small $r$) in the IDR-condensed droplet. By comparison, the variation of [Na$^+$] and [Cl$^-$] is much weaker. For Caprin1, Cl$^-$ is enhanced over Na$^+$ in the Caprin1 condensed phase (small-$r$ $G_{pq}/\rho_p^0\rho_q^0$ of the former larger than the latter in **Figure 8d**), but the reverse is seen for pY-Caprin1 (**Figure 8e**). This FTS-predicted difference, most likely arising from the positive net charge on Caprin1 and the smaller negative net charge on pY-Caprin1, is consistent with the MD results in **Figure 4e-h** and **Appendix 1—figure 3**.

## FTS rationalizes experimentally observed residue-specific binding of Caprin1 with ATP-Mg

The propensities for (ATP-Mg)$^{2-}$, Na$^+$, and Cl$^-$ to associate with each residue $i$ along the Caprin1 IDR ($i = 1, 2, \ldots, 103$) in FTS are quantified by the residue-specific integrated correlation $\mathcal{G}_{pq}^{(i)}/\rho_{p,i}^0\rho_q^0$ in **Figure 8f**, which is the integral of the corresponding $G_{pq}^{(i)}(r)$ from $r = 0$ to a relative short cutoff distance $r = r_{contact}$ to provide a relative contact frequency for residue $i$ and ionic species q to be in spatial proximity (**Materials and methods** and **Appendix 1**). Notably, the residue-position-dependent integrated correlation for (ATP-Mg)$^{2-}$ varies significantly, exhibiting much larger values near the N-terminal and a little before the C-terminal but weaker correlation elsewhere (**Figure 8f**, red symbols). The two regions of high integrated correlation (i.e., favorable association) coincide with regions with high sequence concentration of positively charged residues. This FTS prediction is remarkably similar to the experimental NMR finding that binding between (ATP-Mg)$^{2-}$ and Caprin1 occurs strongly at the arginine-rich N- and C-terminal regions, as indicated by the volume ratio $V/V_0$ data in Figure 1C of **Kim et al., 2021** that quantifies the ratio of peaks in NMR spectra in the presence and absence of trace

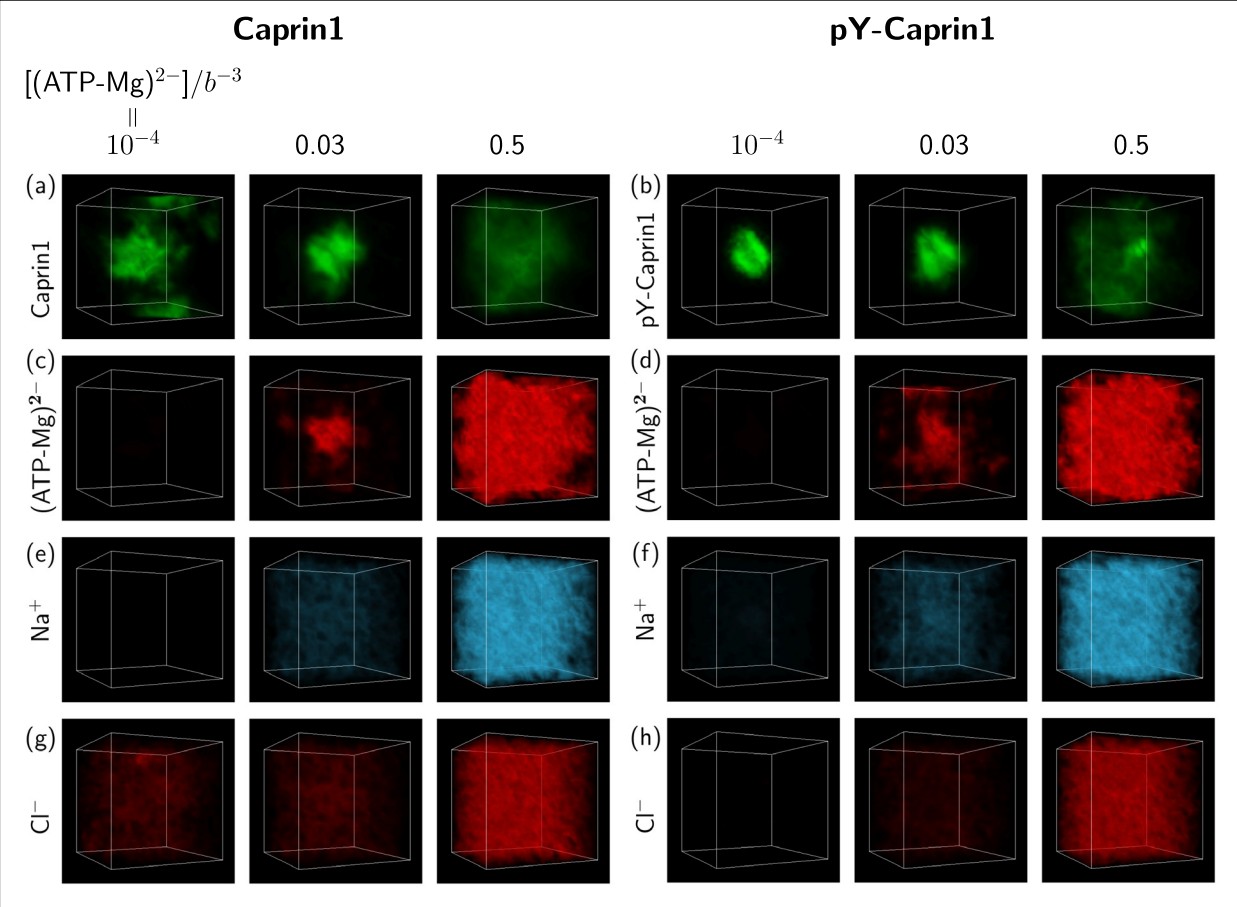

**Figure 9.** FTS rationalizes colocalization of ATP-Mg with the Caprin1 condensate. FTS snapshots are from simulations at $l_B = 7b$ (same as that for *Figure 8*). Spatial distributions of real positive parts of the density fields for the protein (**a, b**), (ATP-Mg)$^{2-}$ (**c, d**), Na$^+$ (**e, f**), and Cl$^-$ (**g, h**) components are shown by three snapshots each for Caprin1 (left panels) and pY-Caprin1 (right panels) at different [(ATP-Mg)$^{2-}$] values as indicated. Colocalization of (ATP-Mg)$^{2-}$ with the Caprin1 condensed droplet is clearly seen in the [(ATP-Mg)$^{2-}$]/$b^{-3}$ = 0.03 panel of (**c**).

amounts of ATP-Mn. For comparison with the FTS results, this set of experimental data is replotted as $1 - V/V_0$ in *Figure 8f* (grey symbols, right vertical axis) to illustrate the similarity in experimental and theoretical trends because $1 - V/V_0$ is expected to trend with contact frequency. Corresponding FTS results for Na$^+$ and Cl$^-$ in *Figure 8f* exhibit much less residue-position-dependent variation, with Cl$^-$ displaying only slightly enhanced association in the same arginine-rich regions, and Na$^+$ showing even less variation, presumably because the positive charges on Caprin1 are already essentially neuralized by the locally associated (ATP-Mg)$^{2-}$ or Cl$^-$ ions. The theory-experiment agreement in *Figure 8f* regarding ATP-Caprin1 interactions indicates once again that electrostatics is an important driving force underlying many aspects of experimentally observed Caprin1–(ATP-Mg)$^{2-}$ association.

## FTS snapshots of [ATP-Mg]-modulated reentrant phase behaviors and Caprin1-ATP-Mg colocalization

The above FTS-predicted trends are further illustrated in *Figure 9* by field snapshots. Such FTS snapshots are generally useful for visualization and heuristic understanding (*McCarty et al., 2019*; *Pal et al., 2021*; *Wessén et al., 2022a*), including insights into subtler aspects of spatial arrangements exemplified by recent studies of subcompartmentalization entailing either co-mixing or demixing in multiple-component LLPS that are verifiable by explicit-chain MD (*Pal et al., 2021*; *Wessén et al., 2022a*). Now, trends deduced from the correlation functions in *Figure 8* are buttressed by the representative snapshots in *Figure 9*: As the bead density of (ATP-Mg)$^{2-}$ is increased from $10^{-4}b^{-3}$ to $0.03b^{-3}$ to $0.5b^{-3}$, the spatial distribution of Caprin1 evolves from an initially dispersed state to a concentrated droplet to a (reentrant) dispersed state again (*Figure 9a*), whereas the initial dense

pY-Caprin1 droplet becomes increasingly dispersed monotonically (*Figure 9b*). Colocalization of $(ATP\text{-}Mg)^{2-}$ with both the Caprin1 (*Figure 9c*) and pY-Caprin1 (*Figure 9d*) droplets is clearly visible at $[(ATP\text{-}Mg)^{2-}] = 0.03b^{-3}$, though the degree of colocalization is appreciably higher for Caprin1 than for pY-Caprin1. This is likely because the positive net charge of Caprin1 is more attractive to $(ATP\text{-}Mg)^{2-}$. By comparison, variations in $Na^+$ and $Cl^-$ distribution between Caprin1/pY-Caprin1 dilute and condensed phases are not so discernible in *Figure 9e–h*, consistent with the small differences in the corresponding FTS correlation functions (*Figure 8d and e*).

### Robustness of general trends predicted by FTS

We have also assessed the generality of the results in *Figures 8 and 9* by considering three variations in the molecular species treated by FTS: (i) Caprin1 or pY-Caprin1 with only $Na^+$ and $Cl^-$ but no $(ATP\text{-}Mg)^{2-}$ (*Appendix 1—figure 4*), (ii) Caprin1 with $(ATP\text{-}Mg)^{2-}$ and either $Na^+$ or $Cl^-$ (but not both) to maintain overall charge neutrality or pY-Caprin1 with $(ATP\text{-}Mg)^{2-}$ and $Na^+$ as counterion but no $Cl^-$ (*Appendix 1—figure 5*), and (iii) Caprin1 or pY-Caprin1 with $ATP^{4-}$, $Mg^{2+}$, $Na^+$ and $Cl^-$ (*Appendix 1—figure 6*). Despite these variations in FTS models, *Appendix 1—figures 4–6* consistently show reentrant behavior for Caprin1 but not pY-Caprin1 and *Appendix 1—figures 5 and 6* both exhibit colocalization of ATP with condensed Caprin1, suggesting that these features are robust consequences of the basic electrostatics at play in Caprin1/pY-Caprin1 + ATP-Mg + NaCl systems.

## Discussion

It is reassuring that, in agreement with experiment, all of our electrostatics-based theoretical approaches consistently predict salt-dependent reentrant phase behaviors for Caprin1, whereas pY-Caprin1 LLPS propensity decreases monotonically with increasing salt (*Figures 2, 4, 8 and 9*). This effect applies to small monovalent salts exemplified by $Na^+$ and $Cl^-$ as well as to our electrostatics-based single- and multiple-bead models of $(ATP\text{-}Mg)^{2-}$ or $ATP^{4-}$, with ATP exhibiting a significant colocalization with the Caprin1 condensed phase (*Figures 2g, h, k, l , and 9c*) attributable to the higher valency of $(ATP\text{-}Mg)^{2-}$ and $ATP^{4-}$ than that of monovalent ions. As mentioned above, the difference in salt-dependent LLPS of Caprin1 and pY-Caprin1 originates largely from the polyelectrolytic nature of Caprin1 and the polyampholytic nature of pY-Caprin1 (*Lin et al., 2020*) corresponding, respectively, to the 'high net charge' and 'screening' classes of IDPs in a more recent analysis (*MacAinsh et al., 2024*).

### Related studies of electrostatic effects on biomolecular condensates

Our theoretical predictions are also largely in agreement with recent computational studies on salt concentrations in the dilute versus condensed phases (*Zheng et al., 2020*) and salt-dependent reentrant behaviors (*Krainer et al., 2021*) of other biomolecular condensates, including explicit-water, explicit-ion atomic simulations with preformed condensates of the N-terminal RGG domain of LAF-1 (*Zheng et al., 2020*) and of the highly positive proline-arginine 25-repeat dipeptide $PR_{25}$ (*Garaizar and Espinosa, 2021*).

A recent study examines salt-dependent reentrant LLPSs of full-length FUS (WT and G156E mutant), TDP-43, bromodomain-containing protein 4 (Brd4), sex-determining region Y-box 2 (Sox2), and annexin A11 (*Krainer et al., 2021*). Unlike the requirement of a nonzero monovalent salt concentration for Caprin1 LLPS, LLPS is observed for all these six proteins with KCl, NaCl, or other salts at concentrations as low as 50 mM. Also unlike Caprin1, their protein condensates dissolve at intermediate salt then re-appear at higher salt, a phenomenon the authors rationalize by a tradeoff between decreasing favorability of cation-anion interactions and increasing favorability of cation-cation, cation-π, hydrophobic, and other interactions with increasing monovalent salt (*Krainer et al., 2021*).

Two reasons may account for this difference. First, Caprin1 does not phase separate at low salt because it is a relatively strong polyelectrolyte (NCPR = +13/103 = +0.126). By comparison, five of the six proteins in *Krainer et al., 2021* are much weaker polyelectrolytes or not at all, with NCPR = +14/526 = +0.0266, +13/526 = +0.0247, −7/80 = −0.0875, 0, and +3/326 = +0.00920, respectively, for FUS (WT, mutant), TDP-43, Brd4, and A11. Apparently, their weak electrostatic repulsions can be overcome by favorable nonelectrostatic interactions alone to enable LLPS.

Second, compared to Caprin1, the proteins in *Krainer et al., 2021* are either significantly larger (WT and mutant FUS) or significantly more hydrophobic and aromatic (the other four proteins), both

properties are conducive to LLPS. For instance, although Sox2's NCPR = +14/88 = +0.159 is higher than that of Caprin1, among Sox2's amino acid residues, 21/88 = 23.9% are large hydrophobic or aromatic residues leucine (L), isoleucine (I), valine (V), methionine (M), phenylalanine (F), or tryptophan (W), and 17/88 = 19.3% are large aliphatic residues L, I, V, or M. This amino acid composition suggests that hydrophobic or π-related interactions in Sox2 can be sufficient to overcome electrostatic repulsion to effectuate LLPS at zero salt. In contrast, the Caprin1 IDR contains merely one L; only 10/103 = 9.7% of the residues of Caprin1 are in the L, I, V, M, F, W hydrophobic/aromatic category and only 6/103 = 5.8% are in the L, I, V, M aliphatic category. The corresponding aliphatic fractions of TDP-43, Brd4 and A11, at 21/80 = 26.3%, 33/132 = 25%, and 90/326 = 27.6%, respectively, are also significantly higher than that of Caprin1.

## Effects of salt on biomolecular LLPS

Effects of salts on LLPS, including partitioning of salt into polymer-rich phases, are of long-standing interest in polymer physics (*Jedlinska and Riggleman, 2023*). In the biomolecular condensate context, the versatile functional roles of salts are highlighted by the interplay between electrostatic and cation-π interactions (*Hazra and Levy, 2023*; *Kim et al., 2017*), salts' modulating effects on heat-induced LLPSs of RNAs (*Wadsworth et al., 2023*), their regulation of condensate liquidity (*Morishita et al., 2023*), and even their potential impact in extremely high-salt exobiological environments (*Fetahaj et al., 2021*). While some of these recent studies focus primarily on salts' electrostatic screening effects without changing the signs of the effective polymer charge-charge interaction (*Hazra and Levy, 2023*), effective attractions between like charges bridged by salt or other oppositely-charged ions (*Orkoulas et al., 2003*) as illustrated by Caprin1 (*Figures 5f and 6*) and a recent study of A1-LCD (*MacAinsh et al., 2024*) are likely needed to account for phenomena such as salt-induced dimerization of highly charged, medically relevant arginine-rich cell-penetrating short peptides (*Tesei et al., 2017*; *Liu et al., 2022*). In this regard, it should be noted that positively and negatively charged salt ions can also coordinate with backbone carbonyls and amides, respectively, in addition to coordinating with charged amino acid sidechains (*MacAinsh et al., 2024*). The impact of such effects, which are not considered in the present coarse-grained models, should be ascertained by further investigations using atomic simulations (*MacAinsh et al., 2024*; *Rauscher and Pomès, 2017*; *Zheng et al., 2020*).

## Tielines in protein-salt phase diagrams

In view of Caprin1's polyelectrolytic nature, the mildly negative tieline slopes in *Figure 2a and b* are consistent with rG-RPA predictions for a fully charged polyelectrolyte (Figure 10a of *Lin et al., 2020*). This depletion of monovalent salt in the condensed phase is similar to that observed in the complex coacervation of oppositely charged polyelectrolytes (*Radhakrishna et al., 2017*; *Li et al., 2018*; *Herrera et al., 2023*). By comparison, the positive rG-RPA tieline slopes for polyampholytic pY-Caprin1 (*Figure 2c and d*), confirmed by MD in *Figure 4f and h*, are appreciably steeper than that predicted for fully charged (±1) diblock polyampholytes by rG-RPA and the essentially flat tielines predicted by FTS (Fig. 10b of *Lin et al., 2020* and Figure 7 of *Danielsen et al., 2019*). Whether this difference originates from the presence of divalently charged (−2) phosphorylated sites in pY-Caprin1 remains to be elucidated. In any event, tieline analysis is generally instrumental for revealing details, such as stoichiometry, of the interactions driving multiple-component biomolecular (*Lin et al., 2022*; *Posey et al., 2024*; *Qian et al., 2022*); rG-RPA should be broadly useful as a computationally efficient tool for this purpose (*Lin et al., 2020*).

## Counterion valency

Our rG-RPA prediction that the maximum condensed-phase [Caprin1] at low [Na$^+$] is substantially higher with divalent than with monovalent counterions is in line with early findings that higher-valency counterions are more effective in bridging polyelectrolyte interactions to favor LLPS (*Olvera de la Cruz et al., 1995*) and recent observations that salt ions with higher valencies enhance biomolecular LLPS (*Lenton et al., 2021*; *Crabtree et al., 2023*). The possibility that this counterion/salt effect on LLPS may be exploited more generally for biological functions and/or biomedical applications remains to be further explored. In this regard, while recognizing that ATP can engage in π-related interactions (*Kota et al., 2024*; *Kang et al., 2018*; *Kang et al., 2019*), our electrostatics-based perspective of ATP-dependent reentrant phase behaviors is consistent with recent observations on polylysine LLPS

modulated by enzymatically catalyzed ATP turnovers (*Herrera et al., 2023*; *Nakashima et al., 2018*). More broadly, differential effects of salt ions on biomolecular LLPS can also arise from the sizes and charge densities of the ions—properties related to the Hofmeister phenomena (*Hribar et al., 2002*; *Lo Nostro and Ninham, 2012*)—even for ions with the same valency (*Posey et al., 2024*). These features should be addressed in future theoretical models as well.

### Prospective extensions of the present theoretical methodology

Beyond the above comparisons, further experimental testing of other aspects of our theoretical predictions should be pursued, especially those pertaining to pY-Caprin1. Future theoretical efforts should address a broader range of scenarios by independent variations of $[ATP^{4-}]$, $[Mg^{2+}]$, $[Na^+]$, $[Cl^-]$ and to account for nonelectrostatic aspects of ATP-Mg dissociation (*Wessén et al., 2022b*) with predictions such as tieline slopes analyzed in detail to delineate effects of sizes, charge densities (*Posey et al., 2024*), and configurational entropy of salt ions (*Adhikari et al., 2018*) as well as solvent quality (*Li et al., 2021*). In addition to our basic modeling constructs, the impact of excluded volume and solvent/cosolute-mediated temperature-dependent effective interactions should be incorporated. Excluded volume is known to affect LLPS (*Danielsen et al., 2019*), demixing of IDP species in condensates (*Pal et al., 2021*), and partition of salt ions in polymer LLPS (*Li et al., 2018*). Moreover, LCST can be driven not only by hydrophobicity (*Cinar et al., 2019*; *Lin et al., 2018*; *Dignon et al., 2019a*) but also by electrostatics, as suggested by experiment on complex coacervates of oppositely charged polyelectrolytes (*Ali et al., 2019*). Bringing together these features into a comprehensive formulation will afford a more accurate physical picture.

### Summary

To recapitulate, we have employed three complementary theoretical and computational approaches to account for the interplay between sequence pattern, phosphorylation, counterion, and salt in the phase behaviors of IDPs. Application to the Caprin1 IDR and its phosphorylated variant pY-Caprin1 provides physical rationalization for a variety of trends observed in experiments, including reentrance behaviors and very substantial ATP colocalization. These findings support a significant—albeit not exclusive—role of electrostatics in these biophysical phenomena, providing physical insights into effects of sequence-specific charge-charge interactions on ATP-modulated physiological functions of biomolecular condensates such as regulation of ion concentrations. The approach developed here should be of general utility as a computationally efficient tool for hypothesis generation, design of new experiments, exploration and testing of biophysical scenarios, as well as a starting point for more sophisticated theoretical/computational modeling.

## Materials and methods

Further details of the experimental and theoretical/computational methodologies outlined below are provided in *Appendix 1*.

### Experimental sample preparation

The low complexity 607–709 domain of Caprin1 was expressed and purified as before (*Kim et al., 2019*; *Kim et al., 2021*). WT Caprin1 was used in all experiments except those on [NaCl] dependence reported in *Table 1* and *Figure 3a*, for which a double mutant was used because residue pairs N623-G624 and N630-G631 in WT Caprin1 form isoaspartate (IsoAsp) glycine linkages over time which alters the charge distribution of the IDR (*Wong et al., 2020*).

### Phosphorylation of the Caprin1 IDR

Phosphorylation of the WT Caprin1 IDR was performed as described in our prior study (*Kim et al., 2019*) by using the kinase domain of mouse Eph4A (587-896) (*Wiesner et al., 2006*) with an N-terminal His-SUMO tag.

### Determination of phase diagrams

We established phase diagrams for Caprin1 and pY-Caprin1 by measuring the protein concentrations in dilute and condensed phases across a range of [NaCl]s (*Figure 1c and d*). Initially homogenizing the

two phases of the demixed samples into a milky dispersion through vortexing, ~200 μL aliquots were then incubated in a PCR thermocycler with a heated lid at 90 °C, in triplicate, for a minimum of one hour. During incubation, the condensed phase settled and formed a clear phase at the bottom. For concentration measurements, the samples were diluted in 6 M GdmCl and 20 mM NaPi (pH 6.5). The dilute phase (top layer) was analyzed through a tenfold dilution of 10 μL samples, and the condensed phase (bottom layer) was analyzed through 250- to 500-fold dilution of 2 or 10 μL samples. Notably, using a positive displacement pipettor (Eppendorf) and tips was essential for accurately pipetting the viscous condensed phase.

## Concentrations of salt and ATP-Mg in dilute and condensed phases

Inductively coupled plasma optical emission spectroscopy (ICP-OES) measurements of [Na$^+$] were performed using a Thermo Scientific iCAP Pro ICP-OES instrument in axial mode. ICP-OES was also used to determine [ATP] and [Mg$^{2+}$] (**Table 2**). The detection of phosphorus and magnesium served as proxies for quantifying ATP and Mg$^{2+}$ levels, respectively. Standard curves were prepared using solutions with known [ATP] and [Mg$^{2+}$], ranging from 0 to 90 ppm for ATP and 0–25 ppm for Mg$^{2+}$.

## Caprin1 phase separation propensity at high-salt concentrations

A 6 mM solution of double-mutant Caprin1 IDR (see above) in buffer (25 mM sodium phosphate, pH 7.4) was prepared by exchanging (3 times) the purified protein after size exclusion chromatography using centrifugal concentrators (3 kDa, EMD Millipore). Caprin samples for turbidity measurements were prepared by taking 0.5 μL of the above solution and diluting it into buffer (25 mM sodium phosphate, pH 7.4) containing varying [NaCl]s ranging from 0 to 4.63 M, in a sample volume of 9 μL, so as to achieve [Caprin1] of 300 μM. After rigorous mixing, 5 μL samples were loaded into a μCuvette G1.0 (Eppendorf). OD600 measurements (**Figure 3a**) were recorded three times using a BioPhotometer D30 (Eppendorf).

## [ATP-Mg]-dependent Caprin1 phase behaviors

Turbidity assays were conducted using the method we described previously (**Wong et al., 2020**).

## Sequence-specific theory of heteropolymer phase separation

As detailed in **Wessén et al., 2022a**; **Lin et al., 2023**, an example of the sequence-specific polymer theories (**Lin et al., 2016**; **Lin et al., 2020**) is that for a solution with a single species of charged heteropolymers in $n_{\mathrm{p}}$ copies, $n_{\mathrm{c}}$ counterions (same type), and $n_{\mathrm{s}}$ salt ions (same type, but different from the counterions). Each polymer chain has $N$ monomers (residues) with charge sequence $|\sigma\rangle = [\sigma_1, \sigma_2, ...\sigma_N]^{\mathrm{T}}$ in vector notation, where $\sigma_i \in \{0, \pm 1, -2\}$ is the charge of the $i$th residue. The counterions and salt ions are monomers carrying $z_{\mathrm{c}}$ and $z_{\mathrm{s}}$ charges, respectively. The particle-based partition function is given by

$$\mathcal{Z} = \frac{1}{n_{\mathrm{p}}!n_{\mathrm{c}}!n_{\mathrm{s}}!n_{\mathrm{w}}!} \int \prod_{\alpha=1}^{n_{\mathrm{p}}} \prod_{i=1}^{N} d\mathbf{R}_{\alpha,i} \prod_{a=1}^{n_{\mathrm{s}}+n_{\mathrm{c}}} d\mathbf{r}_a e^{-\mathcal{T}[\mathbf{R}]-\mathcal{U}[\mathbf{R},\mathbf{r}]} , \tag{2}$$

where $n_{\mathrm{w}}$ denotes the number of water (solvent) molecules, $\mathbf{R}_{\alpha,i}$ is the position vector of the $i$th residue of the $\alpha$th polymer, $\mathbf{r}_a$ is the position vector of the $a$th small ion. $\mathcal{T}$ accounts for polymer chain connectivity modeled by a Gaussian elasticity potential with Kuhn length $l$. $\mathcal{U}$ describes the interactions among all molecular components of the system, here consisting only of Coulomb electrostatics (el) and excluded-volume (ex) for simplicity, viz., $\mathcal{U} = \mathcal{U}_{\mathrm{el}} + \mathcal{U}_{\mathrm{ex}}$. Their interaction strengths are governed by the Bjerrum length $l_{\mathrm{B}}$ and the two-body excluded volume parameter $v_2$. By introducing conjugate fields $\psi(\mathbf{r})$, $w(\mathbf{r})$ and applying the Hubbard-Stratonovich transformation, the system defined by the particle-based partition function in **Equation 2** is recast as a field theory of $\psi, w$ in which their interactions with polymer, salt, and counterion are described, respectively, by single-molecule partition functions $\mathcal{Q}_{\mathrm{p}}$, $\mathcal{Q}_{\mathrm{s}}$, and $\mathcal{Q}_{\mathrm{c}}$. For instance,

$$\mathcal{Q}_{\mathrm{p}}[\psi, w] = \int \prod_{i=1}^{N} d\mathbf{R}_i \exp\left(-\mathcal{H}_{\mathrm{p}}\left[\mathbf{R}; \psi, w\right]\right) \tag{3}$$

where $\mathcal{H}_{\mathrm{p}}$ is the single-polymer Hamiltonian and the chain label $\alpha$ is dropped.

## Renormalized-Gaussian random-phase-approximation (rG-RPA)

Following *Sawle and Ghosh, 2015*; *Lin et al., 2020*, $\mathcal{H}_p$ can be separated into a Gaussian-chain Hamiltonian with an effective (renormalized) Kuhn length $l_1 = xl$ and a remaining term, $\mathcal{H}_p = \mathcal{H}_p^0 + \mathcal{H}_p^1$, where

$$\mathcal{H}_p^0 = \frac{3}{2l^2 x} \sum_{i=1}^{N-1} \left( \mathbf{R}_{i+1} - \mathbf{R}_i \right)^2, \tag{4a}$$

$$\mathcal{H}_p^1 = \frac{3}{2l^2} \left( 1 - \frac{1}{x} \right) \sum_{i=1}^{N-1} \left( \mathbf{R}_{i+1} - \mathbf{R}_i \right)^2 - i \sum_{i=1}^{N} \left[ \sigma_i \psi(\mathbf{R}_i) + w(\mathbf{R}_i) \right], \tag{4b}$$

with $i^2 = -1$. By requiring the observable polymer square end-to-end distance be properly quantified by $\mathcal{H}_p^0$, $x$ can be approximated by variational theory (*Sawle and Ghosh, 2015*). RPA can then be applied to the renormalized Gaussian (rG) chain system with $l \to xl$ and a corresponding scaling of the contour length to arrive at an improved theory, rG-RPA, for sequence-specific LLPS.

## Explicit-ion coarse-grained molecular dynamics (MD)

The MD model in this work augments a class of implicit-water coarse-grained models (*Dignon et al., 2018*; *Das et al., 2020*) that utilize a 'slab' approach for efficient equilibration (*Silmore et al., 2017*) by incorporating explicit small ions. As before (*Das et al., 2020*), the total MD potential energy $U_T$ is the sum of long-spatial-range electrostatic (el) and short-spatial-range (sr) interactions of the Lennard-Jones (LJ) type as well as bond interactions, that is, $U_T = U_{el} + U_{sr} + U_{bond}$. With small ions, the electrostatic component is given by a sum of polymer-polymer (pp), polymer-ion (pi), and ion-ion (ii) contributions: $U_{el} = U_{el,pp} + U_{el,pi} + U_{el,ii}$. Details of these terms are provided in *Appendix 1*.

## Field-theoretic simulation (FTS)

FTS is useful for sequence-specific multiple-component LLPSs encountered in biomolecular settings. The new applications developed here are based on recent advances (see, e.g., *McCarty et al., 2019*; *Danielsen et al., 2019*; *Pal et al., 2021*; *Fredrickson et al., 2002*; *Lin et al., 2023*; *Fredrickson, 2006*). Consider the field theoretic Hamiltonian

$$H[w, \psi] = \int d\mathbf{r} \left( \frac{[\nabla \psi(\mathbf{r})]^2}{8\pi l_B} + \frac{w(\mathbf{r})^2}{2v_2} \right) - \sum_m n_m \ln \mathcal{Q}_m[\breve{w}, \breve{\psi}], \tag{5}$$

where $\mathcal{Q}_m$ is single-molecule partition function (here $m$ labels the components in the system, *Equation 3*) and the breves denote convolution with $\Gamma$, i.e., for a generic field $\phi$, $\breve{\phi}(\mathbf{r}) = \Gamma \star \phi(\mathbf{r}) \equiv \int d\mathbf{r}' \Gamma(\mathbf{r} - \mathbf{r}')\phi(\mathbf{r}')$; here $\phi = w, \psi$, and $\Gamma$ is a Gaussian smearing function (*Lin et al., 2023*). FTS utilizes the Complex-Langevin (CL) method (*Parisi, 1983*; *Klauder, 1983*) by introducing an artifical CL time variable ($t$), viz., $w(\mathbf{r}) \to w(\mathbf{r}, t)$, $\psi(\mathbf{r}) \to \psi(\mathbf{r}, t)$ and letting the system evolve in CL time in accordance with a collection of Langevin equations

$$\frac{\partial \phi(\mathbf{r}, t)}{\partial t} = -\frac{\delta H}{\delta \phi(\mathbf{r}, t)} + \eta_\phi(\mathbf{r}, t), \quad \phi = w, \psi, \tag{6}$$

where the Gaussian noise $\eta_\phi(\mathbf{r}, t)$ satisfies $\langle \eta_\phi(\mathbf{r}, t)\eta_{\phi'}(\mathbf{r}', t') \rangle = 2\delta_{\phi, \phi'}\delta(\mathbf{r} - \mathbf{r}')\delta(t - t')$. Thermal averages of thermodynamic observables are then computed as asymptotic CL time averages of the corresponding field operators. Spatial information about condensation and proximity of various components is readily gleaned from density-density correlation functions (*Pal et al., 2021*; *Lin et al., 2023*),

$$G_{m,n}(|\mathbf{r} - \mathbf{r}'|) = \langle \hat{\rho}_m(\mathbf{r})\hat{\rho}_n(\mathbf{r}') \rangle, \tag{7}$$

where $m, n$ are labels for the components in the model system. For instance, $m$ may represent all polymer beads (denoted 'p') irrespective of the sequence positions of the beads $[\hat{\rho}_p(\mathbf{r}) = \sum_{\alpha=1}^{n_p} \sum_{i=1}^{N} \Gamma(\mathbf{r} - \mathbf{R}_{\alpha,i})]$, and $n$ may represent all six beads in our ATP-Mg model (*Figure 8a*). One may also define

$$G_{pq}^{(i)}(|\mathbf{r} - \mathbf{r}'|) \equiv \langle \hat{\rho}_{p,i}(\mathbf{r})\hat{\rho}_q(\mathbf{r}') \rangle, \tag{8}$$

where ($i$) represents the $i$th residue along a protein chain [$\hat{\rho}_{p,i}(\mathbf{r}) \equiv \sum_{\alpha=1}^{n_p} \Gamma(\mathbf{r} - \mathbf{R}_{\alpha,i})$ is the density of the $i$th residue among all the protein chains], and q = (ATP-Mg)$^{2-}$, Na$^+$, or Cl$^-$. With this definition, residue-specific relative contact frequencies are estimated by integrating *Equation 8* over a spherical volume within a small inter-component distance $r_{contact}$:

$$\mathcal{G}_{pq}^{(i)} \equiv 4\pi \int_0^{r_{contact}} dr\, r^2 G_{pq}^{(i)}(r). \tag{9}$$

For the normalized $\mathcal{G}_{pq}^{(i)}/\rho_{p,i}^0 \rho_q^0$ plotted in *Figure 8f*, $\rho_{p,i}^0$ and $\rho_q^0$ are bulk (overall) densities, respectively, of the $i$th protein residue and of (ATP-Mg)$^{2-}$ or small ions, and $r_{contact} \approx 1.5b$ is used to characterize contacts. Further details are provided in *Appendix 1*.

## Acknowledgements

This work was supported by Canadian Institutes of Health Research (CIHR) grant NJT-155930 and Natural Sciences and Engineering Research Council of Canada (NSERC) grant RGPIN-2018–04351 to HSC, CIHR grant FDN-148375, NSERC grant RGPIN-2016–06718, and Canada Research Chairs Program to JDFK as well as CIHR grant FDN-503573 to LEK. THK was supported by a CIHR Banting Postdoctoral Fellowship and the H. L. Holmes Award for Postdoctoral Studies from the National Research Council of Canada. AKR was supported by a CIHR Postdoctoral Fellowship. We are grateful for the computational resources provided generously by Compute/Calcul Canada and the Digital Research Alliance of Canada.

## Additional information

### Funding

| Funder | Grant reference number | Author |
| --- | --- | --- |
| Canadian Institutes of Health Research | NJT-155930 | Hue Sun Chan |
| Natural Sciences and Engineering Research Council of Canada | RGPIN-2018-04351 | Hue Sun Chan |
| Canadian Institutes of Health Research | FDN-148375 | Julie D Forman-Kay |
| Natural Sciences and Engineering Research Council of Canada | RGPIN-2016-06718 | Julie D Forman-Kay |
| Canadian Institutes of Health Research | FDN-503573 | Lewis E Kay |

The funders had no role in study design, data collection and interpretation, or the decision to submit the work for publication.

### Author contributions

Yi-Hsuan Lin, Suman Das, Tanmoy Pal, Jonas Wessén, Conceptualization, Software, Formal analysis, Investigation, Methodology, Writing – original draft; Tae Hun Kim, Conceptualization, Data curation, Formal analysis, Investigation, Methodology; Atul Kaushik Rangadurai, Conceptualization, Data curation, Investigation, Methodology; Lewis E Kay, Julie D Forman-Kay, Conceptualization, Supervision, Funding acquisition, Investigation, Project administration; Hue Sun Chan, Conceptualization, Formal analysis, Supervision, Funding acquisition, Investigation, Writing – original draft, Project administration, Writing – review and editing

### Author ORCIDs

Jonas Wessén ⓘ https://orcid.org/0000-0002-5904-8442
Lewis E Kay ⓘ https://orcid.org/0000-0002-4054-4083

Julie D Forman-Kay ⓘ https://orcid.org/0000-0001-8265-972X
Hue Sun Chan ⓘ https://orcid.org/0000-0002-1381-923X

Reviewer #1 (Public review): https://doi.org/10.7554/eLife.100284.3.sa1
Reviewer #2 (Public review): https://doi.org/10.7554/eLife.100284.3.sa2
Reviewer #3 (Public review): https://doi.org/10.7554/eLife.100284.3.sa3
Author response https://doi.org/10.7554/eLife.100284.3.sa4

## Additional files

### Supplementary files
MDAR checklist

### Data availability
All data generated or analyzed during this study are included in the manuscript, including source data for figures.

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

## Appendix 1

### Supplementary materials and methods

### Experimental information additional to that in the maintext

#### Sample preparation – Wildtype (WT) Caprin1

As stated in the maintext, WT Caprin1 was used in all reported experiments except those on [NaCl] dependence presented in maintext *Table 1* and *Figure 3a*. The amino acid sequence of WT Caprin1 is given in *Appendix 1—figure 1* (all supporting figures are provided in this *Appendix 1*). The preparation of sample WT Caprin1 is now briefly described as follows: Caprin1 with an N-terminal His-SUMO tag was produced in BL21 (DE3)-RIPL Codon Plus *E. coli* cells. These cells were cultured until an optical density at 600 nm (OD600) of 0.6 at 37 °C and then induced with 0.5 mM IPTG for overnight expression at 23 °C. The harvested cells were suspended in a lysis buffer containing 6 M guanidinium chloride (GdmCl), 25 mM Tris, 500 mM NaCl, 20 mM imidazole, 2 mM β-mercaptoethanol (BME), at pH 8.0, and lysed via sonication. The supernatant, post-sonication, was applied to Ni-NTA (Cytiva) and washed with lysis, wash (25 mM Tris, 500 mM NaCl, 20 mM imidazole, 2 mM BME, at pH 8.0), and elution (25 mM Tris, 500 mM NaCl, 300 mM imidazole, 2 mM BME, at pH 8.0) buffers. Post-elution, the sample was treated with ULP1 during dialysis against a dialysis buffer (25 mM Tris, 250 mM NaCl, and 2 mM BME at pH 8.0). This step was followed by His-SUMO tag removal through Ni-NTA column chromatography. Final purification of Caprin1 was performed using FPLC with a Superdex 75 16/60 column, equilibrated with a gel filtration buffer (3 M GdmCl, 25 mM Tris, 500 mM NaCl, 2 mM BME, pH 8.0). The protein fractions were then dialyzed twice to remove GdmCl before use in experiments.

#### Sample preparation – Double-mutant (N623T, N630T) variant of Caprin1

Our sample preparation for the double-mutant variant used in [NaCl] dependence studies reported in maintext *Table 1* and *Figure 3a* proceeded as follows. To abolish IsoAsp formation for the salt concentration measurements (*Table 1*) and the [NaCl]-dependent turbidity measurements in *Figure 3a*, we used a double mutant of the Caprin1 IDR (N623T,N630T) in which the two asparagine residues are mutated to threonine. This double mutant, which has been shown to exhibit a similar propensity to phase separate as the WT Caprin1 IDR (*Toyama et al., 2022*), was purified as described previously (*Wong et al., 2020*; *Toyama et al., 2022*).

Purified Caprin1 was first exchanged into buffer (25 mM sodium phosphate, pH 7.4) via dialysis and was concentrated to ~6 mM using 3 kDa centrifugal Amicon concentrators (EMD Millipore). The pH of the concentrated protein was adjusted to 7.4 using concentrated hydrochloric acid. Phase-separated samples of Caprin1 were prepared by addition of a concentrated stock solution of NaCl (25 mM sodium phosphate, pH 7.4, 4 M NaCl) to achieve a bulk (overall) salt concentration of 300 mM NaCl. Condensed and dilute phases of Caprin1 were transferred all together into an Eppendorf tube using a syringe. After rigorous vortexing, the phase-separated samples were incubated at the desired temperatures using a thermocycler with a heated lid (95 °C). At least 1 hr was required to allow droplets to form a large condensed phase droplet at the bottom of the tube.

2 µL of condensed and dilute phases were pipetted into 48 µL of a 2.8 M urea solution (U4883, Sigma) in MilliQ water in a 15 mL falcon tube, using a positive and an air displacement pipette, respectively. The outside of the tips were wiped with a KimWipe to remove excess protein, prior to transferring into the urea solutions. Following transfer, the samples were digested for inductively coupled plasma optical emission spectroscopy (ICP-OES) measurements by the addition of 630 µL of concentrated nitric acid (67%, NX0407, Sigma) and 630 µL hydrogen peroxide (95321, Sigma), and incubated in an oven at 60 °C for 54 hr. Post digestion, the sample tubes were cooled at room temperature and centrifuged. 40 µL of hydrogen peroxide was then added to the samples. No bubbles were observed, indicating the completion of the digestion process. The samples were then bought up to 12 mL using MilliQ water, to achieve a final nitric acid concentration of 3.5%. Blank samples for the condensed and dilute phases were prepared by pipetting 2 µL of MilliQ water using a positive and an air displacement pipette, respectively, and subsequently following the digestion protocol described above. Sodium standards (0.1, 0.2, 0.5, 1, 2, 4, 8 and 10 ppm) in 3.5% nitric acid for ICP-OES measurements were prepared by dilution of sodium standard solution (00462, Sigma) with MilliQ water and concentrated nitric acid (67%, NX0407, Sigma). All samples were filtered using

0.22 µm syringe filters prior to ICP-OES. Condensed and dilute phases were drawn in triplicate at each temperature.

## Phosphorylation of the Caprin1 IDR

The purified protein was initially concentrated to 25–50 µM in a reaction buffer comprising 25 mM Tris pH 7.4, 50 mM KCl, 10 mM MgCl$_2$, 3 mM ATP and 2 mM DTT. This mixture was then placed into a dialysis tubing with a 3 kDa cut-off. To the protein sample, purified His-SUMO-Eph4A was added to 5–10 µM, and the reaction mixture was subsequently dialyzed against 4 liters of the same reaction buffer, either at room temperature or at 4 °C overnight. Mass spectrometry indicates that the resulting sample consists mainly of a mixture of Caprin1 IDRs with six or seven phosphorylations and a very small fraction of IDRs with five phosphorylations (*Appendix 1—figure 2*). After confirming the phosphorylation status, pY-Caprin1 was further purified using a Superdex 75 (16/600) column.

## Determination of phase diagrams

The initial homogenization described in the maintext ensures that the condensed phase in small droplets can rapidly equilibrate with the dilute phase. Absorbance at 280 nm was measured and converted to concentration using the Beer-Lambert law, with an extinction coefficient (ε) of 10,430 M$^{-1}$cm$^{-1}$, based on the molecular masses of 11,108 Da for Caprin1 and 11,668 Da for pY-Caprin1. The reported concentration values and uncertainties, calculated as means and standard deviations, were derived from triplicate measurements.

## Concentrations of salt and ATP-Mg in dilute and condensed phases

ICP-OES measurements were performed in triplicate for each sample. Mean value and uncertainty for the salt concentration were obtained by taking the average and standard deviation over the triplicate samples at the given temperature (maintext *Table 1*). Specific details of sample preparation for the set of ATP-Mg–dependent experiments are provided above in this *Appendix 1*. As for salt-dependent experiments, these measurements were performed in triplicate and standard deviations were calculated to assess experimental uncertainties (maintext *Table 2*).

## Caprin1 phase separation propensity at high-salt concentrations

Averages and standard deviations over the three OD600 measurements were reported by the blue symbols in maintext *Figure 3a*.

## [ATP-Mg]-dependent Caprin1 phase behaviors

A brief summary of the turbidity assays in *Wong et al., 2020* that we utilized is as follows: The WT Caprin1 IDR was diluted to a 200 µM concentration using a buffer composed of 25 mM HEPES and 2 mM DTT at pH 7.4, with varying levels of ATP-Mg. Samples were prepared with ATP-Mg concentrations ranging from 0 to 40 mM. Following thorough mixing, 5 µL of each sample was placed into a µCuvette G1.0 (Eppendorf), and OD600 was measured using a BioPhotometer D30 (Eppendorf). This procedure was performed three times for analysis of experimental uncertainties (red symbols in maintext *Figure 3a*).

## Sequence-specific theory of heteropolymer phase separation – Summary of key steps in the field-theoretic formulation

The following is a more extensive summary to supplement the brief outline in *Materials and methods* of the maintext provided under the heading 'sequence-specific theory of heteropolymer phase separation'. In general, sequence-specific polymer field theories (*Lin et al., 2016*; *Lin et al., 2020*) are constructed to model systems of polymers with various salt and counterions. Further details are available from our recent publications (e.g. *Lin et al., 2023*; *Wessén et al., 2022a* and references therein).

Using the same notation for the partition function $\mathcal{Z}$ in *Equation 2*, with $\mathcal{T} + \mathcal{U}$ being the Hamiltonian in units of the product $k_\mathrm{B}T$ of Boltzmann's constant $k_\mathrm{B}$ and absolute temperature $T$, the connectivity term $\mathcal{T}$ is given by

$$\mathcal{T}[\mathbf{R}] = \frac{3}{2l^2} \sum_{\alpha=1}^{n_\mathrm{p}} \sum_{i=1}^{N-1} \left( \mathbf{R}_{\alpha,i+1} - \mathbf{R}_{\alpha,i} \right)^2 \tag{S1}$$

with [**R**] being shorthand for [{**R**$_{\alpha,i}$}]. Considering the case when the total potential energy $\mathscr{U}$ is taking one of its simplest forms, in that it serves only to model the two-body (pairwise) interactions among polymer residues, salt ions, and counterions, we further confine the interaction types in our formulation to Coulomb electrostatics (el) and excluded-volume (ex). As stated in the maintext, their interaction strengths are governed by the Bjerrum length $l_B$ ($l_B = e^2/(4\pi\epsilon_0\epsilon_r k_B T)$, where $e$ is protonic charge, $\epsilon_0$ is vacuum permittivity, and $\epsilon_r$ is relative permittivity), and the two-body excluded volume parameter $v_2$. We now provide the precise field-theoretic forms for the Coulomb electrostatic potential $\mathscr{U}_{el}$ and two-body excluded volume interaction $\mathscr{U}_{ex}$ terms in $\mathscr{U} = \mathscr{U}_{el} + \mathscr{U}_{ex}$ by introducing the number density operators

$$\hat{\rho}_p(\mathbf{r}) = \sum_{\alpha=1}^{n_p}\sum_{i=1}^{N}\delta(\mathbf{r} - \mathbf{R}_{\alpha,i}) \,, \tag{S2a}$$

$$\hat{\rho}_s(\mathbf{r}) = \sum_{a=1}^{n_s}\delta(\mathbf{r} - \mathbf{r}_a) \,, \tag{S2b}$$

$$\hat{\rho}_c(\mathbf{r}) = \sum_{a=n_s+1}^{n_s+n_c}\delta(\mathbf{r} - \mathbf{r}_a) \,, \tag{S2c}$$

for the monomers (residues or beads) of the polymer (p), salt (s), and counterion (c), respectively, where $\delta$ represents the Dirac $\delta$ distribution. Solvent (water) degrees of freedom [$n_w$ in **Equation 2** for $\mathcal{Z}$ in the maintext] are not included in **Equation S2** above because they are only used for the incompressibility constraint in our rG-RPA formulation as an approximate treatment of excluded volume, and solvents are not treated explicitly at all in the present field-theoretic simulation (FTS), that is, the $n_w$ factor is dropped for FTS. The corresponding charge density operators for the number density operators in **Equation S2** are

$$\hat{c}_p(\mathbf{r}) = \sum_{\alpha=1}^{n_p}\sum_{i=1}^{N}\sigma_i\delta(\mathbf{r} - \mathbf{R}_{\alpha,i}), \tag{S3a}$$

$$\hat{c}_s(\mathbf{r}) = z_s\sum_{a=1}^{n_s}\delta(\mathbf{r} - \mathbf{r}_a) = z_s\hat{\rho}_c, \tag{S3b}$$

$$\hat{c}_c(\mathbf{r}) = z_c\sum_{a=n_s+1}^{n_s+n_c}\delta(\mathbf{r} - \mathbf{r}_a) = z_c\hat{\rho}_s, \tag{S3c}$$

For $\hat{c}_p$, $\sigma_i = +1$ for arginine and lysine, $\sigma_i = -1$ for aspartic and glutamic acids, $\sigma_i = -2$ for phosphorylated tyrosine, and $\sigma_i = 0$ for all other amino acid residues in the Caprin1/pY-Caprin1 sequences studied (as stated in the maintext). $\mathscr{U}_{el}$ and are now given by

$$\mathscr{U}_{el} = \frac{l_B}{2}\int d\mathbf{r}d\mathbf{r}'\left[\hat{c}_p(\mathbf{r}) + \hat{c}_s(\mathbf{r}) + \hat{c}_c(\mathbf{r})\right]\frac{1}{|\mathbf{r} - \mathbf{r}'|}\left[\hat{c}_p(\mathbf{r}') + \hat{c}_s(\mathbf{r}') + \hat{c}_c(\mathbf{r}')\right] \tag{S4a}$$

$$\mathscr{U}_{ex} = \frac{v_2}{2}\int d\mathbf{r}\left[\hat{\rho}_p(\mathbf{r}) + \hat{\rho}_s(\mathbf{r}) + \hat{\rho}_c(\mathbf{r})\right]^2. \tag{S4b}$$

As mentioned in the maintext, two conjugate fields, $\psi(\mathbf{r})$ for Coulomb interaction and $w(\mathbf{r})$ for excluded volume, are then introduced to linearize the density operators that are quadratic in $\mathscr{U}_{el}$ and $\mathscr{U}_{ex}$ by applying the Hubbard-Stratonovich transformation (**Stratonovich, 1958**; **Hubbard, 1959**), resulting in a reformulated partition function $\mathcal{Z}' \equiv (n_p!n_s!n_c!n_w!)\mathcal{Z}$ [with $\mathcal{Z}$ given by **Equation 2** of maintext] expressed as a functional integral over the fields $\psi$ and $w$:

$$\mathcal{Z}' = \int \mathscr{D}\psi\mathscr{D}w \exp\left[-\int d\mathbf{r}\left(\frac{[\nabla\psi(\mathbf{r})]^2}{8\pi l_B} + \frac{w(\mathbf{r})^2}{2v_2}\right) + n_p\ln\mathcal{Q}_p + n_s\ln\mathcal{Q}_s + n_c\ln\mathcal{Q}_c\right], \tag{S5}$$

where $\mathcal{Q}_p$, $\mathcal{Q}_s$, and $\mathcal{Q}_c$ are single-molecule partition functions of polymer, salt ion, and counterion, respectively, which are all functionals of $\psi$ and $w$:

$$
\begin{aligned}
\mathcal{Q}_{\mathrm{p}}[\psi, w] &= \int \prod_{\tau=1}^{N} d\mathbf{R}_\tau \exp\left( -\mathcal{H}_{\mathrm{p}}\left[ \mathbf{R}; \psi, w \right] \right) \\
&= \int \prod_{\tau=1}^{N} d\mathbf{R}_\tau \exp\left[ -\frac{3}{2l^2} \sum_{i=1}^{N-1} \left( \mathbf{R}_{i+1} - \mathbf{R}_i \right)^2 - \mathrm{i} \sum_{i=1}^{N} \left( \sigma_i \psi(\mathbf{R}_i) + w(\mathbf{R}_i) \right) \right] ,
\end{aligned}
\tag{S6a}
$$

$$
\mathcal{Q}_{\mathrm{s,c}}[\psi, w] = \int d\mathbf{r} \exp\left[ -\mathrm{i}\left( z_{\mathrm{s,c}} \psi(\mathbf{r}) + w(\mathbf{r}) \right) \right],
\tag{S6b}
$$

wherein $\mathrm{i}$ is the imaginery unit, that is, $\mathrm{i}^2 = -1$.

The $\mathcal{Z}'$ in *Equation S5* can be analyzed via various field theoretic approaches. Two approaches are utilized in the present work: (i) one-loop perturbation expansion is employed to derive analytical theories based upon random-phase-approximation (RPA), and (ii) field-theoretic simulation (FTS) is conducted to compute observables numerically.

## Renormalized-Gaussian random-phase-approximation (rG-RPA)

As mentioned in the maintext, sequence-specific random phase approximation (RPA) has been applied successfully to model electrostatic effects on the LLPSs of various polyampholytic IDRs (*Lin et al., 2016*; *Lin and Chan, 2017*; *Lin et al., 2018*; *Das et al., 2020*; *Wessén et al., 2021*) to obtain behaviorial trends consistent with experiments and explicit-chain simulations; but RPA is less appropriate for polyelectrolytes with large net charge per residue (NCPR; *Ermoshkin and Olvera de la Cruz, 2003*; *Orkoulas et al., 2003*; *Mahdi and Olvera de la Cruz, 2000*) because of RPA's treatment of polymers as ideal Gaussian chains (*Muthukumar, 2017*). This approximation is reasonable for polyampholytes but not for polyelectrolytes. While overall intrachain electrostatic effects in polyampholytes can be mild because of the polymers' nearly zero net charge and thus entail only a minor perturbation on conformational statistics, repulsive electrostatics in polyelectrolytes with significant net charge is strong, leading to more rod-like conformations with statistics deviating significantly from that of Gaussian chains. Consequently, treating polyelectrolytes as Gaussian chains can lead to large errors in theoretical intrachain and interchain residue-residue (monomer-monomer) correlations, resulting in drastically overestimated LLPS propensities (*Muthukumar, 2017*).

The rG-RPA theory was put forth by some of the present authors (*Lin et al., 2020*). For a broad overview, we briefly summarize here the major methodological steps and key results of the theory. Interested readers are referred to *Lin et al., 2020* for further details. As rG-RPA has been designed and verified to tackle polyeletrolyte conformations appropriately (*Lin et al., 2020*), we apply it here to the polyelectrolytic Caprin1 IDR. Because rG-RPA allows for a smooth crossover between polyelectrolytic and polyampholytic systems, Caprin1 and pY-Caprin1 can now be analyzed in a universal theoretical formulation without invocation of ad hoc treatments for their different conformational statistics.

In our formulation of rG-RPA theory, simplifying assumptions are made to the effect that excluded volume is taken into account only between pairs of different polymer chains (no consideration of intrachain excluded volume) and small ions are treated as point charges. Denoting the input 'bare' Kuhn length as $l$, and the total free energy and volume of the system as $F$ and $\Omega$ respectively, the system free energy in units of $k_{\mathrm{B}}T$ per volume $l^3$ is given by

$$
f = \frac{F l^3}{k_{\mathrm{B}} T \Omega} = -s + f_{\mathrm{ion}} + f_0 + f_{\mathrm{p}} .
\tag{S7}
$$

Here $s$ is translational entropy

$$
-s = \frac{\phi_{\mathrm{p}}}{N} \ln \phi_{\mathrm{p}} + \phi_{\mathrm{s}} \ln \phi_{\mathrm{s}} + \phi_{\mathrm{c}} \ln \phi_{\mathrm{c}} + \phi_{\mathrm{w}} \ln \phi_{\mathrm{w}},
\tag{S8}
$$

where $\phi_{\mathrm{p}}$, $\phi_{\mathrm{s}}$, $\phi_{\mathrm{c}}$, and $\phi_{\mathrm{w}} = 1 - \phi_{\mathrm{p}} - \phi_{\mathrm{s}} - \phi_{\mathrm{c}}$ are, respectively, volume fractions of polymers, salt ions, counterions, and solvent, with the last equality following from the incompressibility condition that we have stipulated. The $f_{\mathrm{ion}}$ term in *Equation S7* accounts for the free energy of the small ions via the form

$$
f_{\mathrm{ion}} = -\frac{1}{4\pi} \left[ \ln\left( 1 + \kappa_{\mathrm{D}} l \right) - \kappa l + \frac{1}{2} \left( \kappa_{\mathrm{D}} l \right)^2 \right],
\tag{S9}
$$

where $1/\kappa_D = 1/\sqrt{4\pi l_B \left(z_s^2 \rho_s + z_c^2 \rho_c\right)}$ is the Debye screening length. The term $f_0$ in **Equation S7** is the zeroth-order excluded volume effect given by

$$f_0 = \frac{l^3}{2} v_2 \rho_m^2 \,, \tag{S10}$$

where $\rho_m = n_p N/\Omega$ is the average monmer (residue or bead) density of the polymers in the system and the expression $f_p = -(l^3/\Omega) \ln \mathcal{Z}_p$ for the last term in **Equation S7** is derived from the polymer partition function

$$\mathcal{Z}_p = \int \mathscr{D}\psi \mathscr{D}w \exp\left[-\int d\mathbf{r} \left(\frac{\psi(\mathbf{r})\left(-\nabla^2 + \kappa^2\right)\psi(\mathbf{r})}{8\pi l_B} + \frac{w(\mathbf{r})^2}{2v_2}\right) + n_p \ln \mathcal{Q}_p\right] \,. \tag{S11}$$

An analytical perturbative field theory may now be derived from $\mathcal{Z}_p$ by considering the Taylor expansion of $\ln \mathcal{Q}_p$ up to the second order of $\psi$ and $w$ while omitting terms that do not affect the relative energies of the configurations, viz.,

$$\ln \mathcal{Q}_p \approx -\frac{1}{2}\left[\left\langle \left(\hat{c}_p - \bar{c}_p\right)^2\right\rangle \psi^2 + \left\langle \left(\hat{\rho}_p - \bar{\rho}_p\right)^2\right\rangle w^2 + 2\left\langle \left(\hat{c}_p - \bar{c}_p\right)\left(\hat{\rho}_p - \bar{\rho}_p\right)\right\rangle \psi w\right], \tag{S12}$$

where $\bar{c}_p$ and $\bar{\rho}_p$ are the overall average charge and number densities, respectively, of the polymer [**Equation S2 and S3** above and Equation A37 of **Lin et al., 2020**]. It follows that $\mathcal{Z}_p$ can then be approximated as a Gaussian integral in the Fourier-transformed **k**-space,

$$\mathcal{Z}_p \approx \int \prod_{\mathbf{k}\neq 0} \sqrt{\frac{\nu_k}{v_2}} \frac{d\psi_{\mathbf{k}} dw_{\mathbf{k}}}{2\pi\Omega} \exp\left[-\frac{1}{2\Omega}\sum_{\mathbf{k}\neq 0}\langle\psi_{-\mathbf{k}} \ w_{-\mathbf{k}}| \begin{pmatrix} \nu_k + \rho_m g_{\mathbf{k}}^{cc} & \rho_m g_{\mathbf{k}}^{mc} \\ \rho_m g_{\mathbf{k}}^{mc} & v_2^{-1} + \rho_m g_{\mathbf{k}}^{mm}\end{pmatrix}\begin{vmatrix}\psi_{\mathbf{k}} \\ w_{\mathbf{k}}\end{vmatrix}\right\rangle\right], \tag{S13}$$

where $g_{\mathbf{k}}^{cc}$, $g_{\mathbf{k}}^{mm}$, and $g_{\mathbf{k}}^{mc}$ are charge-charge, mass-mass (i.e., matter-matter), and mass-charge (matter-charge) correlation functions in **k**-space, and $\nu_k = k^2/(4\pi l_B) + (z_s^2 \rho_s + z_c^2 \rho_c)$. The free energy $f_p$ is then given by

$$f_p = \frac{l^3}{2} \int \frac{d^3k}{(2\pi)^3} \ln\left[1 + \rho_m\left(\frac{g_{\mathbf{k}}^{cc}}{\nu_k} + v_2 g_{\mathbf{k}}^{mm}\right) + \frac{v_2}{\nu_{\mathbf{k}}}\rho_m^2\left(g_{\mathbf{k}}^{cc} g_{\mathbf{k}}^{mm} - \left(g_{\mathbf{k}}^{mc}\right)^2\right)\right]. \tag{S14}$$

The correlation functions in **Equation S14** may be estimated by various field-theory approximations. In rG-RPA, they are evaluated using a variational approach to the single-polymer partition functon $\mathcal{Q}_p$ by first expressing the Hamiltonian $\mathcal{H}_p$ in **Equation 3** of the maintext as the sum of an approximate Gaussian-chain Hamiltonian with an effective (renormalized) Kuhn length $xl$ (recall $l$ is the original 'bare' Kuhn length) plus the remaining term:

$$\mathcal{H}_p = \mathcal{H}_p^0 + \mathcal{H}_p^1, \tag{S15}$$

where

$$\mathcal{H}_p^0 = \frac{3}{2l^2 x}\sum_{i=1}^{N-1}\left(\mathbf{R}_{i+1} - \mathbf{R}_i\right)^2, \tag{S16a}$$

$$\mathcal{H}_p^1 = \frac{3}{2l^2}\left(1 - \frac{1}{x}\right)\sum_{i=1}^{N-1}\left(\mathbf{R}_{i+1} - \mathbf{R}_i\right)^2 - i\sum_{i=1}^{N}\left[\sigma_i\psi(\mathbf{R}_i) + w(\mathbf{R}_i)\right] \tag{S16b}$$

are the same equations as **Equation 4a and b** in the maintext and adopt essentially the same form as the unrenormalized **Equation S6a**. To make progress, we take the polymer square end-to-end distance $\mathbf{R}_{EE}^2$ as a key physical observable and require its thermodynamic average to be produced by $\mathcal{H}_p^0$ in good approximation. Based on this premise, a variational theory as described in **Sawle and Ghosh, 2015** is applied to calculate the $x$ parameter in the above **Equation S16**. Details of the derivation are given in the Appendices of **Lin et al., 2020**. Here we show only the variational equation for solving $x$:

$$1 - \frac{1}{x} - \frac{Nl^3}{18(N-1)} \int \frac{d^3k}{(2\pi)^3} \frac{k^2 \Xi_k^x}{\det \Delta_k^x} = 0, \tag{S17}$$

where $\Delta_k^x$ is the 2×2 matrix in **Equation S13** with $l \to xl$ (renormalized Kuhn length), $|i-j| \to |i-j|/x$ (renormalized contour length) and thus $l^2|i-j| \to l^2x|i-j|$ such that the correlation functions in $\Delta_k^x$ become (**Lin et al., 2020**)

$$g_{\mathbf{k}}^{cc} \to g_{k,x}^{cc} = \frac{1}{N} \sum_{i,j=1}^{N} \sigma_i \sigma_j e^{-\frac{1}{6}(kl)^2 x|i-j|}, \tag{S18a}$$

$$g_{\mathbf{k}}^{mm} \to g_{k,x}^{mm} = \frac{1}{N} \sum_{i,j=1}^{N} e^{-\frac{1}{6}(kl)^2 x|i-j|}, \tag{S18b}$$

$$g_{\mathbf{k}}^{mc} \to g_{k,x}^{mc} = \frac{1}{N} \sum_{i,j=1}^{N} \sigma_i e^{-\frac{1}{6}(kl)^2 x|i-j|}, \tag{S18c}$$

and the $\Xi_k^x$ in the integrand on the right hand side of **Equation S17** is now given by

$$\Xi_k^x = \frac{\bar{\xi}_k^x}{\nu_2} + \nu_k \bar{g}_k^x + \rho_m \left( \bar{\xi}_k^x g_{k,x}^{mm} + \bar{g}_k^x g_{k,x}^{cc} - 2\bar{\zeta}_k^x g_{k,x}^{mc} \right) \tag{S19}$$

where

$$\bar{\xi}_k^x = \frac{1}{N} \sum_{i,j=1}^{N} |i-j|^2 \sigma_i \sigma_j e^{-\frac{1}{6}(kl)^2 x|i-j|}, \tag{S20a}$$

$$\bar{g}_k^x = \frac{1}{N} \sum_{i,j=1}^{N} |i-j|^2 e^{-\frac{1}{6}(kl)^2 x|i-j|}, \tag{S20b}$$

$$\bar{\zeta}_k^x = \frac{1}{N} \sum_{i,j=1}^{N} |i-j|^2 \sigma_i e^{-\frac{1}{6}(kl)^2 x|i-j|}. \tag{S20c}$$

## The rG-RPA+FH formulation

Because the above field theory is formulated to focus only on sequence-specific electrostatics and excluded volume, in the form presented above it does not account for short-range attractions such as those arising from π-related and hydrophobic effects; but we need to take these effects into consideration to arrive at a more direct comparison between rG-RPA predictions and experiments, for example as those provided in **Figure 1c and d** of the maintext. To account for these interactions approximately in Caprin1, particularly the interactions involving π-electrons (**Kim et al., 2021**), we introduce, as before (**Lin et al., 2016**), a temperature-dependent Flory-Huggins (FH) interaction to augment the free energy $f$ in **Equation S7** (**Lin et al., 2020**; **Das et al., 2020**), resulting in an overall total free energy

$$f = -s + f_{\text{ion}} + f_0 + f_{\text{p}} - \left( \epsilon_h/T^* + \epsilon_s \right) \rho_{\text{m}}^2, \tag{S21}$$

where $\epsilon_h$ and $\epsilon_s$ are the enthalpic and entropic components, respectively, of the mean-field Flory-Huggins interaction for favorable non-electrostatic attraction, and $T^* = l/l_{\text{B}}$ is the reduced temperature. With this augmented rG-RPA+FH system free energy $f$ in hand, we solve the solute and solvent concentrations in dilute and condensed phases by balancing the chemical potentials of each solute component and their osmotic pressures in the two phases. When the salt concentration is assumed for simplicity to be uniform throughout the system (**Figure 1c and d** of the maintext), the system only has $\rho_{\text{m}}$ as a variable and the binodal phase separation concentrations are readily obtained by solving the standard common tangent conditions (**Lin et al., 2023**; **Lin et al., 2017b**). The Flory-Huggins parameters $\epsilon_h$ and $\epsilon_s$ fitted to the experimental data are $\epsilon_h = 1.0$, $\epsilon_s = 0.0$ for the Caprin1 phase diagrams in **Figure 1c** and $\epsilon_h = 1.0$, $\epsilon_s = -1.5$ for the pY-Caprin1 phase diagrams in

*Figure 1d*. When the uniform salt concentration restriction is removed to allow for fully varying salt and polymer concentrations (*Figure 2* of the maintext), the final concentrations in the two phases depend on their initial bulk (overall) concentrations. The corresponding two-dimensional, polymer-salt phase diagram (at a given temperature $T$) is obtained by similar balancing conditions (*Lin et al., 2017a*). As stated in the maintext, the two-dimensional rG-RPA+FH phase diagrams in *Figure 2* are for $T = 300$ K.

## rG-RPA-predicted effects of counterion valency on Caprin1 LLPS

As discussed in the maintext, with monovalent salt (Na$^+$), rG-RPA predicts that Caprin1 does not undergo LLPS at [Na$^+$] = 0 when the counterion (Cl$^-$) is monovalent (maintext *Figure 2a and b*), but Caprin1 LLPS is possible at [Na$^+$] = 0 when the counterion is divalent (maintext *Figure 2e and f*). As mentioned in the maintext, a likely physical reason for this effect is the difference in configurational entropy loss of monovalent vs divalent counterions in the Caprin1-condensed phase. Apparently, when [Cl$^-$] is just sufficient to balance the net positive charge of Caprin1 (i.e. when [Na$^+$] = 0), the entropic cost of concentrating Cl$^-$ in a Caprin1-condensed phase cannot be overcome for Caprin1 LLPS to occur. The entropic cost will be lessened (and thus more favorable to Caprin1 LLPS) when there are more Cl$^-$ ions beyond what is necessary to balance the net positive charge of Caprin1, corresponding to a situation with nonzero [Na$^+$] from the added NaCl to supply the additional Cl$^-$ ions. In comparison, when the counterion is divalent [(ATP-Mg)$^{2-}$ in our case], the number of counterions needed for balancing the positive net charge of Caprin1 is half of that when the counterion is monovalent. It follows that the entropic cost of concentrating the divalent counterion in the Caprin1-condensed phase is less and consequently, at least in the present situation, no added salt is needed for Caprin1 LLPS.

## Explicit-ion coarse-grained explicit-chain molecular dynamics (MD) simulation

Coarse-grained MD simulations are performed with the GPU version of HOOMD-Blue software (*Anderson et al., 2020*; *Anderson et al., 2008*) using the slab method that has been developed recently to allow for simulations of relatively large number of polymers (*Silmore et al., 2017*) and applied to liquid-liquid phase separation (LLPS) of intrinsically disordered proteins (IDPs) (*Dignon et al., 2018*). This general MD protocol has been utilized extensively, (*Joseph et al., 2021*; *Dignon et al., 2019b*) including by our group (*Das et al., 2020*; *Das et al., 2018a*).

Within the methodological framework of this coarse-grained simulation protocol, we introduce explicit small ions into our present simulations because they are necessary to account for subtle experimental observations that are not readily reproduced by using implicit-ion electrostatic screening. Simulations in the present study are performed with 100 chains of the Caprin1 IDR (wildtype or variants) at four salt concentrations ([NaCl]): (i) at [NaCl] = 0 where the system is neutralized by adding appropriate number (1300) of chloride (Cl$^-$) ions, (ii) neutralized and at [NaCl] = 200 mM by adding 15,000 pairs of explicit Na$^+$ and Cl$^-$ ions, (iii) neutralized and at [NaCl] = 480 mM by adding the same number of 15,000 pairs of explicit Na$^+$ and Cl$^-$ ions, and (iv) neutralized and at [NaCl] = 960 mM again with 15,000 pairs of explicit Na$^+$ and Cl$^-$ ions. As will be described below, specific small-ion concentrations in (ii), (iii), and (iv) are implemented by varying the size of the final simulation box. A similar procedure is used for simulation of pY-Caprin1 IDR phase behaviors under these four [NaCl] values. Because each pY-Caprin1 IDR has a net −1 charge, the only difference with the Caprin1 IDR simulation is that neutralization of the pY-Caprin1 chain requires 100 Na$^+$ ions instead of 1300 Cl$^-$ ions. The amino acid sequences simulated using coarse-grained MD in the present study are provided in *Appendix 1—figure 1*. Note that the experimental pY-Caprin1 sample is highly phosphorylated, consisting mainly of a mixture of Caprin1 IDRs with six or seven phosphorlations, with only a very small fraction of IDRs with five phosphorylations, and essentially no population with fewer than five phosphorylations (*Appendix 1—figure 2*). As stated in the maintext, for the sake of simplicity in our theoretical/computational models, we use only the Caprin1 IDR with all seven tyrosines phosphorylated (referred to simply as pY-Caprin1 in *Appendix 1—figure 1*) to model the behaviors of this experimental sample, partly to avoid the combinatoric complexity of sequences with five or six phosphorlations, which would entail 21 and 7 possible different sequences, respectively, with currently unknown population fractions.

## Coarse-grained MD interaction potentials

Following prior works (*Dignon et al., 2018*; *Das et al., 2020*), each amino acid residue is modeled by a single bead. Beads representing different amino acid residues have different masses, sizes, and engage in pairwise interactions with different strengths (*Dignon et al., 2018*; *Das et al., 2020*). Following the notations of our earlier simulation works (*Das et al., 2018b*; *Das et al., 2020*; *Das et al., 2018a*), we consider $n_p$ number of polymers labeled as $\mu$, $v = 1, 2,...n_p$, with each polymer consisting of $N$ beads labeled by $i, j = 1, 2, \ldots, N$, and $n_c$ counterions to neutralize the charged polymers. Coarse-grained MD is readily applicable to studying variations in LLPS properties among RtoK variants (*Das et al., 2020*) considered here for Caprin1 (*Appendix 1—figure 1*). In contrast, since RtoK substitutions do not change the sequence charge pattern of any given sequence, rG-RPA theory as formulated above does not account for their effects on LLPS, although polymer field theory can be extended to incorporate such effects in more sophisticated formulations (*Wessén et al., 2022a*).

For salt-dependent LLPS ($n_s \neq 0$), we consider $n_+$ small cations and $n_-$ small anions. These small cations and anions are classified as salt ions or counterions depending on the net charge of the polymer (see maintext as well as discussion below in this *Appendix 1*). The small ions are labeled, respectively, by $\gamma = 1, 2, \ldots, n_+$ and $\beta = 1, 2, \ldots, n_-$, and they correspond to Na$^+$ and Cl$^-$ in the present coarse-grained MD simulations. As stated in the maintext, our total MD potential energy is the sum of electrostatic (el), short-spatial-range (sr) Lennard-Jones (LJ)-type, and bonding (bond) interactions:

$$U_T = U_{el} + U_{sr} + U_{bond} . \tag{S22}$$

For our systems of interest, the electrostatic part is a sum of polymer-polymer (pp), polymer–small-ion (pi), and small-ion–small-ion (ii) contributions:

$$U_{el} = U_{el,pp} + U_{el,pi} + U_{el,ii} . \tag{S23}$$

As before (*Wessén et al., 2022a*), the polymer-polymer potential energy is given by

$$U_{el,pp} = \frac{e^2}{8\pi\epsilon_0\epsilon_r} \sum_{\mu,\nu=1}^{n_p} \sum_{i,j=1}^{N} \left(1 - \delta_{\mu\nu}\delta_{ij}\right) \frac{\sigma_i\sigma_j}{r_{\mu i,\nu j}} , \tag{S24}$$

where, as in the above field-theoretic formulation, $e$ is protonic charge, $\epsilon_0$ is vacuum permittivity, and $\epsilon_r$ is relative permittivity (dielectric constant). Here, $r_{\mu i,\nu j}$ is the spatial distance between the $i$th residue of the $\mu$th polymer and the $j$th residue of the $\nu$th polymer. The Kronecker symbol $\delta$ signals exclusion of the self-interacting $\mu = \nu$, $i = j$ terms in the summations (irrespective of the values of these terms) because $1 - \delta_{xy} = 0$ if $x = y$ and $1 - \delta_{xy} = 1$ otherwise. In units of the protonic charge $e$, $\sigma_i$ of unphosphorylated amino acid residue beads are taken from *Dignon et al., 2018*. Except lysine and arginine ($\sigma_i = +1$), glutamic and aspartic acid ($\sigma_i = -1$), histidine ($\sigma_i = +0.5$)—but note that there is no histidine in the Caprin1/pY-Caprin1 sequences simulated here, and phosphorylated tyrosine ($\sigma_i = -2$), all other amino acid residues are assigned zero charge. Similarly, the interaction between polymers and small ions is given by

$$
\begin{aligned}
U_{el,pi} &= \frac{e^2}{4\pi\epsilon_0\epsilon_r} \sum_{\mu=1}^{n_p} \sum_{i=1}^{N} \left\{ \sum_{k=+,-} \left[ \sum_{\gamma(k)=1}^{n_\gamma(k)} \frac{\sigma_i\sigma_k}{r_{\mu i,\gamma(k)}} \right] \right\} \\
&= \frac{e^2}{4\pi\epsilon_0\epsilon_r} \sum_{\mu=1}^{n_p} \sum_{i=1}^{N} \left[ \sum_{\gamma=1}^{n_+} \frac{\sigma_i\sigma_+}{r_{\mu i,\gamma}} + \sum_{\beta=1}^{n_-} \frac{\sigma_i\sigma_-}{r_{\mu i,\beta}} \right] ,
\end{aligned}
\tag{S25}
$$

where, in the term after the first equality, the summation $\sum_k$ (enclosed in curly brackets) is over small ion types, summation indices $\gamma(+)$ and $\gamma(-)$ label, respectively, the positively and negatively charged small ions, $n_\gamma(+)$ and $n_\gamma(-)$ are the total numbers of these ions, and $r_{\mu i,\gamma(k)}$ is the spatial distance between the $i$th residue of the $\mu$th polymer chain and the small ion labeled by $\gamma(k)$. After the second equality, the two terms in $\sum_k$ are written explicity, now with $r_{\mu i,\gamma/\beta}$ being the spatial distance between the $i$th residue of the $\mu$th polymer chain and the $\gamma/\beta$ ($\gamma$ or $\beta$)-labeled positive/negative small ion as well as $\sigma_+$ and $\sigma_-$ being the charges of the small positive and negative small ions, respectively. Following *Joung and Cheatham, 2008*, we take $\sigma_+ = +1$ for Na$^+$ and $\sigma_- = -1$ for Cl$^-$. Depending on the net charge of the polymer, counterions can be included in either the $n_+$ or $n_-$ count. For instance, for the positive charged Caprin1 IDR, the total number $n_-$ of negatively charged small ions (Cl$^-$) includes the numbers $n_c$ counted as counterions and $n_s$ counted as salt ions.

The interaction between the small ions is given by

$$U_{\text{el,ii}} = \frac{e^2}{8\pi\epsilon_0\epsilon_{\text{r}}} \sum_{k,k'=+,-} \left[ \sum_{\gamma(k)=1}^{n_\gamma(k)} \sum_{\gamma'(k')=1}^{n'_\gamma(k')} \left(1 - \delta_{kk'}\delta_{\gamma(k)\gamma'(k')}\right) \frac{\sigma_k\sigma_{k'}}{r_{\gamma(k),\gamma'(k')}} \right] , \tag{S26}$$

where $r_{\gamma(k),\gamma'(k')}$ is the spatial distance between two small ions. For the MD simulations in this work, the positively and negatively charged small ions correspond to Na$^+$ and Cl$^-$ respectively. **Equation S26** is equivalent to

$$U_{\text{el,ii}} = \frac{e^2}{4\pi\epsilon_0\epsilon_{\text{r}}} \left[ \sum_{\gamma=1}^{n_+-1} \sum_{\gamma'=\gamma+1}^{n_+} \frac{\sigma_+^2}{r_{\gamma\gamma'}} + \sum_{\beta=1}^{n_--1} \sum_{\beta'=\beta+1}^{n_-} \frac{\sigma_-^2}{r_{\beta\beta'}} + \sum_{\gamma=1}^{n_+} \sum_{\beta=1}^{n_-} \frac{\sigma_+\sigma_-}{r_{\gamma\beta}} \right] , \tag{S27}$$

where $r_{xy}$ is the spatial distance between a pair of small ions labeled by $x$ and $y$. As rationalized previously (**Das et al., 2020**) in the context of experimental measurements of dielectric properties of biological systems (**Tros et al., 2017**), we use $\epsilon_{\text{r}} = 40$, a value lower than the ~80 dielectric constant of bulk water, for all simulations reported in the present work.

Short-spatial-range non-bonded LJ interactions are similarly constituted by three components, viz., those for polymer-polymer ($U_{\text{sr,pp}}$), polymer–small-ion ($U_{\text{sr,pi}}$), and small-ion–small-ion ($U_{\text{sr,ii}}$) interactions:

$$U_{\text{sr}} = U_{\text{sr,pp}} + U_{\text{sr,pi}} + U_{\text{sr,ii}} . \tag{S28}$$

Here we adopt the Kim-Hummer (KH) (**Kim and Hummer, 2008**) interaction scheme for $U_{\text{sr,pp}}$. KH is based on the Miyazawa-Jernigan (MJ) statistical potential (**Miyazawa and Jernigan, 1996**) derived from folded globular protein structures in the Protein Data Bank (PDB) and is therefore expected to reflect the energetics of polypeptides, especially the driving forces pertinent to protein folding, its limitations (**Das et al., 2020**) notwithstanding. Our previous work shows that the KH potential is adequate for rationalizing the rank ordering of LLPS propensities of the N-terminal IDR of DEAD-box RNA helicase Ddx4 and its charge scrambled and arginine-to-lysine (RtoK) variants. KH also rationalizes the rank ordering of LLPS propensities of WT and an RtoK variant of LAF-1 (**Das et al., 2020**). We therefore stipulate that the KH interaction scheme is appropriate, at least as a first approximation, to address the LLPS propensities of Caprin1 and its RtoK variants. The degree to which the differences in simulated LLPS propensity among these Caprin1 variants are affected by how interactions involving K and R are treated differently by the model potential function (**Das et al., 2020**; **Joseph et al., 2021**; **Tesei et al., 2021**) should be further explored in the future. As before, $U_{\text{sr,pp}}$ takes the following form (**Das et al., 2020**; **Dignon et al., 2018**):

$$\begin{aligned} U_{\text{sr,pp}} &= \frac{1}{2} \sum_{\mu,\nu=1}^{n_p} \sum_{i,j=1}^{N} \left(1 - \delta_{\mu\nu}\delta_{ij}\right) (U_{\text{KH}})_{\mu i,\nu j} , \\ (U_{\text{KH}})_{\mu i,\nu j} &= U_{\text{LJ}} + (1 - \lambda_{ij}^{\text{KH}})\,\epsilon_{ij} \quad \text{if } r \le 2^{1/6}\,a_{ij} \\ &= \lambda_{ij}^{\text{KH}} U_{\text{LJ}} \qquad\qquad\quad \text{otherwise} \end{aligned} \tag{S29}$$

in which

$$(U_{\text{LJ}})_{\mu i,\nu j} = U_{\text{LJ}}(\epsilon_{\mu i,\nu j}, a_{\mu i,\nu j}, r_{\mu i,\nu j}) = 4\epsilon_{\mu i,\nu j} \left[ \left(\frac{a_{\mu i,\nu j}}{r_{\mu i,\nu j}}\right)^{12} - \left(\frac{a_{\mu i,\nu j}}{r_{\mu i,\nu j}}\right)^{6} \right] \tag{S30}$$

and $a_{\mu i,\nu j} = a_{ij} = (a_i + a_j)/2$, where the van der Waals diameters $a_i$ and $a_j$, depend only, respectively, on the amino acid residue type (one of twenty) for residue $i$ and residue $j$ (Table S1 of **Dignon et al., 2018**). In contrast, the parameters $\lambda_{ij}^{\text{KH}}$ and $\epsilon_{\mu i,\nu j} = \epsilon_{ij}$ depend on both the residue types of residues $i$ and $j$. Values for $\epsilon_{ij}$ are provided in Table S3 of **Dignon et al., 2018**. The formula for $\lambda_{ij}^{\text{KH}} = \pm 1$ is given by Equation 5 of **Dignon et al., 2018** as well as Equations S10 and S11 of **Das et al., 2020**.

For the LJ interactions between polymers and small ions, we recognize that while the coarse-grained KH parameters are based on statistical analysis of known folded protein structures, LJ interaction parameters for small ions are typically scaled to match certain physical and chemical properties (**Joung and Cheatham, 2008**). Thus, it is not straightforward to postulate an interaction scheme based upon first principles. To make progress and to maintain simplicity of our model, we adopt the LJ form [$U_{\text{LJ}}$ in **Equation S30**] for $U_{\text{sr,pi}}$ with a uniform $\epsilon_{\mu i,\nu j} = \epsilon_{ij} = 0.142$ (denoted

$\epsilon_{\mathrm{p}\pm} \equiv 0.142$) for all residue-small ion pairs. This $\epsilon_{ij} = \epsilon_{\mathrm{p}\pm}$ value is equal to that for a pair of alanine residues in the KH potential and is neither too strong nor too weak among $\epsilon_{ij}$ values for pairwise interactions between amino acid residues. Accordingly,

$$U_{\mathrm{sr,pi}} = \sum_{\mu=1}^{n_{\mathrm{p}}} \sum_{i=1}^{N} \left[ \sum_{\gamma=1}^{n_+} U_{\mathrm{LJ}}(\epsilon_{\mathrm{p}\pm}, a_{i+}, r_{\mu i,\gamma}) + \sum_{\beta=1}^{n_-} U_{\mathrm{LJ}}(\epsilon_{\mathrm{p}\pm}, a_{i-}, r_{\mu i,\beta}) \right] , \tag{S31}$$

where $a_{i+} = (a_i + a_+)/2$, $a_{i-} = (a_i + a_-)/2$, with $a_+$ and $a_-$ being, respectively, the van der Waals diameters of the positively and negatively charged small ions. For the present MD simulations, $a_+ = a_{\mathrm{Na}^+}$, $a_- = a_{\mathrm{Cl}^-}$, and their values are adopted from *Joung and Cheatham, 2008*. Similarly, for small-ion–small-ion LJ interactions,

$$U_{\mathrm{el,ii}} = \left[ \sum_{\gamma=1}^{n_+-1} \sum_{\gamma'=\gamma+1}^{n_+} U_{\mathrm{LJ}}(\epsilon_+, a_+, r_{\gamma,\gamma'}) + \sum_{\beta=1}^{n_--1} \sum_{\beta'=\beta+1}^{n_-} U_{\mathrm{LJ}}(\epsilon_-, a_-, r_{\beta,\beta'}) \right.$$
$$\left. + \sum_{\gamma=1}^{n_+} \sum_{\beta=1}^{n_-} U_{\mathrm{LJ}}(\epsilon_{+-}, a_{+-}, r_{\gamma,\beta}) \right] , \tag{S32}$$

where $\epsilon_+$ and $\epsilon_-$ are, respectively, the LJ interaction energy parameter for the positively and negatively charged small ions, $\epsilon_{+-} = (\epsilon_+\epsilon_-)^{1/2}$, and $a_{+-} = (a_+ + a_-)/2$. For the present MD simulations, the $\epsilon_+ = \epsilon_{\mathrm{Na}^+}$ and $\epsilon_- = \epsilon_{\mathrm{Cl}^-}$ values are also adopted from *Joung and Cheatham, 2008*.

As before, the bond-length energy term $U_{\mathrm{bond}}$ in *Equation S22* for chain connectivity is modeled by a harmonic potential,

$$U_{\mathrm{bond}} = \frac{K_{\mathrm{bond}}}{2} \sum_{\mu=1}^{n_{\mathrm{p}}} \sum_{i=1}^{N-1} (r_{\mu i,\nu i+1} - l)^2 . \tag{S33}$$

Following previous studies (*Das et al., 2020*; *Dignon et al., 2018*), Kuhn length $l = 3.8$ Å is taken as the Cα–Cα virtual bond length for *trans* polypeptides and $K_{\mathrm{bond}} = 10$ kJ mol$^{-1}$ Å$^{-2}$ .

## Simulation protocol

In each of our coarse-grained MD simulations, the IDR chains and ions are initially placed randomly in a sufficiently large cubic box of dimensions 300×300×300 Å$^3$. Energy minimization is then performed using the FIRE algorithm (available in the HOOMD-Blue package) which includes removal of steric clashes among the initially placed amino acid beads. Next, the system is compressed at a low temperature of 100 K at 1 atm pressure for a period of 50 ns using the Martyna-Tobias-Klein (MTK) thermostat and barostat (*Martyna et al., 1994*; *Tuckerman et al., 2006*) with a coupling constant of 1 ps. The equations of motion are integrated using velocity-Verlet algorithm with a timestep of 20 fs. Periodic boundary conditions are applied in all three directions. The electrostatic interaction is computed using the PPPM algorithm (*LeBard et al., 2012*) available in the package. We use a cut-off distance of 15 Å for the short-spatial-range non-bonded interactions. After this initial *NPT* step, we compress the simulation box again along the three dimensions for a period of 50 ns until it reaches a sufficiently high density, using Langevin dynamics for an *NVT* ensemble with a friction coefficient of $1\,\mathrm{ps}^{-1}$. At the end of this compression step, the dimensions of the simulation box for Caprin1 and its four RtoK variants are 115×115×115 Å$^3$ for [NaCl] = 0 (no small ions beside counterions) and 155×155×155 Å$^3$ for [NaCl] = 200 mM, 480 mM, and 960 mM. For pY-Caprin1, the corresponding dimensions are 115×115×115 Å$^3$ for [NaCl] = 0 (no small ions beside counterions) and 155×155×155 Å$^3$ for [NaCl] = 200 mM, 480 mM, and 960 mM. Next, the system is expanded along one of the spatial dimensions (taken as the *z*-axis) using isotropic linear scaling for 10 ns while keeping the temperature constant at 100 K. For Caprin1 and its four RtoK variants, the simulation box length in the *z*-direction is expanded 14 times for [NaCl] = 0, 33.6 times for [NaCl] = 200 mM, 14 times for [NaCl] = 480 mM, and 7 times for [NaCl] = 960 mM. For pY-Caprin1, the expansion factors along the *z*-direction are 10 times for [NaCl] = 0, 37.07 times for [NaCl] = 200 mM, 15.47 times for [NaCl] = 480 mM, and 7.73 times for [NaCl] = 960 mM. Note that the simulation box volumes for Caprin1, its RtoK variants, and pY-Caprin1 after this last expansion are identical for the same [NaCl] because the same numbers of polymer chains and small ions are used. The practical reason for keeping the number of Na$^+$ and Cl$^-$ ions constant for the higher salt concentration is to minimize computational cost. In other words, the three salt concentrations (200 mM, 480 mM and 960 mM) are achieved here by using different box dimensions. After the last box expansion, *NVT* equilibration using the Langevin thermostat with a friction coefficient of 1 ps$^{-1}$ is performed for 2 µs at various temperatures.

Final production run is then carried out for another 4 µs with the same Langevin thermostat using a much lower friction coefficient of 0.01 ps$^{-1}$ for sampling efficiency. The snapshots are saved every 1 ns for further analysis. Detailed descriptions of how to construct a phase diagram from simulation trajectories are provided in *Dignon et al., 2018* and our previous works (*Das et al., 2020*; *Lin et al., 2023*). This simulation protocol and the above-described coarse-grained MD model are used to produce the phase diagrams, distributions, and snapshots in *Figures 4–7* of the maintext and *Appendix 1—figure 3*.

## Comparison with atomic simulations with a preformed condensate

As mentioned in the maintext, explicit-water, explicit-ion atomic simulations in the presence of a preformed condensate of the N-terminal RGG domain of LAF-1 with a net charge of +4 produce enhanced Cl$^-$ and depleted Na$^+$ in the IDR-condensed phase (*Zheng et al., 2020*). This trend is consistent with our implicit-water, explicit-ion MD result for Caprin1 with net charge +13 (maintext *Figure 4c*). By comparison, corresponding atomic simulations in the presence of a preformed condensate of the low complexity domain of FUS with a net charge of −2 produce a significant depletion of Cl$^-$ and a minor depletion of Na$^+$ in the IDR-condensed phase (*Zheng et al., 2020*), which is opposite to the trend seen here for pY-Caprin1 with net charge −1 in maintext *Figure 4d* and *Appendix 1—figure 3*. Whether this difference is caused by the multiple phosphorylated sites with a −2 charge in pY-Caprin1 remains to be elucidated.

## Field-theoretic simulation (FTS) for multiple-component LLPS

Biomolecular condensates in vivo can contain hundreds of protein and nucleic acid species. Therefore, to address their biophysical properties and biochemical functions, theories—starting with rudimentary constructions—are needed for multiple-component LLPS. As a first appproximation and a tool for conceptualization, we find it valuable to utilize FTS—especially recently developed FTS approaches for biomolecular LLPS (*McCarty et al., 2019*; *Pal et al., 2021*; *Fredrickson et al., 2002*; *Fredrickson, 2006*) and their extensions—to gain insights into the energetic basis of sequence-specific spatial distributions of various biomolecular components in and out of phase-separated condensates.

FTS enjoys the fundamental advantage that it is not limited by some of the approximations in analytical theories such as RPA and rG-RPA because FTS accounts fully for all field fluctuations in principle. FTS is thus a valuable alternative to analytical theories, though it is computationally more costly in practice and can be impeded by lattice-related artifacts and limitations arising from the small FTS simulation box sizes necessitated by numerical tractability. We view analytical theories and FTS as complementary.

The starting point of FTS is a statistical field theory [e.g., *Equation S5*, which is equivalent to maintext *Equation 5*]. To avoid numerical instabilities, we treat polymer beads and ions as (smeared) Gaussian distributions (*Wang, 2010*) instead of the point particles stipulated by the Dirac δ-functions in *Equations S2 and S3*. For simplicity, this regularization is implemented using a common component-independent width $\bar{a}$ irrespective of chemical species by making the general replacements $\delta(\mathbf{r} - \mathbf{r}_a) \rightarrow \Gamma(\mathbf{r} - \mathbf{r}_a)$ and $\delta(\mathbf{r} - \mathbf{R}_{\alpha,i}) \rightarrow \Gamma(\mathbf{r} - \mathbf{R}_{\alpha,i})$ in *Equations S2 and S3* where $\Gamma(\mathbf{r}) = e^{-r^2/2\bar{a}^2}/(2\pi\bar{a}^2)^{3/2}$, $r^2 = |\mathbf{r}|^2$.

A general field-theoretic Hamiltonian applicable to a system comprising of one or more charged polymer species including Caprin1, pY-Caprin1, (ATP-Mg)$^{2-}$ (maintext *Figure 8a*), ATP$^{4-}$ and small ions such as Na$^+$, Cl$^-$, and Mg$^{2+}$ is given by

$$H[w, \psi] = \int d\mathbf{r} \left( \frac{(\nabla\psi(\mathbf{r}))^2}{8\pi l_{\mathrm{B}}} + \frac{w(\mathbf{r})^2}{2v_2} \right) - \sum_m n_m \ln \mathcal{Q}_m[\breve{w}, \breve{\psi}], \tag{S34}$$

where the $\mathcal{Q}_m$ functionals ($m$ labels system components) are in general complex when evaluated beyond quadratic order in the fields. *Equation S34* above is identical to *Equation 5* of maintext and formally equivalent to the Hamiltonian in *Equation S5*. Because of the above-described Gaussian smearing, the fields in the arguments of $\mathcal{Q}_m$ are now convoluted with $\Gamma$, i.e. $\phi(\mathbf{r}) \rightarrow \breve{\phi}(\mathbf{r}) = \Gamma \star \phi(\mathbf{r}) \equiv \int d\mathbf{r}' \Gamma(\mathbf{r} - \mathbf{r}')\phi(\mathbf{r}')$, where the generic $\phi = w, \psi$.

## Complex Langevin evolution in fictitious time

A simulation approach developed in the 1980s to handle the complex nature of $H[w, \psi]$ and obtain statistical (Boltzmann) averages is the Complex Langevin (CL) method (*Parisi, 1983*; *Klauder, 1983*), which analytically continues the fields $w$ and $\psi$ into their respective complex planes and introduces a fictitious (artificial, unphysical) time-coordinate $t$ on which $w(\mathbf{r}, t)$ and $\psi(\mathbf{r}, t)$ now depend. The CL time evolution is governed by stochastic Langevin differential equations [maintext *Equation 6*], as follows:

$$\frac{\partial \phi(\mathbf{r}, t)}{\partial t} = -\frac{\delta H}{\delta \phi(\mathbf{r}, t)} + \eta_\phi(\mathbf{r}, t), \quad \phi = w, \psi, \tag{S35}$$

where $\eta_\phi(\mathbf{r}, t)$ is real-valued Gaussian noise with zero mean:

$$\langle \eta_\phi(\mathbf{r}, t) \eta_{\phi'}(\mathbf{r}', t') \rangle = 2\delta_{\phi, \phi'} \delta(\mathbf{r} - \mathbf{r}') \delta(t - t'). \tag{S36}$$

Thermal averages of any thermodynamic observable $\hat{\mathcal{O}}[\mathbf{R}, \mathbf{r}]$ can then be computed in the field picture (indicated by '$\langle \ldots \rangle_\mathrm{F}$' with subscript 'F') using a corresponding field operator $\tilde{\mathcal{O}}[w, \psi]$ through averages over all possible equilibrium field configurations, which in turn translate into asymptotic CL time averages with no final dependence on the fictitious time variable $t$, that is

$$\langle \tilde{\mathcal{O}}[w, \psi] \rangle_\mathrm{F} \equiv \frac{\int \mathscr{D}w \int \mathscr{D}\psi \, \tilde{\mathcal{O}}[w, \psi] \, e^{-H[w, \psi]}}{\int \mathscr{D}w \int \mathscr{D}\psi \, e^{-H[w, \psi]}} = \lim_{t_\mathrm{max} \to \infty} \frac{1}{t_\mathrm{max}} \int_0^{t_\mathrm{max}} dt \, \tilde{\mathcal{O}}[w(\mathbf{r}, t), \psi(\mathbf{r}, t)]. \tag{S37}$$

The Langevin *Equation S35* involves functional derivatives of the Hamiltonian with respect to the complex fields, which are formally evaluated as

$$\frac{\delta H}{\delta w(\mathbf{r})} = \mathrm{i} \sum_m \tilde{\rho}_m(\mathbf{r}) + \frac{1}{v_2} w(\mathbf{r}), \tag{S38a}$$

$$\frac{\delta H}{\delta \psi(\mathbf{r})} = \mathrm{i} \sum_m \tilde{c}_m(\mathbf{r}) - \frac{1}{4\pi l_\mathrm{B}} \nabla^2 \psi(\mathbf{r}) \tag{S38b}$$

where

$$\tilde{\rho}_m(\mathbf{r}) = \mathrm{i} n_m \frac{\delta \ln \mathcal{Q}_m[\breve{w}, \breve{\psi}]}{\delta w(\mathbf{r})}, \tag{S39a}$$

$$\tilde{c}_m(\mathbf{r}) = \mathrm{i} n_m \frac{\delta \ln \mathcal{Q}_m[\breve{w}, \breve{\psi}]}{\delta \psi(\mathbf{r})} \tag{S39b}$$

are field operators corresponding, respectively, to number- and charge density of chemical component $m$.

## Number density correlation functions

Information about the polymer-polymer, polymer-ion, ion-ion association and ion partitioning into the condensate can be gleaned from number density-number density correlation functions [maintext *Equation 7*]

$$G_{m,n}(|\mathbf{r} - \mathbf{r}'|) = \langle \hat{\rho}_m(\mathbf{r}) \hat{\rho}_n(\mathbf{r}') \rangle, \tag{S40}$$

which can be computed in field theory (*Pal et al., 2021*) as

$$G_{m,n \neq m}(|\mathbf{r} - \mathbf{r}'|) = \langle \tilde{\rho}_m(\mathbf{r}) \tilde{\rho}_n(\mathbf{r}') \rangle_\mathrm{F}, \tag{S41a}$$

$$G_{m,m}(|\mathbf{r} - \mathbf{r}'|) = \frac{\mathrm{i}}{v_2} \langle \tilde{\rho}_m(\mathbf{r}) w(\mathbf{r}') \rangle_\mathrm{F} - \sum_{n \neq m} \langle \tilde{\rho}_m(\mathbf{r}) \tilde{\rho}_n(\mathbf{r}') \rangle_\mathrm{F}. \tag{S41b}$$

The $G_{m,n}$ functions are useful for assessing Caprin1 and pY-Caprin1 phase separation and the colocalization of ATP-Mg with the polymer condensed droplet (maintext *Figure 8b–e*). Information

with higher spatial resolution can also be provided by $G_{m,n}$ if we identify component $m$ with individual polymer bead (labeled by $i$) along a chain sequence.

In some situations, the physical implications of density-density correlation functions $G_{m,n}(r)$ are more apparent when normalized by the component bulk (overall) densities $\rho_m^0$ and $\rho_n^0$, as discussed in the maintext in connection with the correlation functions shown in *Figure 8*.

For small ions that are each represented by a single Gaussian distribution, the density operator is given by

$$\tilde{\rho}_m(\mathbf{r}) = \frac{n_m}{\Omega \mathcal{Q}_k} \Gamma \star e^{-\mathrm{i}[\breve{w}(\mathbf{r}) + z_m \breve{\psi}(\mathbf{r})]}, \tag{S42}$$

and the charge density operator is $\tilde{c}_m(\mathbf{r}) = z_m \tilde{\rho}_m(\mathbf{r})$, where $z_m$ is the charge of ion species $m$ and, as defined above, $\Omega$ is the system volume. For polymers (denoted 'p'), the density and charge-density operators are calculated using forward (subscript '$F$') and backward (subscript '$B$') chain propagators $q_F(\mathbf{r}, i)$ and $q_B(\mathbf{r}, i)$ as follows, with $i$ being the label for the beads/monomers along the polymer chain:

$$\mathcal{Q}_{\mathrm{p}} = \frac{1}{\Omega} \int d\mathbf{r}\, q_F(\mathbf{r}, N)\,, \tag{S43a}$$

$$\tilde{\rho}_{\mathrm{p}}(\mathbf{r}) = \frac{n_{\mathrm{p}}}{\Omega \mathcal{Q}_{\mathrm{p}}} \Gamma \star \sum_{i=1}^{N} q_F(\mathbf{r}, i) q_B(\mathbf{r}, i) e^{\mathrm{i}[\breve{w}(\mathbf{r}) + \sigma_i \breve{\psi}(\mathbf{r})]}, \tag{S43b}$$

$$\tilde{c}(\mathbf{r}) = \frac{n_{\mathrm{p}}}{\Omega \mathcal{Q}_{\mathrm{p}}} \Gamma \star \sum_{i=1}^{N} q_F(\mathbf{r}, i) q_B(\mathbf{r}, i) e^{\mathrm{i}[\breve{w}(\mathbf{r}) + \sigma_i \breve{\psi}(\mathbf{r})]} \sigma_i, \tag{S43c}$$

and the chain propagators are constructed iteratively as

$$q_F(\mathbf{r}, i+1) = e^{-\mathrm{i}[\breve{w}(\mathbf{r}) + \sigma_{i+1} \breve{\psi}(\mathbf{r})]} \left(\frac{3}{2\pi b^2}\right)^{3/2} \int d\mathbf{r}' e^{-3(\mathbf{r}-\mathbf{r}')^2/2b^2} q_F(\mathbf{r}', i)\,, \tag{S44a}$$

$$q_B(\mathbf{r}, i-1) = e^{-\mathrm{i}[\breve{w}(\mathbf{r}) + \sigma_{i-1} \breve{\psi}(\mathbf{r})]} \left(\frac{3}{2\pi b^2}\right)^{3/2} \int d\mathbf{r}' e^{-3(\mathbf{r}-\mathbf{r}')^2/2b^2} q_B(\mathbf{r}', i) \tag{S44b}$$

with the starting $q_F(\mathbf{r}, 1) = e^{-\mathrm{i}[\breve{w}(\mathbf{r}) + \sigma_1 \breve{\psi}(\mathbf{r})]}$, $q_B(\mathbf{r}, N) = e^{-\mathrm{i}[\breve{w}(\mathbf{r}) + \sigma_N \breve{\psi}(\mathbf{r})]}$, and we use $b$ for Kuhn length ($b = l$) in the present FTS formulation to conform to the notation in our published FTS studies (*Pal et al., 2021*; *Wessén et al., 2022a*; *Lin et al., 2023*). In the present work, the correlation functions in maintext *Figure 8* and *Appendix 1—figure 4*; *Appendix 1—figure 5*; *Appendix 1—figure 6* are computed by integrating pertinent CL fictitious-time evolution equations defined in *Equation S35* using the first order semi-implicit method of *Lennon et al., 2008*.

## Residue-specific Caprin1–(ATP-Mg) association

As outlined in *Materials and methods* of the maintext, residue-specific properties of the polymers in our FTS systems can be gleaned from the $G_{m,n}$ function in *Equation S40* by identifying $m$ as individual polymer beads (indexed by $i$) along the polymer chain sequence, viz., define $G_{\mathrm{pq}}^{(i)}(|\mathbf{r} - \mathbf{r}'|) \equiv \langle \hat{\rho}_{\mathrm{p},i}(\mathbf{r}) \hat{\rho}_{\mathrm{q}}(\mathbf{r}') \rangle$ where $\hat{\rho}_{\mathrm{p},i}(\mathbf{r}) \equiv \sum_{\alpha=1}^{n_{\mathrm{p}}} \Gamma(\mathbf{r} - \mathbf{R}_{\alpha,i})$, the corresponding operator $\tilde{\rho}_{\mathrm{p},i}(\mathbf{r})$ being equal to the $i$th term in the summation in *Equation S43b*, and q is another component in the FTS system. Accordingly [maintext *Equation 9*],

$$\mathcal{G}_{\mathrm{pq}}^{(i)} \equiv 4\pi \int_0^{r_{\mathrm{contact}}} dr\, r^2 G_{\mathrm{pq}}^{(i)}(r) \tag{S45}$$

with a reasonably small residue-q distance $r_{\mathrm{contact}}$ (spatial separation between residue $i$ and the positions of particles belonging to component q) can be used to represent residue-specific relative residue-q contact frequencies. A normalized version of this quantity is defined by

$$\frac{\mathcal{G}_{\mathrm{pq}}^{(i)}}{\rho_{\mathrm{p},i}^0 \rho_{\mathrm{q}}^0} = 4\pi \int_0^{r_{\mathrm{contact}}} dr\, r^2 g_{\mathrm{pq}}^{(i)}(r)\,, \quad g_{\mathrm{pq}}^{(i)}(r) \equiv \frac{G_{\mathrm{pq}}^{(i)}(r)}{\rho_{\mathrm{p},i}^0 \rho_{\mathrm{q}}^0}\,, \tag{S46}$$

where $\rho_{\mathrm{q}}^0$ and $\rho_{\mathrm{p},i}^0$ are the bulk (overall) densities, respectively, of the q-component and the $i$th residue along the given polymer species. Values of $\mathcal{G}_{\mathrm{pq}}^{(i)}/\rho_{\mathrm{p},i}^0\rho_{\mathrm{q}}^0$ in the above *Equation S46* for p = Caprin1 and q = (ATP-Mg)$^{2-}$, Na$^+$, or Cl$^-$ under the simulation conditions we considered are provided in maintext *Figure 8f*. The variation in $\mathcal{G}_{\mathrm{pq}}^{(i)}/\rho_{\mathrm{p},i}^0\rho_{\mathrm{q}}^0$ for (ATP-Mg)$^{2-}$ with residue position $i$ is largely consistent with the experimental trend of NMR-measured volume ratios on Caprin1-ATP association in *Kim et al., 2021*.

In all the FTS simulations in this study except for a part of the model with no (ATP-Mg)$^{2-}$ described immediately below, we use a cubic simulation box of length $L = N_{\mathrm{L}}\Delta x$ where $N_{\mathrm{L}} = 32$ (i.e., a 32×32×32 lattice) and $\Delta x = b/\sqrt{6}$ is the lattice resolution. In view of the periodic boundary conditions implemented for all three spatial dimensions, the maximum possible physical distance between two volume elements is $\sqrt{3}L/2$. All possible physical distances $r = r_{i,j,k}$ on this cubic simulation box satisfy the relation $r_{i,j,k}^2 = [\min\{i, N_{\mathrm{L}} - i\}^2 + \min\{j, N_{\mathrm{L}} - j\}^2 + \min\{k, N_{\mathrm{L}} - k\}^2]\Delta x^2$, for some $i, j, k = 0, 1, 2, \ldots, 31$. There is thus a finite number of discretized distances between 0 and $\sqrt{3}L/2$. One of these discretized distances, $r = 1.47b \approx 1.50b$ is used for the $\int_0^{r_{\mathrm{contact}}} dr$ integrations in maintext *Equation 9* and *Equations S45 and S46* above.

Further details of the main FTS model utilized for the results in maintext *Figures 8 and 9* as well as alternate FTS models discussed under the heading 'Robustness of general trends predicted by field-theoretic simulation' in the maintext are provided below in ascending order of number of components treated by the model:

## FTS models of Caprin1/pY-Caprin1 with Na$^+$ and Cl$^-$ but no ATP-Mg

FTS is conducted for Caprin1 and pY-Caprin1 at various concentrations of explicit Na$^+$ and Cl$^-$ ions. For all systems, polymer bead concentration is fixed at $n_{\mathrm{p}}N/\Omega = 0.4b^{-3}$. The salt concentrations, here referring to the concentration of the small ion species with the same sign of charge as the net charge of the polymer, are set to $[\mathrm{NaCl}]\,b^3 = 0, 10^{-6}, 10^{-5}, \ldots 10^{-1}, 10^0$. Additional small ions of charge opposite to the polymer net charge are added to achieve overall electric neutrality of the system. For the results in *Appendix 1—figure 4e–h*, simulations are performed in an elongated simulation box on a 24×24×80 lattice with lattice spacing set to the Gaussian smearing length, $\Delta x = \bar{a} = b/\sqrt{6}$. The Complex-Langevin (CL) evolution equations (*Equation S35*) are integrated using a time-step $\Delta t = 10^{-3}b^3$ for $6 \times 10^4$ steps and the system is sampled every 50th step. An equilibration period of $3 \times 10^4$ CL steps, determined by monitoring the equilibration of the chemical potentials for each molecular species, is discarded from each trajectory. Eight independent simulations are run for each combination of salt concentration and Caprin1 or pY-Caprin1. All simulations are performed at $l_{\mathrm{B}} = 7b$ and $v_2 = 0.0068b^3$. The density profiles shown in *Appendix 1—figure 4e and f* are obtained by averaging the real part of the field theoretic polymer bead density operator over the $x$ and $y$ dimensions (i.e., the dimensions corresponding to the short sides of the simulation box). The resulting one-dimensional density snapshots were then individually centred around their center-of-mass $z$-coordinate $z_{\mathrm{c.o.m.}}$ before taking the trajectory average to give the profiles in *Appendix 1—figure 4e and f*. The shaded uncertainty bands in these plots indicate the root-mean-squared difference among the eight independent runs. The density profiles are then used to estimate the coexisting condensed and dilute phases shown in *Appendix 1—figure 4g and h*. Here, the condensed-phase concentration is obtained as the average density at $z_{\mathrm{c.o.m.}}$, whereas the dilute-phase concentrations are estimated as the average density among the 10 and 60 $z$-coordinates (for Caprin1 and pY-Caprin1, respectively) furthest from $z_{\mathrm{c.o.m.}}$. Consistent with rG-RPA and explicit-chain MD, these FTS models show reentrant behavior—albeit subtle—for Caprin1 at low [protein]s (an LLPS region is seen in *Appendix 1—figure 4g* at intermediate [NaCl] but not at higher or lower [NaCl]) but not for pY-Caprin1 (no such feature in *Appendix 1—figure 4h*). The elongated box field-theoretic simulations were performed using the BioFTS Python package which is publicly available at https://github.com/jwessen/BioFTS (*Wessén, 2025*) under an MIT open-source license.

## FTS models for Caprin1/pY-Caprin1 with (ATP-Mg)$^{2-}$ and either Na$^+$ or Cl$^-$

Simulations are performed at $l_{\mathrm{B}} = 5b$ on a periodic 32×32×32 grid (see above) with CL time step $\Delta t = 0.002$. All ATP$^{4-}$s and Mg$^{2+}$s are assumed to be in the complex (ATP-Mg)$^{2-}$ form with charge sequence $(-1\,-1\,-1\,-1\,+1\,+1)$ as depicted in maintext *Figure 8a*. Bulk Caprin1 and pY-Caprin1 bead densities in their respective simulation systems are both set at $0.1b^{-3}$. For Caprin1, depending on the concentration of (ATP-Mg)$^{2-}$, counterions Na$^+$ or Cl$^-$ (but not both) are added to maintain overall electric neutrality of the FTS system. For pY-Caprin1, Na$^+$ is added as counterions to maintain overall

electric neutrality. Results from this set of models are provided in *Appendix 1—figure 5*. The bands representing sampling uncertainties in the correlation function plots in maintext *Figure 8b–e* and *Appendix 1—figure 5* (top) and *Appendix 1—figure 6a, b* are standard deviations across eight independent simulations. If we take the model Kuhn length $b$ in FTS as the Cα-Cα virtual bond length ≈ 3.8 Å of polypeptides, a unit bead concentration of $b^{-3}$ is equivalent to ≈30 M. Because $(ATP-Mg)^{2-}$ is modeled by six beads (*Figure 8a* of maintext), a model bead concentration of $(ATP-Mg)^{2-}$ (denoted as $[(ATP-Mg)^{2-}]$ in our FTS results) which is equal to $b^{-3}$ reported for the present FTS results, is equivalent to a molar concentration of ≈5 M of $(ATP-Mg)^{2-}$. As mentioned above, since excluded volume is often significantly underestimated in FTS (*Pal et al., 2021*), we do not directly compare FTS model $(ATP-Mg)^{2-}$ concentrations with experimental $(ATP-Mg)^{2-}$ concentrations, which tend to be substantially lower. Instead, physical insights are gleaned from the trend of variation of model concentrations.

## FTS models for Caprin1/pY-Caprin1 with $(ATP-Mg)^{2-}$, $Na^+$ and $Cl^-$

We use this set of models for the results in maintext *Figure 8* and *Figure 9*. Simulations are performed at $l_B = 7b$ on a periodic 32×32×32 grid (see above) with CL time step $\Delta t = 0.005$. Again, all $ATP^{4-}$s and $Mg^{2+}$s are assumed to be in the complex $(ATP-Mg)^{2-}$ form with charge sequence (−1 −1 −1 −1 +1 +1) as depicted in maintext *Figure 8a*. Bulk Caprin1 and pY-Caprin1 bead densities in their respective simulation systems are both set at $0.1b^{-3}$. Three concentrations of $(ATP-Mg)^{2-}$ are studied: $[(ATP-Mg)^{2-}] = 0.0001b^{-3}$, $0.03b^{-3}$, and $0.5b^{-3}$. With overall electric neutrality of the simulation system in mind, for Caprin1 (WT), the bulk (overall) densities for $Na^+$ and $Cl^-$ are $[Na^+]$ = $4[(ATP-Mg)^{2-}]/6$ and $[Cl^-] = 2[(ATP-Mg)^{2-}]/6 + 13[Caprin1]/103$. For pY-Caprin1, $[Na^+] = 4[(ATP-Mg)^{2-}]/6 + [pY-Caprin1]/103$ and $[Cl^-] = 2[(ATP-Mg)^{2-}]/6$.

## FTS models for Caprin1/pY-Caprin1 with $ATP^{4-}$, $Mg^{2+}$, $Na^+$ and $Cl^-$

In contrast to the above models, here we consider $ATP^{4-}$ and $Mg^{2+}$ as independent components. That is, they can freely dissociate if the favorable electric interaction between them is insufficiently strong. In this set of models, $ATP^{4-}$ is taken as a four-bead charge sequence (−1 −1 −1 −1) whereas $Mg^{2+}$ is modeled by a single bead with charge 2+ [instead of the two −1 beads in the $(ATP-Mg)^{2-}$ model in maintext *Figure 8a*]. As before, bulk (overall) Caprin1 and pY-Caprin1 bead densities in their respective simulation systems are both set at $0.1b^{-3}$, and the same three $[(ATP-Mg)^{2-}] = 0.0001b^{-3}$, $0.03b^{-3}$, and $0.5b^{-3}$ are studied. For Caprin1 (WT), the bulk (overall) densities for $Mg^{2+}$, $ATP^{4-}$, $Na^+$ and $Cl^-$ are given by $[Mg^{2+}] = [ATP^{4-}]/4$, $[Na^+] = [ATP^{4-}]$, and $[Cl^-]=2[Mg^{2+}]+13[Caprin1]/103$. For pY-Caprin1, $[Mg^{2+}] = [ATP^{4-}]/4$, $[Na^+] = [ATP^{4-}] + [pY-Caprin1]/103$, and $[Cl^-] = 2[Mg^{2+}]$. Results from this set of FTS models are provided in *Appendix 1—figure 6*.

WT:
SRGVSRGGSRGARGLMNGYRGPANGFRGGYDGYRPSFSNTPNSGYTQSQFSAPRDYSGYQRD
GYQQNFKRGSGQSGPRGAPRGRGGPPRPNRGMPQMNTQQVN

4Rto4K$_N$:
SKGVSKGGSKGAKGLMNGYRGPANGFRGGYDGYRPSFSNTPNSGYTQSQFSAPRDYSGYQR
DGYQQNFKRGSGQSGPRGAPRGRGGPPRPNRGMPQMNTQQVN

4Rto4K$_M$:
SRGVSRGGSRGARGLMNGYRGPANGFKGGYDGYKPSFSNTPNSGYTQSQFSAPKDYSGYQK
DGYQQNFKRGSGQSGPRGAPRGRGGPPRPNRGMPQMNTQQVN

4Rto4K$_C$:
SRGVSRGGSRGARGLMNGYRGPANGFRGGYDGYRPSFSNTPNSGYTQSQFSAPRDYSGYQRD
GYQQNFKKGSGQSGPKGAPKGKGGPPRPNRGMPQMNTQQVN

15Rto15K:
SKGVSKGGSKGAKGLMNGYKGPANGFKGGYDGYKPSFSNTPNSGYTQSQFSAPKDYSGYQK
DGYQQNFKKGSGQSGPKGAPKGKGGPPKPNKGMPQMNTQQVN

pY Caprin1:
SRGVSRGGSRGARGLMNGpYRGPANGFRGGpYDGpYRPSFSNTPNSGpYTQSQFSAPRDpYSG
pYQRDGpYQQNFKRGSGQSGPRGAPRGRGGPPRPNRGMPQMNTQQVN

**Appendix 1—figure 1.** Sequences of wildtype (WT) and variant Caprin1 IDRs studied in this work. Positively charged arginine (R) and lysine (K) residues are shown, respectively, in dark and light blue, negatively charged aspartic acid (D) residues and phosphorylated tyrosines (pY) are shown in red. Other residues are in black.

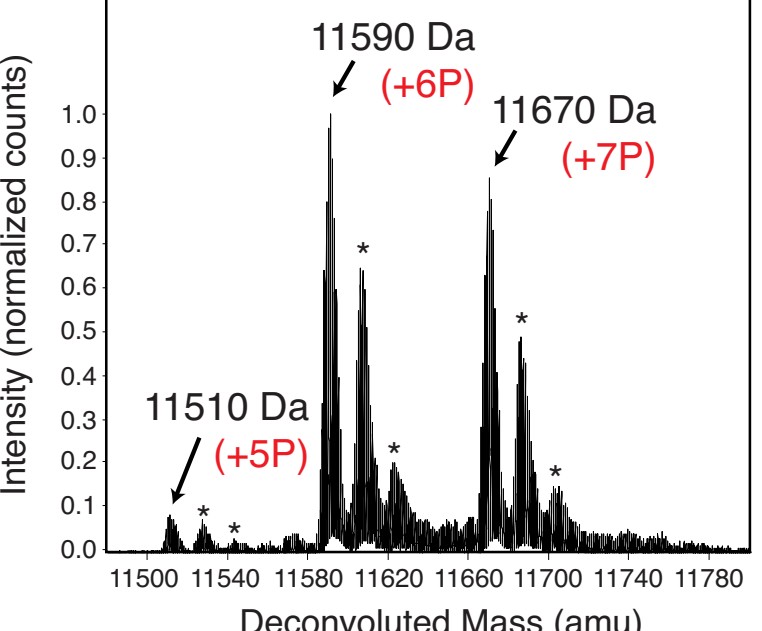

**Appendix 1—figure 2.** Mass spectrometry analysis of pY-Caprin1. The graph plots deconvoluted mass (in atomic mass units, amu) on the horizontal axis against intensity (normalized counts) on the vertical axis. Peaks are observed at 11,510 Da (+5 phosphate groups,+5 P), 11,590 Da (+6 P), and 11,670 Da (+7 P). Asterisks mark the peaks of pY-Caprin1 with oxidized methionine residues.

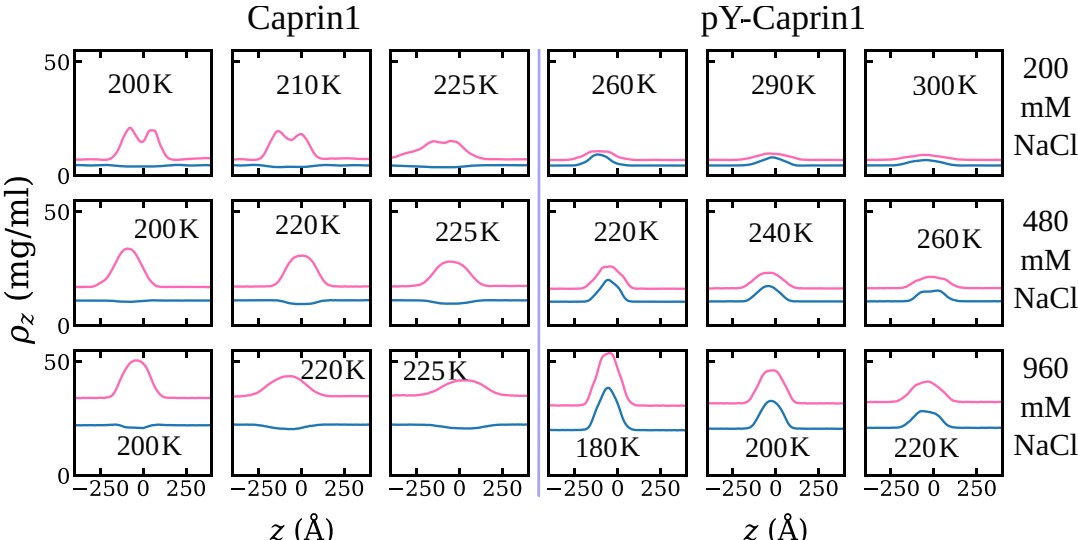

**Appendix 1—figure 3.** Explicit-ion coarse-grained molecular dynamics simulation of salt and counterion mass density profiles in protein-dilute and protein-condensed phases of Caprin1 and pY-Caprin1. Mass density profile $\rho_z$ for $Na^+$ (blue) and $Cl^-$ (red) in units of mg/ml for the Caprin1 (three left columns) and pY-Caprin1 (three right columns) systems are shown as in *Figure 4e and f* of the maintext. Regions with elevated $[Cl^-]$ here coincide with positions of the condensed protein droplets. Overall [NaCl] values used for the simulations are provided on the right.

The online version of this article includes the following source data for appendix 1—figure 3:

**Appendix 1—figure 3—source data 1.** Numerical data for the coarse-grained molecular dynamics-simulated results plotted in *Appendix 1—figure 3*.

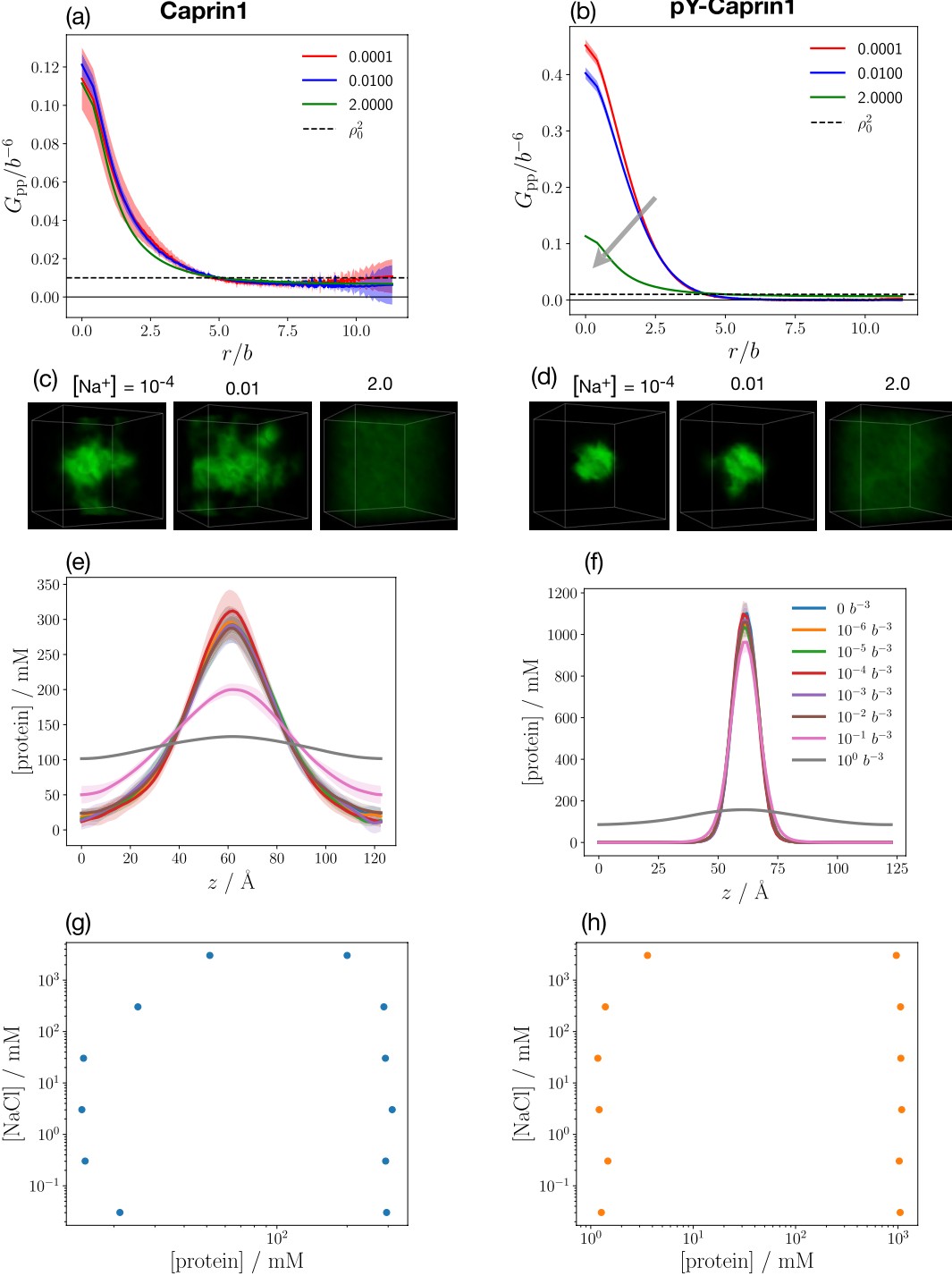

**Appendix 1—figure 4.** FTS models for Caprin1 and pY-Caprin1 with only Na$^+$ and Cl$^-$ but no ATP-Mg. (**a, b**) Protein-protein correlation functions [maintext *Equation 7*] for Caprin1 (**a**) and pY-Caprin1 (**b**) at three different [Na$^+$]s, color coded in units of $b^{-3}$ as provided. In each figure, the baseline value of protein-protein correlation function $(\rho_p^0)^2$ (square of overall protein concentration) is marked by the horizontal dashed line. Phase separation is indicated by large-$r$ correlation function values falling below this baseline. The grey arrow in (**b**) marks the direction of increasing [Na$^+$]. (**c, d**) Field snapshots for the Caprin1 (**c**) and pY-Caprin1 (**d**) systems at different [Na$^+$] values. The above results are obtained at Bjerrum length $l_B = 7b$. (**e–h**) Results from an alternate FTS model using an elongated simulation box similar to that utilized for our explicit-ion coarse-grained MD. (**e, f**) Protein concentration profiles computed at different NaCl concentrations for Caprin1 (**e**) and pY-Caprin1 (**f**) [color code for density profiles provided in **f**]. (**g, h**) salt-protein phase diagrams obtained from the concentration profiles in (**e, f**) for Caprin1 (**g**) and pY-Caprin1 (**h**).

The online version of this article includes the following source data for appendix 1—figure 4:

**Appendix 1—figure 4—source data 1.** Numerical data for the theoretical results plotted in *Appendix 1—figure 4a, b, e–h*.

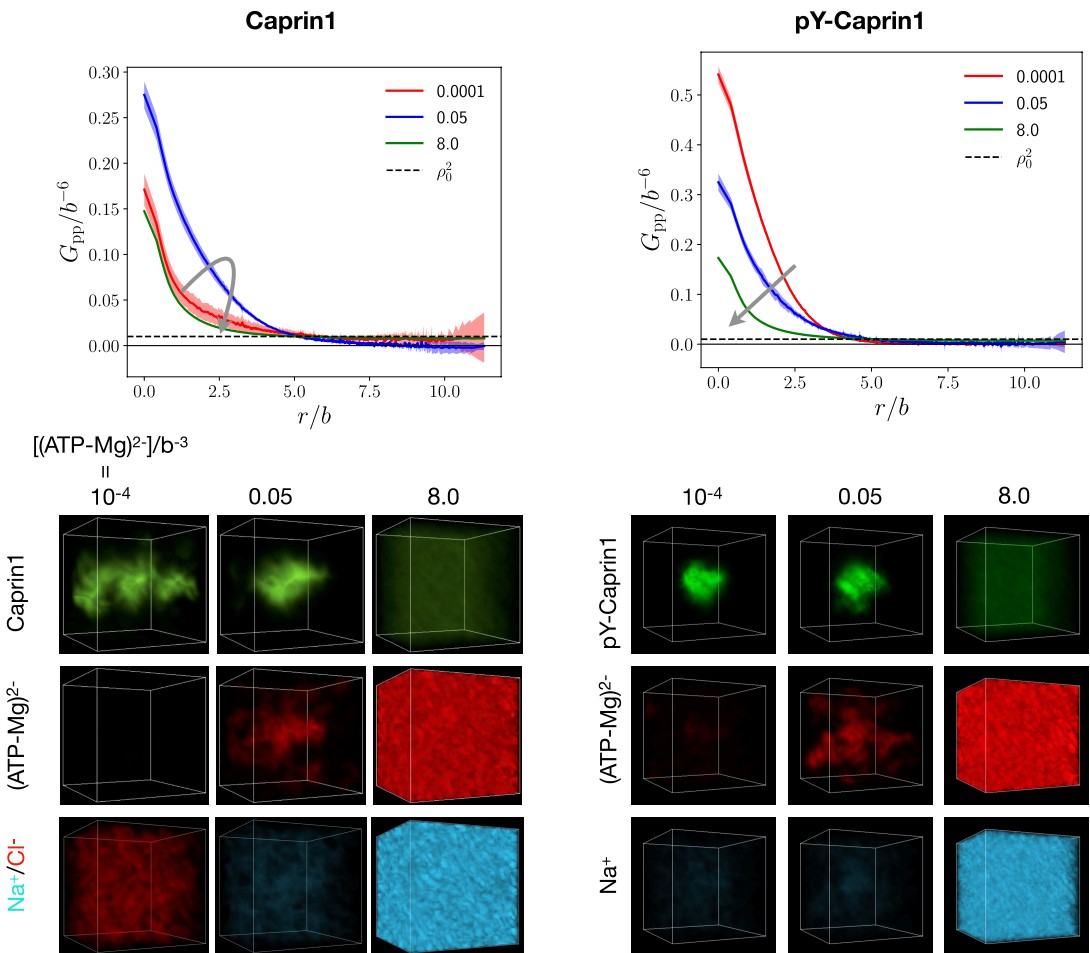

**Appendix 1—figure 5.** Alternate FTS models for Caprin1 and pY-Caprin1 with $(ATP\text{-}Mg)^{2-}$ and either $Na^+$ or $Cl^-$ (but not both) to maintain overall electric neutrality. For Caprin1, which has a net positive charge, $(ATP\text{-}Mg)^{2-}$ is the counterion. Depending on $[(ATP\text{-}Mg)^{2-}]$, either $Cl^-$ is included as an additional counterion (when $[(ATP\text{-}Mg)^{2-}]$ is insufficient to balance the positive charges on Caprin1), or $Na^+$ is included as salt ion (when $[(ATP\text{-}Mg)^{2-}]$ overcompensates the positive charges on Caprin1). $Na^+$ and $Cl^-$ are not included together in this simplified formulation. For pY-Caprin1, which has a net negative charge, $Na^+$ is used as counterion, and its concentration depends on $[(ATP\text{-}Mg)^{2-}]$ in such a way that electric neutrality of the entire system is maintained. Top panels: protein-protein correlation functions at different concentrations of $(ATP\text{-}Mg)^{2-}$ (color coded in units of $b^{-3}$ as provided). Horizontal dashed lines are $(\rho_p^0)^2$ baselines as in *Appendix 1—figure 4a, b*. Grey arrows indicate increasing $[(ATP\text{-}Mg)^{2-}]$. Bottom panels: Field snapshots for system components at different $[(ATP\text{-}Mg)^{2-}]$ (as indicated) for the Caprin1 (left panels) and pY-Caprin1 (right panels) systems. Results here are obtained at $l_B = 7b$.

The online version of this article includes the following source data for appendix 1—figure 5:

**Appendix 1—figure 5—source data 1.** Numerical data for the theoretical results plotted in *Appendix 1—figure 5a and b*.

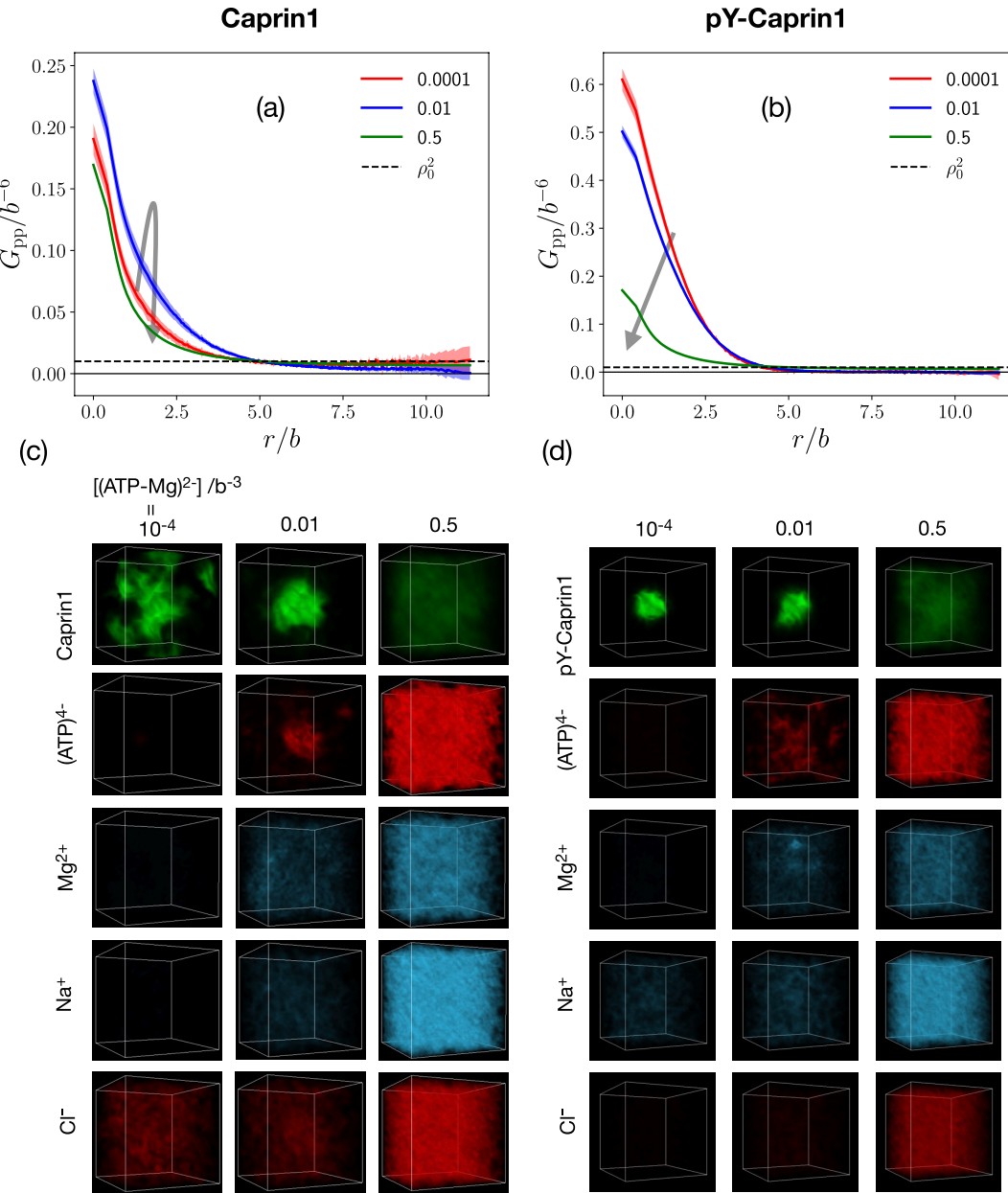

**Appendix 1—figure 6.** Alternate FTS models for Caprin1 or pY-Caprin1 with $ATP^{4-}$, $Mg^{2+}$, $Na^+$ and $Cl^-$, wherein $(ATP-Mg)^{2-}$ is assumed to be fully dissociable. Results are obtained for $l_B = 7b$ and presented in the same style as that in *Figure 8b, c* and *Figure 9* of the maintext as well as *Appendix 1—figure 5*.

The online version of this article includes the following source data for appendix 1—figure 6:

**Appendix 1—figure 6—source data 1.** Numerical data for the theoretical results plotted in *Appendix 1—figure 6a and b*.

