## [Editor Report · eLife Assessment]

In this **important** study, the authors employed three types of theoretical/computational models (coarse-grained molecular dynamics, analytical theory and field-theoretical simulations) to analyze the impact of salt on protein liquid-liquid phase separation. These different models reinforce each other and together provide **convincing** evidence to explain distinct salt effects on ATP mediated phase separation of different variants of caprin1. The insights and general approach are broadly applicable to the analysis of protein phase separation. Still, modeling at the coarse-grained level misses key effects that have been revealed by all-atom simulations, including salt-backbone coordination and strengthening of pi-type interactions by salt.

---

## [Referee Report · Reviewer #1 (Public review)]

Summary:

The authors used multiple approaches to study salt effects in liquid-liquid phase separation (LLPS). Results on both wild-type Caprin1 and mutants and on different types of salts contribute to a comprehensive understanding.

Strengths:

The main strength of this work is the thoroughness of investigation. This aspect is highlighted by the multiple approaches used in the study, and reinforced by the multiple protein variants and different salts studied.

Weaknesses:

(1) The multiple computational approaches are a strength, but they're cruder than explicit-solvent all-atom molecular dynamics (MD) simulations and may miss subtle effects of salts. In particular, all-atom MD simulations demonstrate that high salt strengthens pi-types of interactions (ref. 42 and MacAinsh et al, https://www.biorxiv.org/content/10.1101/2024.05.26.596000v3).

(2) The paper can be improved by distilling the various results into a simple set of conclusions. By example, based on salt effects revealed by all-atom MD simulations, MacAinsh et al. presented a sequence-based predictor for classes of salt dependence. Wild-type Caprin1 fits right into the "high net charge" class, with a high net charge and a high aromatic content, showing no LLPS at 0 NaCl and an increasing tendency of LLPS with increasing NaCl. In contrast, pY-Caprin1 belongs to the "screening" class, with a high level of charged residues and showing a decreasing tendency of LLLPS.

(3) Mechanistic interpretations can be further simplified or clarified. (i) Reentrant salt effects (e.g., Fig. 4a) are reported but no simple explanation seems to have been provided. Fig. 4a,b look very similar to what has been reported as strong-attraction promotor and weak-attraction suppressor, respectively (ref. 50; see also PMC5928213 Fig. 2d,b). According to the latter two studies, the "reentrant" behavior of a strong-attraction promotor, CL- in the present case, is due to Cl-mediated attraction at low to medium [NaCl] and repulsion between Cl- ions at high salt. Do the authors agree with this explanation? If not, could they provide another simple physical explanation? (ii) The authors attributed the promotional effect of Cl- to counterion-bridged interchain contacts, based on a single instance. There is another simple explanation, i.e., neutralization of the net charge on Caprin1. The authors should analyze their simulation results to distinguish net charge neutralization and interchain bridging; see MacAinsh et al.

(4) The authors presented ATP-Mg both as a single ion and as two separate ions; there is no explanation of which of the two versions reflects reality. When presenting ATP-Mg as a single ion, it's as though it forms a salt with Na^+^. I assume NaCl, ATP, and MgCl2 were used in the experiment. Why is Cl^-^ not considered? Related to this point, it looks ATP is just another salt ion studied and much of the Results section is on NaCl, so the emphasis of ATP "Diverse Roles of ATP" in the title is somewhat misleading.

Comments on revisions:

This revision addressed all my previous comments.

---

## [Referee Report · Reviewer #2 (Public review)]

Summary:

In this paper, Lin and colleagues aim to understand the role of different salts on the phase behavior of a model protein of significant biological interest, Caprin1, and its phosphorylated variant, pY-Caprin1. To achieve this, the authors employed a variety of methods to complement experimental studies and obtain a molecular-level understanding of ion partitioning inside biomolecular condensates. A simple theory based on rG-RPA is shown to capture the different salt dependencies of Caprin1 and pY-Caprin1 phase separation, demonstrating excellent agreement with experimental results. The application of this theory to multivalent ions reveals many interesting features with the help of multicomponent phase diagrams. Additionally, the use of CG model-based MD simulations and FTS provides further clarity on how counterions can stabilize condensed phases.

Strengths:

The greatest strength of this study lies in the integration of various methods to obtain complementary information on thermodynamic phase diagrams and the molecular details of the phase separation process. The authors have also extended their previously proposed theoretical approaches, which should be of significant interest to other researchers. Some of the findings reported in this paper, such as bridging interactions, are likely to inspire new studies using higher-resolution atomistic MD simulations.

---

## [Referee Report · Reviewer #3 (Public review)]

Authors first use rG-RPA to reproduce two observed trends. Caprin1 does not phase separate at very low salt but then undergoes LLPS with added salt while further addition of salt reduces its propensity to LLPS. On the other hand pY-Caprin1 exhibits a monotonic trend where the propensity to phase separate decreases with the addition of salt. This distinction is captured by a two component model and also when salt ions are explicitly modeled as a separate species with a ternary phase diagram. The predicted ternary diagrams (when co and counter ions are explicitly accounted for) also predict the tendency of ions to co-condense or exclude proteins in the dense phase. Predicted trends are generally in line with the measurement for Cparin1. Next, the authors seek to explain the observed difference in phase separation when Arginines are replaced by Lysines creating different variants. In the current rG-RPA type models both Arginine (R) and Lysine (K) are treated equally since non-electrostatic effects are only modeled in a mean-field manner that can be fitted but not predicted. For this reason, coarse grain MD simulation is suitable. Moreover, MD simulation affords structural features of the condensates. They used a force field that is capable of discriminating R and K. The MD predicted degrees of LLPS of these variants again is consistent with the measurement. One additional insight emerges from MD simulations that a negative ion can form a bridge between two positively charged residues on the chain. These insights are not possible to derive from rG-RPA. Both rG-RPA and MD simulation become cumbersome when considering multiple types of ions such as Na, Cl, [ATP] and [ATP-Mg] all present at the same time. FTS is well suited to handle this complexity. FTS also provides insights into the co-localization of ions and proteins that is consistent with NMR. By using different combinations of ions they confirm the robustness of the prediction that Caprin1 shows salt-dependent reentrant behavior, adding further support that the differential behavior of Caprin1, and pY-Caprin1 is likely to be mediated by charge-charge interactions.

Comments on revisions:

The authors addressed my comments and it is ready for publication.

---

## [Author Response]

The following is the authors’ response to the current reviews.

**Reviewer #1 (Public Review):**
Comments on revisions:This revision addressed all my previous comments.
**Reviewer #3 (Public Review):**
Comments on revisions:The authors addressed my comments and it is ready for publication.

We are grateful for the reviewers’ effort and are encouraged by their generally positive assessment of our manuscript.

**Reviewer #1 (Recommendations For The Authors):**
This revision addressed all my previous comments. The only new issue concerns the authors’ response to the following comment of reviewer 3:(2) Authors note ”monovalent positive salt ions such as Na+ can be attracted, somewhat counterintuitively, into biomolecular condensates scaffolded by positively-charged polyelectrolytic IDRs in the presence of divalent counterions”. This may be due to the fact that the divalent negative counterions present in the dense phase (as seen in the ternary phase diagrams) also recruit a small amount of Na+.Author reply: The reviewer’s comment is valid, as a physical explanation for this prediction is called for. Accordingly, the following sentence is added to p. 10, lines 27-29:Here are my comments on this issue. Most IDPs with a net positive charge still have negatively charged residues, which in theory can bind cations. In fact, Caprin1 has 3 negatively charged residues (same as A1-LCD). All-atom simulations of MacAinsh et al (ref 72) have shown that these negatively charged residues bind Na+; I assume this effect can be captured by the coarsegrained models in the present study. Moreover, all-atom simulations showed that Na+ has a strong tendency to be coordinated by backbone carbonyls, which of course are present on all residues. Suggestions:(a) The authors may want to analyze the binding partners of Na+. Are they predominantly the3 negatively charged residues, or divalent counterions, or both?(b) The authors may want to discuss the potential underestimation of Na+ inside Caprin1 condensates due to the lack of explicit backbone carbonyls that can coordinate Na+ in their models. A similar problem applies to backbone amides that can coordinate anions, but to a lesser extent (see Fig. 3A of ref 72).

The reviewer’s comments are well taken. Regarding the statement in the revised manuscript “This phenomenon arises because the positively charge monovalent salt ions are attracted to the negatively charged divalent counterions in the protein-condensed phase.”, it should be first noted that the statement was inferred from the model observation that Na+ is depleted in condensed Caprin1 (Fig. 2a) when the counterion is monovalent (an observation that was stated almost immediately preceding the quoted statement). To make this logical connection clearer as well as to address the reviewer’s point about the presence of negatively charged residues in Caprin1, we have modified this statement in the Version of Record (VOR) as follows:

“This phenomenon most likely arises from the attraction of the positively charge monovalent salt ions to the negatively charged divalent counterions in the proteincondensed phase because although the three negatively charged D residues in Caprin1 can attract Na+, it is notable that Na+ is depleted in condensed Caprin1 when the counterion is monovalent (Fig. 2a).”

The reviewer’s suggestion (a) of collecting statistics of Na+ interactions in the Caprin1 condensate is valuable and should be attempted in future studies since it is beyond the scope of the present work. Thus far, our coarse-grained molecular dynamics has considered only monovalent Cl− counterions. We do not have simulation data for divalent counterions.

Following the reviewer’s suggestion (b), we have now added the following sentence in Discussion under the subheading “Effects of salt on biomolecular LLPS”:

“In this regard, it should be noted that positively and negatively charged salt ions can also coordinate with backbone carbonyls and amides, respectively, in addition to coordinating with charged amino acid sidechains (MacAinsh et al., eLife 2024). The impact of such effects, which are not considered in the present coarse-grained models, should be ascertained by further investigations using atomic simulations (MacAinsh et al., eLife 2024; Rauscher & Pom`es, eLife 2017; Zheng et al., J Phys Chem B 2020).”

Here we have added a reference to Rauscher & Pom`es, eLife 2017 to more accurately reflect progress made in atomic simulations of biomolecular condensates.

More generally, regarding the reviewer’s comments on the merits of coarse-grained versus atomic approaches, we re-emphasize, as stated in our paper, that these approaches are complementary. Atomic approaches undoubtedly afford structurally and energetically high-resolution information. However, as it stands, simulations of the assembly-disassembly process of biomolecular condensate are nonideal because of difficulties in achieving equilibration even for a small model system with < 10 protein chains (MacAinsh et al., eLife 2024) although well-equilibrated simulations are possible for a reasonably-sized system with ∼ 30 chains when the main focus is on the condensed phase (Rauscher & Pom`es, eLife 2017). In this context, coarse-grained models are valuable for assessing the energetic role of salt ions in the thermodynamic stability of biomolecular condensates of physically reasonable sizes under equilibrium conditions.

In addition to the above minor additions, we have also added citations in the VOR to two highly relevant recent papers: Posey et al., J Am Chem Soc 2024 for salt-dependent biomolecular condensation (mentioned in Dicussion under subheadings “Tielines in protein-salt phase diagrams” and “Counterion valency” together with added references to Hribar et al., J Am Chem Soc 2002 and Nostro & Ninham, Chem Rev 2012 for the Hofmeister phenomena discussed by Posey et al.) and Zhu et al., J Mol Cell Biol 2024 for ATP-modulated reentrant behavior (mentioned in Introduction). We have also added back a reference to our previous work Lin et al., J Mol Liq 2017 to provide more background information for our formulation.

**Reviewer #2 (Recommendations For The Authors):**
The authors have done a great job addressing previous comments.

We thank this reviewer for his/her effort and are encouraged by the positive assessment of our revised manuscript.

---

The following is the authors’ response to the original reviews.

**Reviewer #1 (Public Review):**
Summary:The authors used multiple approaches to study salt effects in liquid-liquid phase separation (LLPS). Results on both wild-type Caprin1 and mutants and on different types of salts contribute to a comprehensive understanding.Strengths:The main strength of this work is the thoroughness of investigation. This aspect is highlighted by the multiple approaches used in the study, and reinforced by the multiple protein variants and different salts studied.

We are encouraged by this positive overall assessment.

Weaknesses: (1) The multiple computational approaches are a strength, but they’re cruder than explicit-solvent all-atom molecular dynamics (MD) simulations and may miss subtle effects of salts. In particular, all-atom MD simulations demonstrate that high salt strengthens pi-types of interactions (ref. 42 and MacAinsh et al, https://www.biorxiv.org/content/10.1101/2024.05.26.596000v3).

The relative strengths and limitations of coarse-grained vs all-atom simulation are now more prominently discussed beginning at the bottom of p. 5 through the first 8 lines of p. 6 of the revised manuscript (page numbers throughout this letter refer to those in the submitted pdf file of the revised manuscript), with MacAinsh et al. included in this added discussion (cited as ref. 72 in the revised manuscript). The fact that coarse-grained simulation may not provide insights into more subtle structural and energetic effects afforded by all-atom simulations with regard to *π*-related interaction is now further emphasized on p. 11 (lines 23–30), with reference to MacAinsh et al. as well as original ref. 42 (Krainer et al., now ref. 50 in the revised manuscript).

(2) The paper can be improved by distilling the various results into a simple set of conclusions. By example, based on salt effects revealed by all-atom MD simulations, MacAinsh et al. presented a sequence-based predictor for classes of salt dependence. Wild-type Caprin1 fits right into the “high net charg”e class, with a high net charge and a high aromatic content, showing no LLPS at 0 NaCl and an increasing tendency of LLPS with increasing NaCl. In contrast, pY-Caprin1 belongs to the “screening” class, with a high level of charged residues and showing a decreasing tendency of LLPS.

This is a helpful suggestion. We have now added a subsection with heading “Overview of key observations from complementary approaches” at the beginning of the “Results” section on p. 6 (lines 18–37) and the first line of p. 7. In the same vein, a few concise sentences to summarize our key results are added to the first paragraph of “Discussion” (p. 18, lines 23– 26). In particular, the relationship of Caprin1 and pY-Caprin1 with the recent classification by MacAinsh et al. (ref. 72) in terms of “high net charge” and “screening” classes is now also stated, as suggested by this reviewer, on p. 18 under “Discussion” (lines 26–30).

(3) Mechanistic interpretations can be further simplified or clarified. (i) Reentrant salt effects (e.g., Fig. 4a) are reported but no simple explanation seems to have been provided. Fig. 4a,b look very similar to what has been reported as strong-attraction promotor and weak-attraction suppressor, respectively (ref. 50; see also PMC5928213 Fig. 2d,b). According to the latter two studies, the “reentrant” behavior of a strong-attraction promotor, CL- in the present case, is due to Cl-mediated attraction at low to medium [NaCl] and repulsion between Cl- ions at high salt. Do the authors agree with this explanation? If not, could they provide another simple physical explanation? (ii) The authors attributed the promotional effect of Cl- to counterionbridged interchain contacts, based on a single instance. There is another simple explanation, i.e., neutralization of the net charge on Caprin1. The authors should analyze their simulation results to distinguish net charge neutralization and interchain bridging; see MacAinsh et al.

The relationship of Cl− in bridging and neutralizing configurations, respectively, with the classification of “strong-attraction promoter” and “weak-attraction suppressor” by Zhou and coworkers is now stated on p. 13 (lines 29–31), with reference to original ref. 50 by Ghosh, Mazarakos & Zhou (now ref. 59 in the revised manuscript) as well as the earlier patchy particle model study PMC5928213 by Nguemaha & Zhou, now cited as ref. 58 in the revised manuscript. After receiving this referee report, we have conducted an extensive survey of our coarse-grained MD data to provide a quantitative description of the prevalence of counterion (Cl−) bridging interactions linking positively charged arginines (Arg+s) on different Caprin1 chains in the condensed phase (using the [Na+] = 0 case as an example). The newly compiled data is reported under a new subsection heading “Explicit-ion MD offers insights into counterion-mediated interchain bridging interactions among condensed Caprin1 molecules” on p. 12 (last five lines)–p. 14 (first 10 lines) [∼ 1_._5 additional page] as well as a new Fig. 6 to depict the statistics of various Arg+–Cl−–Arg+ configurations, with the conclusion that a vast majority (at least 87%) of Cl− counterions in the Caprin1-condensed phase engage in favorable condensation-driving interchain bridging interactions.

(4) The authors presented ATP-Mg both as a single ion and as two separate ions; there is no explanation of which of the two versions reflects reality. When presenting ATP-Mg as a single ion, it’s as though it forms a salt with Na+. I assume NaCl, ATP, and MgCl2 were used in the experiment. Why is Cl- not considered? Related to this point, it looks like ATP is just another salt ion studied and much of the Results section is on NaCl, so the emphasis of ATP “Diverse Roles of ATP” in the title is somewhat misleading.

We model ATP and ATP-Mg both as single-bead ions (in rG-RPA) and also as structurally more realistic short multiple-bead polymers (in field-theoretic simulation, FTS). We have now added discussions to clarify our modeling rationale in using and comparing different models for ATP and ATP-Mg, as follows:

p. 8 (lines 19–36):

“The complementary nature of our multiple methodologies allows us to focus sharply on the electrostatic aspects of hydrolysis-independent role of ATP in biomolecular condensation by comparing ATP’s effects with those of simple salt. Here, Caprin1 and pY-Caprin1 are modeled minimally as heteropolymers of charged and neutral beads in rG-RPA and FTS. ATP and ATP-Mg are modeled as simple salts (singlebead ions) in rG-RPA whereas they are modeled with more structural complexity as short charged polymers (multiple-bead chains) in FTS, though the latter models are still highly coarse-grained. Despite this modeling difference, rG-RPA and FTS both rationalize experimentally observed ATP- and NaCl-modulated reentrant LLPS of Caprin1 and a lack of a similar reentrance for pY-Caprin1 as well as a prominent colocalization of ATP with the Caprin1 condensate. Consistently, the same contrasting trends in the effect of NaCl on Caprin1 and pY-Caprin1 are also seen in our coarse-grained MD simulations, though polymer field theories tend to overestimate LLPS propensity [99]. The robustness of the theoretical trends across different modeling platforms underscores electrostatics as a significant component in the diverse roles of ATP in the context of its well-documented ability to modulate biomolecular LLPS via hydrophobic and *π*-related effects [63, 65, 67].”

Here, the last sentence quoted above addresses this reviewer’s question about our intended meaning in referring to “diverse roles of ATP” in the title of our paper. To make this point even clearer, we have also added the following sentence to the Abstract (p. 2, lines 12–13):

*“...* The electrostatic nature of these features complements ATP’s involvement in *π*-related interactions and as an amphiphilic hydrotrope, *...”*

Moreover, to enhance readability, we have now added pointers in the rG-RPA part of our paper to anticipate the structurally more complex ATP and ATP-Mg models to be introduced subsequently in the FTS part, as follows:

p. 9 (lines 13–15):

“As mentioned above, in the present rG-RPA formulation, (ATP-Mg)^2−^ and ATP^4−^ are modeled minimally as a single-bead ion. They are represented by charged polymer models with more structural complexity in the FTS models below.”

p. 11 (lines 8–11):

These observations from analytical theory will be corroborated by FTS below with the introduction of structurally more realistic models of (ATP-Mg) ^2−^, ATP^4−^ together with the possibility of simultaneous inclusion of Na^+^, Cl−, and Mg^2+^ in the FTS models of Caprin1/pY-Caprin1 LLPS systems.

**Reviewer #2 (Public Review):**
Summary:In this paper, Lin and colleagues aim to understand the role of different salts on the phase behavior of a model protein of significant biological interest, Caprin1, and its phosphorylated variant, pY-Caprin1. To achieve this, the authors employed a variety of methods to complement experimental studies and obtain a molecular-level understanding of ion partitioning inside biomolecular condensates. A simple theory based on rG-RPA is shown to capture the different salt dependencies of Caprin1 and pY-Caprin1 phase separation, demonstrating excellent agreement with experimental results. The application of this theory to multivalent ions reveals many interesting features with the help of multicomponent phase diagrams. Additionally, the use of CG model-based MD simulations and FTS provides further clarity on how counterions can stabilize condensed phases.Strengths:The greatest strength of this study lies in the integration of various methods to obtain complementary information on thermodynamic phase diagrams and the molecular details of the phase separation process. The authors have also extended their previously proposed theoretical approaches, which should be of significant interest to other researchers. Some of the findings reported in this paper, such as bridging interactions, are likely to inspire new studies using higher-resolution atomistic MD simulations.Weaknesses:The paper does not have any major issues.

We are very encouraged by this reviewer’s positive assessment of our work.

**Reviewer #3 (Public Review):**
Authors first use rG-RPA to reproduce two observed trends. Caprin1 does not phase separate at very low salt but then undergoes LLPS with added salt while further addition of salt reduces its propensity to LLPS. On the other hand pY-Caprin1 exhibits a monotonic trend where the propensity to phase separate decreases with the addition of salt. This distinction is captured by a two component model and also when salt ions are explicitly modeled as a separate species with a ternary phase diagram. The predicted ternary diagrams (when co and counter ions are explicitly accounted for) also predict the tendency of ions to co-condense or exclude proteins in the dense phase. Predicted trends are generally in line with the measurement for Cparin1 [sic]. Next, the authors seek to explain the observed difference in phase separation when Arginines are replaced by Lysines creating different variants. In the current rG-RPA type models both Arginine (R) and Lysine (K) are treated equally since non-electrostatic effects are only modeled in a meanfield manner that can be fitted but not predicted. For this reason, coarse grain MD simulation is suitable. Moreover, MD simulation affords structural features of the condensates. They used a force field that is capable of discriminating R and K. The MD predicted degrees of LLPS of these variants again is consistent with the measurement. One additional insight emerges from MD simulations that a negative ion can form a bridge between two positively charged residues on the chain. These insights are not possible to derive from rG-RPA. Both rG-RPA and MD simulation become cumbersome when considering multiple types of ions such as Na, Cl, [ATP] and [ATP-Mg] all present at the same time. FTS is well suited to handle this complexity. FTS also provides insights into the co-localization of ions and proteins that is consistent with NMR. By using different combinations of ions they confirm the robustness of the prediction that Caprin1 shows salt-dependent reentrant behavior, adding further support that the differential behavior of Caprin1, and pY-Caprin1 is likely to be mediated by charge-charge interactions.

We are encouraged by this reviewer’s positive assessment of our manuscript.

**Reviewer #1 (Recommendations For The Authors):**
Analysis:Analyze the simulation results to distinguish net charge neutralization and interchain bridging; see MacAinsh et al.

Please see response above to points (3) and (4) under “Weaknesses” in this reviewer’s public review. We have now added a 1.5-page subsection starting from the bottom of p. 12 to the top of p. 14 to discuss a new extensive analysis of Arg^+^–Cl^−^–Arg^+^ configurations to identify bridging interactions, with key results reported in a new Fig. 6 (p. 42). Recent results from MacAinsh, Dey & Zhou (cited now as ref. 72) are included in the added discussion. Relevant advances made in MacAinsh et al., including clarification and classification of salt-mediated interactions in the phase separation of A1-LCD are now mentioned multiple times in the revised manuscript (p. 5, lines 19–20; p. 6, lines 2–5; p. 11, line 30; p. 14, line 10; p. 18, lines 28–29; and p. 20, line 4).

Writing and presentation(1) Cite subtle effects that may be missed by the coarser approaches in this study

Please see response above to point (1) under “Weaknesses” in this reviewer’s public review.

(2) Try to distill the findings into a simple set of conclusions

Please see response above to point (2) under “Weaknesses” in this reviewer’s public review.

(3) Clarify and simplify physical interpretations

Please see response above to point (2) under “Weaknesses” in this reviewer’s public review.

(4) Explain the treatment of ATP-Mg as either a single ion or two separate ions; reconsider modifying the reference to ATP in the title

Please see response above to point (4) under “Weaknesses” in this reviewer’s public review.

(5) Minor points:p. 4, citation of ref 56: this work shows ATP is a driver of LLPS, not merely a regulator (promotor or suppressor)

This citation to original ref. 56 (now ref. 63) on p. 4 is now corrected (bottom line of p. 4).

p. 7 and throughout: “using bulk [Caprin1]” – I assume this is the initial overall Caprin1 concentration. It would avoid confusion to state such concentrations as “initial” or “initial overall”

We have now added “initial overall concentration” in parentheses on p. 8 (line 4) to clarify the meaning of “bulk concentration”.

p. 7 and throughout: both mM (also uM) and mg/ml have been used as units of protein concentration and that can cause confusion. Indeed, the authors seem to have confused themselves on p. 9, where 400 (750) mM is probably 400 (750) mg/ml. The same with the use of mM and M for salt concentrations (400 mM Mg2+ but 0.1 and 1.0 M Na+)

Concentrations are now given in both molarity and mass density in Fig. 1 (p. 37), Fig. 2 (p. 38), Fig. 4 (p. 40), and Fig. 7 (p. 43), as noted in the text on p. 8 (lines 4–5). Inconsistencies and errors in quoting concentrations are now corrected (p. 10, line 18, and p. 11, line 2).

p. 7, “LCST-like”: isn’t this more like a case of a closed coexistence curve that contains both UCST and LCST?

The discussion on p. 8 around this observation from Fig. 1d is now expanded, including alluding to the theoretical possibility of a closed co-existence curve mentioned by this reviewer, as follows:

“Interestingly, the decrease in some of the condensed-phase [pY-Caprin1]s with decreasing *T* (orange and green symbols for ≲ 20◦C in Fig. 1d trending toward slightly lower [pY-Caprin1]) may suggest a hydrophobicity-driven lower critical solution temperature (LCST)-like reduction of LLPS propensity as temperature approaches ∼ 0◦C as in cold denaturation of globular proteins [7,23] though the hypothetical LCST is below 0◦C and therefore not experimentally accessible. If that is the case, the LLPS region would resemble those with both an UCST and a LCST [4]. As far as simple modeling is concerned, such a feature may be captured by a FH model wherein interchain contacts are favored by entropy at intermediate to low temperatures and by enthalpy at high temperatures, thus entailing a heat capacity contribution in *χ*(*T*), with ϵh→ϵh(T),ϵs→ϵs(T) [7,109,110] beyond the temperature-independent *ϵh* and *ϵs* used in Fig. 1c,d and Fig. 2. Alternatively, a reduction in overall condensed-phase concentration can also be caused by formation of heterogeneous locally organized structures with large voids at low temperatures even when interchain interactions are purely enthalpic (Fig. 4 of ref. [111]).”

p. 8 “Caprin1 can undergo LLPS without the monovalent salt (Na+) ions LLPS regions extend to [Na+] = 0 in Fig. 2e,f”: I don’t quite understand what’s going on here. Is the effect caused by a small amount of counterion (ATP-Mg) that’s calculated according to eq 1 (with z s set to 0)?

The discussion of this result in Fig. 2e,f is now clarified as follows (p. 10, lines 8–14 in the revised manuscript):

“The corresponding rG-RPA results (Fig. 2e–h) indicate that, in the present of divalent counterions (needed for overall electric neutrality of the Caprin1 solution), Caprin1 can undergo LLPS without the monvalent salt (Na+) ions (LLPS regions extend to [Na+] = 0 in Fig. 2e,f; i.e., ρs = 0, ρc > 0 in Eq. (1)), because the configurational entropic cost of concentrating counterions in the Caprin1 condensed phase is lesser for divalent (zc = 2) than for monovalent (zc = 1) counterions as only half of the former are needed for approximate electric neutrality in the condensed phase.”

p. 9 “Despite the tendency for polymer field theories to overestimate LLPS propensity and condensed-phase concentrations”: these limitations should be mentioned earlier, along with the very high concentrations (e.g., 1200 mg/ml) in Fig. 2

This sentence (now on p. 11, lines 11–18) is now modified to clarify the intended meaning as suggested by this reviewer:

“Despite the tendency for polymer field theories to overestimate LLPS propensity and condensed-phase concentrations quantitatively because they do not account for ion condensation [99]—which can be severe for small ions with more than ±1 charge valencies as in the case of condensed [Caprin1] ≳ 120 mM in Fig. 2i–l, our present rG-RPA-predicted semi-quantitative trends are consistent with experiments indicating “

In addition, this limitation of polymer field theories is also mentioned earlier in the text on p. 6, lines 30–31.

**Reviewer #2 (Recommendations For The Authors):**
(1) he current version of the paper goes through many different methodologies, but how these methods complement or overlap in terms of their applicability to the problem at hand may not be so clear. This can be especially difficult for readers not well-versed in these methods. I suggest the authors summarize this somewhere in the paper.

As mentioned above in response to Reviewer #1, we have now added a subsection with heading “Overview of key observations from complementary approaches” at the beginning of the “Results” section on p. 6 (lines 18–37) and the first line of p. 7 to make our paper more accessible to readers who might not be well-versed in the various theoretical and computational techniques. A few sentences to summarize our key results are added as well to the first paragraph of “Discussion” (p. 18, lines 23–26).

(2) It wasn’t clear if the authors obtained LCST-type behavior in Figure 1d or if another phenomenon is responsible for the non-monotonic change in dense phase concentrations. At the very least, the authors should comment on the possibility of observing LCST behavior using the rG-RPA model and if modifications are needed to make the theory more appropriate for capturing LCST.

As mentioned above in response to Reviewer #1, the discussion regarding possible LCSTtype behanvior in Fig. 1d is now expanded to include two possible physical origins: (i) hydrophobicity-like temperature-dependent effective interactions, and (ii) formation of heterogeneous, more open structures in the condensed phase at low temperatures. Three additional references [109, 110, 111] (from the Dill, Chan, and Panagiotopoulos group respectively) are now included to support the expanded discussion. Again, the modified discussion is as follows:

“Interestingly, the decrease in some of the condensed-phase [pY-Caprin1]s with decreasing *T* (orange and green symbols for ≲ 20◦C in Fig. 1d trending toward slightly lower [pY-Caprin1]) may suggest a hydrophobicity-driven lower critical solution temperature (LCST)-like reduction of LLPS propensity as temperature approaches ∼ 0◦C as in cold denaturation of globular proteins [7,23] though the hypothetical LCST is below 0◦C and therefore not experimentally accessible. If that is the case, the LLPS region would resemble those with both an UCST and a LCST [4]. As far as simple modeling is concerned, such a feature may be captured by a FH model wherein interchain contacts are favored by entropy at intermediate to low temperatures and by enthalpy at high temperatures, thus entailing a heat capacity contribution in *χ*(*T*), with ϵh→ϵh(T),ϵs→ϵs(T) [7,109,110] beyond the temperature-independent *ϵh* and *ϵs* used in Fig. 1c,d and Fig. 2. Alternatively, a reduction in overall condensed-phase concentration can also be caused by formation of heterogeneous locally organized structures with large voids at low temperatures even when interchain interactions are purely enthalpic (Fig. 4 of ref. [111]).”

(3) In Figures 4c and 4d, ionic density profiles could be shown as a separate zoomed-in version to make it easier to see the results.

This is an excellent suggestion. Two such panels are now added to Fig. 4 (p. 40) as parts (g) and (h).

**Reviewer #3 (Recommendations For The Authors):**
I would suggest authors make some minor edits as noted here.(1) Please note down the chi values that were used when fitting experimental phase diagrams with rG-RPA theory in Figure 2a,b. At present there aren’t too many such values available in the literature and reporting these would help to get an estimate of effective chi values when electrostatics is appropriately modeled using rG-RPA.

The *χ*(*T*) values and their enthalpic and entropic components *ϵh* and *ϵs* used to fit the experimental data in Fig. 1c,d are now stated in the caption for Fig. 1 (p. 37). Same fitted *χ*(*T*) values are used in Fig. 2 (p. 38) as it is now stated in the revised caption for Fig. 2. Please note that for clarity we have now changed the notation from ∆*h* and ∆*s* in our originally submitted manuscript to *ϵh* and *ϵs* in the revised text (p. 7, last line) as well as in the revised figure captions to conform to the notation in our previous works [18, 71].

(2) Authors note “monovalent positive salt ions such as Na+ can be attracted, somewhat counterintuitively, into biomolecular condensates scaffolded by positively-charged polyelectrolytic IDRs in the presence of divalent counterions”. This may be due to the fact that the divalent negative counterions present in the dense phase (as seen in the ternary phase diagrams) also recruit a small amount of Na+.

The reviewer’s comment is valid, as a physical explanation for this prediction is called for. Accordingly, the following sentence is added to p. 10, lines 27–29:

“This phenomenon arises because the positively charge monovalent salt ions are attracted to the negatively charged divalent counterions in the protein-condensed phase.”

(3) In the discussion where authors contrast the LLPS propensity of Caprin1 against FUS, TDP43, Brd4, etc, they correctly note majority of these other proteins have low net charge and possibly higher non-electrostatic interaction that can promote LLPS at room temperature even in the absence of salt. It is also worth noting if some of these proteins were forced to undergo LLPS with crowding which is sometimes typical. A quick literature search will make this clear.

A careful reading of the work in question (Krainer et al., ref. 50) does not suggest that crowders were used to promote LLPS for the proteins the authors studied. Nonetheless, the reviewer’s point regarding the potential importance of crowder effects is well taken. Accordingly, crowder effects are now mentioned briefly in the Introduction (p. 4, line 13), with three additional references on the impact of crowding on LLPS added [30–32] (from the Spruijt, Mukherjee, and Rakshit groups respectively). In this connection, to provide a broader historical context to the introductory discussion of electrostatics effects in biomolecular processes in general, two additional influential reviews (from the Honig and Zhou groups respectively) are now cited as well [15, 16].